# Efficient and sustained optogenetic control of sensory and cardiac systems

Alexey Alekseev [1,2,3,4,5,20], Victoria Hunniford [1,3,4,5,20], Maria Zerche[1,2,3,4,6,20], Marcus Jeschke [1,4,7,20], Fadhel El May [1,3,5,20], Anna Vavakou [1,3,4,5,20], Dominique Siegenthaler [3,4,8,9,10,20], Marc A. Hüser[6,11], Svenja M. Kiehn[11], Aida Garrido-Charles [1,2,3,4], Alexander Meyer[6], Adrian Rambousky[1,4,7,12], Theocharis Alvanos[1,5], Isabel Witzke[1,2], Keila Dara Rojas-Garcia [10], Martin D. Draband[11], Lukas Cyganek [3,13,14,15], Eric Klein[16], Patrick Ruther [16,17], Antoine Huet [1,3,5,18,19], Stuart Trenholm [10], Emilie Macé [3,4,8,9], Kathrin Kusch[1,4,5,12], Tobias Bruegmann [3,4,11,14], Bettina J. Wolf [1,3,4,5] ✉, Thomas Mager [1,2,3,4] ✉ & Tobias Moser [1,3,4,5,18] ✉

Optogenetic control is used to manipulate the activity of specific cell types in vivo for a variety of biological and clinical applications. Here we report ChReef, an improved variant of the channelrhodopsin ChRmine. ChReef offers minimal photocurrent desensitization, a unitary conductance of 80 fS and closing kinetics of 30 ms, which together enable reliable optogenetic control of cells at low light levels with good temporal fidelity and sustained stimulation. We demonstrate efficient and reliable red-light pacing and depolarization block of ChReef-expressing cardiomyocyte clusters. We used adeno-associated-virus-based gene transfer to express ChReef in retinal ganglion cells, where it restores visual function in blind mice with light sources as weak as an iPad screen. Toward optogenetic hearing restoration, ChReef enables stimulation of the auditory pathway in rodents and non-human primates with nanojoule thresholds, enabling efficient and frequency-specific stimulation by LED-based optical cochlear implants.

With an ever-growing toolkit of light-sensitive proteins (opsins) and optical devices, optogenetics is routinely used to manipulate the activity of specific cell types in vivo in genetically tractable species such as mice, fish or flies. In medicine, progress in both optogenetic modification of retinal neurons and visual coding by light amplifying goggles has enabled the successful clinical application of optogenetics for vision restoration[1]. Yet, several challenges remain to realize the full potential of optogenetics. For example, the low single-channel conductance of non-selective cation-conducting channelrhodopsins (ChRs; for example, 40 fS for ChR2 (ref. [2])) limits their use. The required high levels of expression and high light doses bear the risk of proteostatic stress and phototoxicity in the target tissue[3]. In the visual system, low single-channel conductance impedes versatile optogenetic stimulation by natural light or computer screens and necessitates the use of light-amplifying goggles for optogenetic vision restoration. The energy demand of state-of-the-art optogenetic control also challenges the power budget of medical devices such as future optical cochlear implants for improved hearing restoration[3,4]. Kalium channelrhodopsins (KCRs) and anion channelrhodopsins (ACRs) with larger single-channel conductance have recently been reported serving efficient inhibition of excitable cells[5,6].

Aiming for a depolarizing ChR for efficient excitation, we targeted the cryptophyte ChR ChRmine that mediates large depolarizing photocurrents and has red-shifted light absorption ($\lambda_{max}$ = 520 nm)[7–9]. This way ChRmine has enabled control of cardiac activity of mice by light applied from outside[10] as well as implant-free deep brain optogenetics[11].

In this Article, we demonstrate that ChRmine shows a comparatively high unitary conductance. Yet, strong desensitization of ChRmine remains an unsolved key problem, impeding applications that require sustained or high-rate optogenetic stimulation. We deciphered the mechanistic underpinnings of ChRmine desensitization and overcome it in the ChRmine T218L/S220A variant ChReef ('ChR that excites efficiently') that provides advanced use when applied to the eye, the ear and cardiomyocytes of the heart.

## Results

### ChRmine shows high unitary conductance and desensitization

Using patch-clamp recordings from neuroblastoma–glioma cells (NG cells) expressing the cryptophyte ChR ChRmine fused to plasma membrane targeting sequences of the inward rectifying potassium channel $K_{ir}2.1$ (trafficking signal and export signal) and enhanced yellow fluorescent protein (eYFP) for improved plasma membrane targeting[7,12,13], we found that the stationary photocurrent of ChRmine is only about 20% of the peak current (stationary–peak ratio = 0.22 ± 0.12 ($n$ = 57; $n$ denotes number of cells); Fig. 1a,d, Extended Data Fig. 1, Supplementary Fig. 1 and Supplementary Table 1). In fact, the ChRmine stationary photocurrent density at saturating light intensity (Fig. 1b; $J_{-60 mV}$ = 21.6 ± 15.8 pA pF$^{-1}$, $n$ = 44) did not exceed that of other state-of-the-art depolarizing ChRs (Fig. 1b and Supplementary Table 2). The stationary photocurrent showed an initial increase with light intensity followed by a decrease to a submaximal value despite increasing light intensity (Fig. 1e), suggesting a substrate (photon) inhibition of the partial type[14] that had, to our knowledge, not been described previously.

As the unitary conductance of ChRs is too small to be directly determined by single-channel recordings, we investigated the variance in photocurrent ensembles using stationary and non-stationary noise analysis[15]. As ChR conductance approaches the detection limit in stationary noise analysis[2,5,6,16], we upscaled data collection by automated patch clamp (Syncropatch 384, Nanion) operated in synchrony with light-emitting diode (LED)-based illumination (Supplementary Fig. 2). This way, we could simultaneously record ensembles of photocurrents from dozens of HEK293 cells (holding potential = −100 mV) elicited by blue light for the efficient ChR2 mutant CatCh[16] or green light for ChRmine (Supplementary Fig. 2).

Strict quality measures, regarding background noise ($I_{RMS, dark}$ < 5.5 pA) and stationary photocurrent size (>200 pA), limited the number of suitable recordings to 14/149 for CatCh and 12/71 for ChRmine. Still, a sample size sufficing statistical comparison could be obtained within days, due to the highly parallelized approach. Fitting the power spectra with Lorentzian functions revealed a significantly larger single-channel conductance of ChRmine compared to CatCh (Fig. 1f,h,i; ChRmine: 88.8 ± 39.6 fS, $n$ = 12 versus CatCh: 34.8 ± 25.1 fS, $n$ = 14, $P$ < 0.001). We confirmed this finding by non-stationary noise analysis (Fig. 1g,i and Supplementary Fig. 3; 111.3 ± 33.2 fS ($n$ = 17); holding potential = −100 mV) using short (5 ms) light pulses at a rate of 0.2 Hz to avoid desensitization. The stationary noise analysis provided further insight into ChRmine function: in addition to the power spectral density in the low frequency range (2.5 to 25 Hz) it also showed a pronounced shoulder at higher frequencies (75 to 125 Hz; Fig. 1h, yellow lines). This likely indicates the presence of a second, short-lived open state. We hypothesize that this state results from absorption of a second photon, inducing a parallel low conducting photocycle that underlies the observed substrate inhibition. Parallel photocycles have been implicated in desensitization of green algal ChRs[17,18].

### Engineering and characterization of ChReef

With the objective of optimizing ChRmine for life sciences and medical applications, we investigated the effect of mutations at the homologous position (F219Y) and the adjacent positions (T218L and S220A) to the F219Y mutation in helix 6 of ChR2, which significantly accelerated channel closing in green algal ChRs[19]. The electrophysiological characterization of the ChRmine mutants was carried out by manual and automated patch-clamp experiments in NG and HEK293 cells, respectively. ChRmine F219Y showed strongly reduced photocurrents and channel-closing kinetics (ChRmine F219Y: $\tau_{off(-60 mV)}$ = 58.1 ± 5.4 ms, $n$ = 19) similar to ChRmine (ChRmine: $\tau_{off(-60 mV)}$ = 63.5 ± 15.7 ms, $n$ = 7) (Extended Data Fig. 2 and Supplementary Fig. 4). We also found similar $\tau_{off}$ values for ChRmine T218L ($\tau_{off(-60 mV)}$ = 59.1 ± 21.3 ms, $n$ = 7) and ChRmine T218L/S220A ('ChReef') ($\tau_{off(-60 mV)}$ = 58.3 ± 12.5 ms, $n$ = 7; $\tau_{off(-60 mV)}$ at 36 °C, 35 ± 3 ms, $n$ = 6) (Extended Data Fig. 2, Supplementary Table 1 and Supplementary Fig. 4), whereas ChRmine S220A had slower channel-closing kinetics ($\tau_{off(-60 mV)}$ = 152.7 ± 19.8 ms, $n$ = 6).

It is worth noting that ChRmine T218L, ChRmine S220A and ChReef showed a regular light dependence (hyperbolic, sigmoidal on log scale; Fig. 1e). Hence, the helix 6 mutants lacked the light-dependent inactivation process found in wild-type ChRmine. In line with this notion, photocurrent desensitization was strongly reduced in the mutants (Fig. 1a,d, Extended Data Fig. 1 and Supplementary Table 1): from ChRmine T218L (stationary–peak ratio = 0.44 ± 0.13, $n$ = 18), to ChRmine S220A (stationary–peak ratio = 0.62 ± 0.14, $n$ = 38) and ChReef (stationary–peak ratio = 0.62 ± 0.15, $n$ = 21). Fluorescence line profile analysis showed no obvious difference in the plasma membrane targeted expression of ChReef and ChRmine (Supplementary Fig. 5). ChReef showed the largest stationary photocurrent density, considerably exceeding most of current state-of-the-art ChRs (Fig. 1b and Supplementary Tables 1 and 2) (ChReef: 97.6 ± 65.0 pA pF$^{-1}$, $n$ = 16 versus ChRmine: 21.6 ± 15.8 pA pF$^{-1}$, $n$ = 44), except for the blue-light-activated and slowly deactivating ($\tau_{off(-60 mV)}$ = 279 ± 86 ms, $n$ = 31) green algal ChR $Co$ChR H94E/L112C/K264T ($Co$ChR-3M)[20] (Extended Data Fig. 2). We consider ChReef better suited for future optogenetic therapies than $Co$ChR-3M because of its more red-shifted action spectrum and its much faster channel-closing kinetics (Fig. 1c, Extended Data Fig. 2 and Supplementary Fig. 6). These properties reduce the risk of phototoxic effects and enables photostimulation at higher

**Fig. 1 | Electrophysiological characterization of ChRmine variants.**
**a**, Stationary–peak ratios of photocurrents of ChRmine (black circles, $n$ = 57), ChRmine S220A (red triangles, $n$ = 38), ChRmine T218L (blue rhombi, $n$ = 18) and ChReef (magenta squares, $n$ = 21). **b**, Stationary photocurrent densities of ChRmine ($n$ = 44, $N$ = 14), ChRmine S220A ($n$ = 35, $N$ = 14), ChRmine T218L ($n$ = 18, $N$ = 8), ChReef ($n$ = 16, $N$ = 5), Chronos (yellow pentagons, $n$ = 14, $N$ = 7), Chrimson (red hexagons, ref. [19]), f-Chrimson (green rhombi, ref. [19]), CatCh (blue triangles, $n$ = 11, $N$ = 7) and ChR2 T159C (cyan triangles, $n$ = 9, $N$ = 4). **c**, $\tau_{off}$ of Chronos ($n$ = 8), Chrimson ($n$ = 5 (ref. [19])), f-Chrimson ($n$ = 5 (ref. [19])), CatCh ($n$ = 9), ChR2 T159C ($n$ = 8), ChRmine ($n$ = 7) and ChReef ($n$ = 7). **d**, Representative photocurrent traces. **e**, Light intensity dependence of normalized stationary photocurrents; solid lines show hyperbolic fits (red, blue, magenta) or substrate inhibition type fit (black). **f,h**, Stationary noise analysis for ChReef (**f**) and ChRmine (**h**): power spectral density (PSD) differences between light and dark conditions, normalized to the fitted value at 0 Hz. Solid and dashed lines indicate single and double Lorentzian fits. **g**, Non-stationary noise analysis: variance versus mean current for a ChReef trace. Thin line, moving average of variance; bold line, linear fit. **i**, Statistical comparison of unitary conductance values for ChRmine (black circles and hexagons, $n_{stat}$ = 12, $n_{non.stat}$ = 17), ChReef (magenta squares and diamonds, $n_{stat}$ = 13, $n_{non.stat}$ = 15) and CatCh (blue triangles, $n_{stat}$ = 14). **j**, Comparison of PSD form factors for ChRmine ($n$ = 12) and ChReef ($n$ = 13), defined as the ratio of ΔPSD values at 7.5–12.5 Hz and 75–125 Hz. Statistical comparisons (all two-sided): Kruskal–Wallis and Mann–Whitney $U$-tests with Bonferroni correction in **a** and **b**, ANOVA with Bonferroni post hoc test in **c**, Tukey's honestly significant difference in **i**, or Student's $t$-test in **j**. Comparison bars indicate adjusted $P$ values for **a**–**c** and exact $P$ values for **i** and **j**; shown only for ChReef in **b** and **c** to reduce clutter. *$P$ < 0.05, **$P$ < 0.01, ***$P$ < 0.001, ****$P$ < 0.0001. Exact $P$ values for **a**–**c** and **i** are in Supplementary Tables 3–6; for **j**, $P$ = 1.68 × 10$^{-9}$. Symbols in **b** and bars in **a**–**c**, **i** and **j** show means; error bars represent s.d. $n$ = number of cells; $N$ = number of transfections.

1

frequencies. Ion-permeability ratio determinations from the shift of the reversal potential upon ion replacement using the Goldman–Hodgkin–Katz equation showed that the ion-permeability ratios were not significantly different in ChReef and ChRmine (Supplementary Fig. 7). ChReef is an unselective cation channel ($P_K/P_{Na} \approx 1$) that shows a considerable proton conductance ($P_H/P_{Na} \approx 10^5$) and virtually no calcium conductance. In comparison, ChR2 has a higher proton and calcium permeability (for example, ChR2 variant CatCh[16]: $P_H/P_{Na} \approx 10^6$,

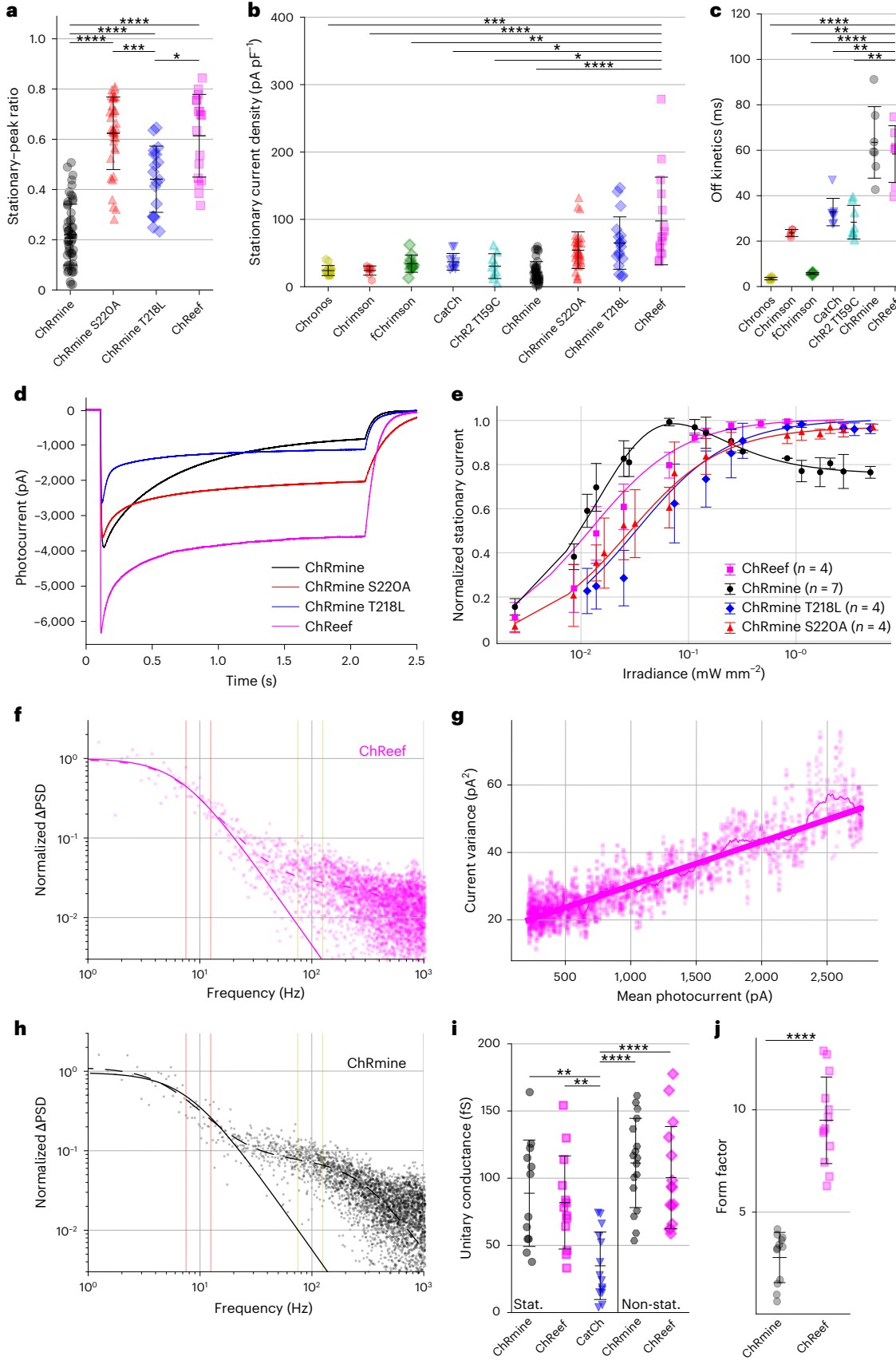

$P_{Ca}/P_{Na} \approx 0.15$[16]). Hence, ChReef provides lower risk for possible adverse effects resulting from $Ca^{2+}$ and $H^+$ influx, which may arise upon long-term optogenetic stimulation.

Next, we determined the single-channel conductance of ChReef which is statistically indistinguishable from ChRmine (Fig. 1i; $81.9 \pm 34.7$ fS, $n = 13$, stationary noise analysis and $100.5 \pm 38.0$ fS, $n = 15$, non-stationary noise analysis). The high-frequency power spectral density component found in ChRmine was not detectable for ChReef (Fig. 1f,h,j), which we propose to reflect a considerably accelerated transition of the parallel photocycle open state into the main photocycle that results in the minimization of the light-dependent inactivation (Fig. 1e). The finding that the channel-closing kinetics of ChRmine and ChReef could be fitted to a mono-exponential function indicates that the short-lived second open state does not noticeably contribute to the stationary photocurrent of ChRmine, in line with the suggested role of a parallel photocycle in photocurrent reduction. The photocurrent desensitization kinetics of the ChRmine variants at saturating light intensities could be approximated by a bi-exponential function ($\tau_{DES1} \approx \tau_{off}$, $\tau_{DES2} > 1$ s; Extended Data Fig. 3). The relative amplitude of the slowly desensitizing component was much smaller in ChReef than in ChRmine (Extended Data Fig. 3), indicating the contribution of a slow process, likely substrate inhibition of the partial type, to photocurrent desensitization in ChRmine. Photocurrent desensitization generally results from the difference in the distribution of open and closed states before (peak current at high intensities) and at steady state (stationary current). Experiments at suboptimal wavelength ($\lambda \approx 632$ nm) and subsaturating intensities (~1 mW mm$^{-2}$) showed that even without sizable peak current the stationary photocurrent of ChReef remained elevated (Supplementary Figs. 8 and 9). The combination of high unitary conductance and minimal light-dependent inactivation qualifies ChReef for efficient and sustained optogenetic stimulation.

### Optogenetic stimulation of cardiac tissue

To assess the use of ChReef for life science applications and future optogenetic therapies, we turned to the cardiac, visual and auditory systems. We first benchmarked ChReef to ChRmine for pacing cardiomyocyte clusters generated from neonatal mouse hearts and with near-complete adeno-associated virus AAV2/9 mediated ChR expression (Fig. 2a,b; $96 \pm 1.8\%$ for ChReef and $98 \pm 0.5\%$ for ChRmine, $N = 2$ isolations and $n = 5$ coverslips, unpaired Student's $t$-test, $P = 0.11$). We achieved reliable optical pacing at 1 Hz with green (510 nm, 5 ms) and red (630 nm, 50 ms) light pulses at irradiances below 100 µW mm$^{-2}$ and 350 µW mm$^{-2}$, respectively, in both ChReef- and ChRmine-cardiomyocytes (Extended Data Fig. 4a–c). Higher pacing rates increased the light requirement, in particular for red light for which we found significantly lower pacing thresholds with ChReef at rates of ≥3 Hz (Extended Data Fig. 4d–f), likely reflecting the higher steady-state currents mediated by ChReef (Supplementary Figs. 8 and 9).

Optogenetic cardiac defibrillation of ventricular arrhythmia[21–23] and cardioversion of atrial arrhythmia[22,24,25] are exciting targets for future clinical translation[26]. Here we compared the efficiency of ChReef and ChRmine for inducing a depolarization block of cardiomyocytes, considered the most effective mechanism for optogenetic termination of cardiac arrhythmia[27]. Continuous illumination with green light prevented electrically evoked contraction of cardiomyocyte clusters with significantly lower light requirements for ChReef than for ChRmine, while no effect was found in non-transduced control cardiomyocyte clusters (Fig. 2c,d). Using red light revealed an even stronger advantage of ChReef for depolarization block of cardiomyocytes: contraction was prevented in 15 out of 16 ChReef-expressing clusters compared to only 7 out of 17 ChRmine-expressing clusters (Fig. 2e,f).

In line with the results from neonatal mouse cardiomyocytes, we found similar effects in human induced pluripotent stem cell (hiPSC) derived cardiomyocytes in which the increase in required light intensities with higher pacing frequencies was also significantly more pronounced for ChRmine compared to ChReef (Supplementary Fig. 10a–d). The depolarization block was only achievable in 6 out of 19 cardiomyocyte clusters expressing ChRmine but in all ChReef-expressing cardiomyocyte clusters using red light (Supplementary Fig. 10e). This seems highly relevant, as sustained depolarization of the heart muscle upon long wavelength light would be required for future transmural optogenetic termination of arrhythmia[28].

### Optogenetic vision restoration

Optogenetic restoration of vision is currently being evaluated in clinical trials. In a phase 1/2a clinical trial (PIONEER), intravitreal injection of adeno-associated viruses (AAVs) is used to express ChrimsonR in retinal ganglion cells (RGCs)[1]. A current disadvantage of this approach is that the low light sensitivity requires light-amplifying goggles that use relatively high light intensities. This limits the dynamic range of optogenetic encoding of visual information and poses a potential risk of phototoxicity for long-term use. Here, we tested the potential of expressing ChReef and ChRmine in RGCs for low light optogenetic vision restoration in the retinal degeneration (rd1) mouse model of retinopathy[29]. The engineered AAV2/9 derivative AAV-PHP.eB carrying either of the eYFP-tagged ChRs under the control of the human synapsin (h*SYN*) promotor were intravitreally injected (Fig. 3a). Despite the use of the general neuronal h*SYN* promoter, opsin expression was largely limited to the RGC and inner plexiform layers of the retina (Extended Data Fig. 5). Opsin expression in coronal brain slices was evident for RGC axons in the lateral geniculate nucleus and superior colliculus (Supplementary Fig. 11). Responses to optogenetic stimulation of the eye to a green LED or an iPad screen positioned close to the mouse eye were recorded by multielectrode arrays inserted into the contralateral primary visual cortex (V1) 3–4 weeks after AAV administration (Fig. 3b). Single-unit V1 activity could be elicited by full-field LED flashes of 200 ms duration at 1 Hz in both ChReef and ChRmine-injected mice (Fig. 3c–f and Extended Data Fig. 6).

For both ChReef and ChRmine, we observed cells that responded at light intensities as low as 0.25 µW mm$^{-2}$ (50 nJ mm$^{-2}$; Fig. 3d (arrows) and Extended Data Fig. 6a), which is around the light intensities observed in indoor settings[30] demonstrating high light sensitivity conferred to RGCs by both ChRs. It is worth noting that ChReef-injected animals showed stronger and more sustained V1 activity compared to ChRmine, especially at low light intensities (Fig. 3c–f and Extended Data Fig. 6).

In further support of high-sensitivity vision restoration, full-field stimuli presented to the eye from an iPad screen also elicited V1 activity in both ChRmine- and ChReef-injected mice (Fig. 3g–i). Again, V1 activity was stronger and more sustained in ChReef compared to ChRmine expressing animals (Fig. 3i). This indicates that ChReef enables the use of computer screens for versatile visual stimulation in vision research and for restoring vision in blind patients at lower light intensities.

To test whether ChReef expression in the retina of rd1 mice enables behavioural responses to light, we performed the light-room/dark-room test (Fig. 3j). This tests an innate behaviour of sighted mice to spend more time in the dark-room[31,32]. First, we used epifluorescence imaging to confirm robust YFP (yellow fluorescent protein) expression in the retina of rd1 mice intravitreally injected with AAV-PHP.eB-hSYN-ChReef (Fig. 3k). Next, we placed injected rd1 mice as well as wild-type control and untreated rd1 mice in the light-room/dark-room environment. To control for a potential room bias, the experiment was repeated four times while alternating which room was lighted. As previously established[31], we found that sighted wild-type mice spent the majority of their time in the dark-room, whereas rd1 mice did not show a preference between the two rooms (Fig. 3l). It is worth noting that all rd1 mice were outside 3 standard deviations of the wild-type distribution. We thus used 3 standard deviations of the wild-type distribution as a threshold for defining whether a mouse was sighted or blind. We found that rd1-ChReef mice were below this threshold and, thus, similar to sighted mice (Fig. 3l), preferred the dark-room revealing partial restoration of vision.

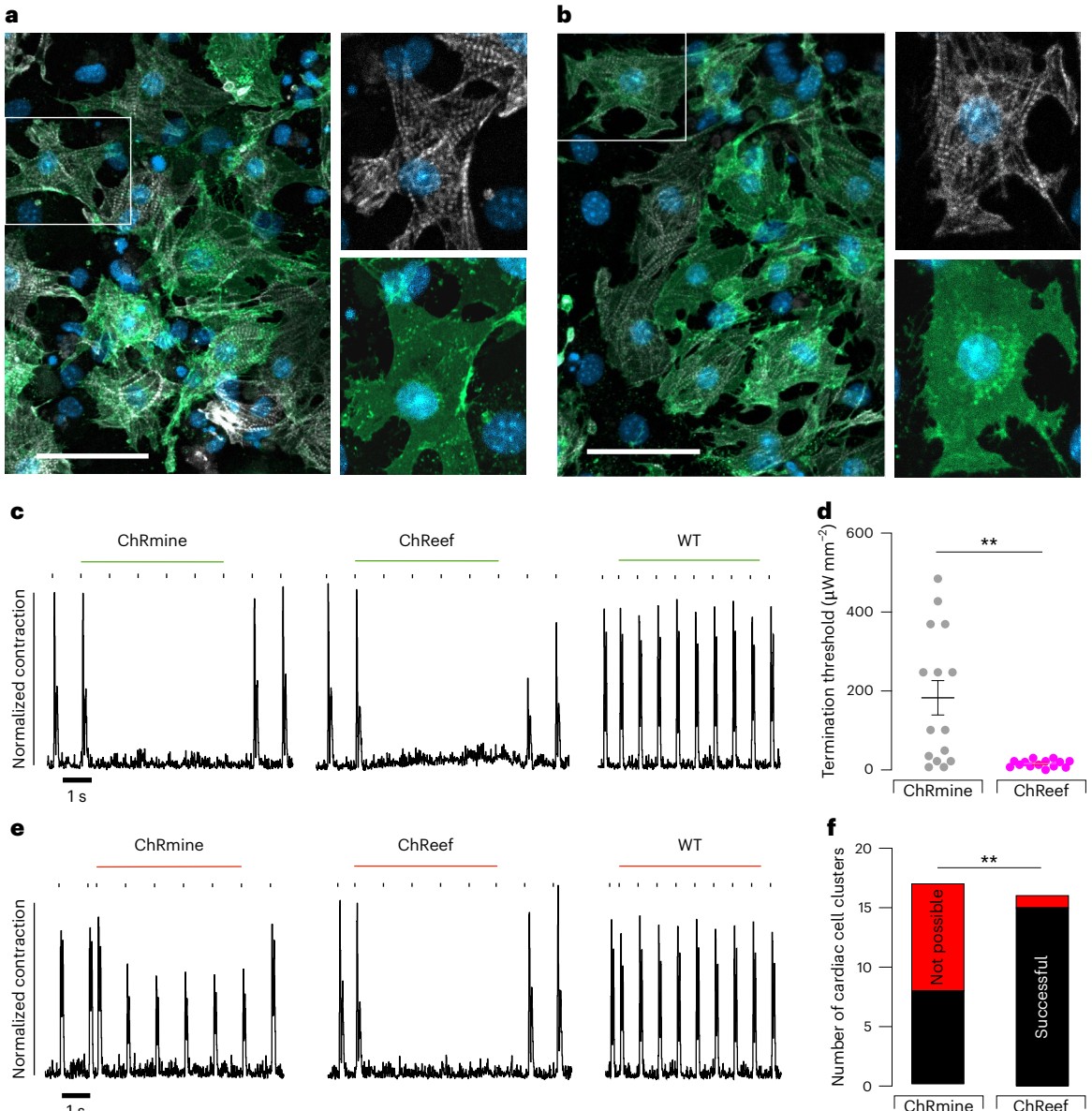

**Fig. 2 | Optical prevention of electrical excitation of cardiac cell clusters.**
**a**,**b**, Representative (≥3) images of a cardiac cell cluster comprising neonatal mouse cardiomyocytes identified by cardiac troponin I staining (white) expressing ChRmine (green, **a**) and ChReef (green, **b**) with nuclei stained by Hoechst in blue. Scale bars, 50 µm. **c**, Representative traces of contractions of cardiomyocyte clusters expressing ChRmine (left), ChReef (middle) and wild-type controls (right, representative example for 4 cardiomyocyte clusters) induced by electrical stimulation (black dots, 0.2 ms biphasic, 45 V) and prevented by continuous illumination with green light (green bar, 510 nm, 5 s; 250 µW mm$^{-2}$ for ChRmine, 14 µW mm$^{-2}$ for ChReef and 1 mW mm$^{-2}$ for wild type (WT)). Note the still existing contractions during the illumination in the wild-type example defined as unsuccessful termination. **d**, Aggregated data of the required light intensities to inhibit electrical excitation with green light for ChRmine (black) and ChReef (pink) expressing cardiomyocyte clusters. Dots indicate the results from individual cardiac clusters and the mean ± s.e.m. Statistical

comparison with a two-sided unpaired Student's *t*-test (*P* = 0.001, *N* = 3 individual isolations and AAV transductions; *n* = 15 for ChRmine expressing cardiomyocyte clusters and *n* = 14 for ChReef). **e**, Representative traces of contractions of cardiomyocyte clusters expressing ChRmine (left), ChReef (middle) and one wild-type cardiac cell cluster (right) induced by electrical stimulation (black dots, 0.2 ms biphasic, 45 V) and the effect of continuous illumination with red light (red bar, 630 nm, 5 s; 1 mW mm$^{-2}$ for ChRmine, 50 µW mm$^{-2}$ for ChReef and 1 mW mm$^{-2}$ for wild type). Note the still existing contractions during the illumination in the ChRmine example defined as unsuccessful termination. **f**, Aggregated data of the efficiency in preventing contractions with red light (maximal light intensity, 1 mW mm$^{-2}$) for ChRmine- and ChReef-expressing cardiomyocyte clusters. Statistical analysis was performed with two-sided Fisher´s exact contingency test (*P* = 0.0028; *N* = 3 individual isolations and AAV transductions; *n* = 17 individual cardiomyocyte clusters for ChRmine and *n* = 16 for ChReef). **\*\****P* < 0.01.

## Optogenetic stimulation of the auditory pathway

Optogenetics promises to improve the restoration of hearing by cochlear implants[3,4]. Electrical cochlear implants (eCIs) used in currently approximately 1 million otherwise deaf users bypass dysfunctional or lost sensory hair cells by direct electrical stimulation of spiral ganglion neurons (SGNs). eCIs typically enable one-to-one conversation in a quiet environment, yet, understanding speech in background noise remains a major unmet clinical need[33]. Optogenetic SGN stimulation could transform cochlear-implant hearing as light, different from current, can be confined in space. Hence, optical cochlear implants (oCIs) provide near-natural spectral selectivity of optogenetic SGN stimulation, outperforming the eCI[34,35]. One of the challenges for clinical

translation of the oCI is the increased power budget that results from upscaling the number of stimulation channels (64 planned in oCIs versus 12–24 in eCIs) and the higher energy required per stimulus[3,4]. Using the ChRs currently available, pulse energy thresholds for optogenetic SGN stimulation range from 1,500 nJ to 12,000 nJ (ref. 4) compared to 50 nJ for eCI[36]. Reducing the required pulse energy by greater light efficiency of optogenetic SGN stimulation remains a key objective en route to clinical translation of the oCI. Here, we used physiological analysis of the auditory system in mice and gerbils for assessing the potential of ChReef for energy-efficient optogenetic SGN stimulation.

First, we studied optogenetic stimulation at the levels of SGN population as well as of individual SGNs and cochlear nucleus neurons in C57B/6J mice 3–10 weeks after early postnatal intracochlear injection of AAV2/9 carrying ChReef or ChRmine. Expression of both ChR variants under the control of the human synapsin promotor was found in SGNs at all cochlear turns (Fig. 4b), and line profile analysis confirmed targeting to the cell membrane (Fig. 4c). The transduced SGNs per square micrometre were similar in apical and basal turns, while the proportion of transduced SGNs in the middle turn was higher in ChReef-injected cochleae (Fig. 4d; $P = 0.003$, Student's $t$-test). Optically evoked auditory brainstem responses (oABRs; Fig. 4a–g) were elicited in all mice upon stimulation with green (522 nm, optimal excitation) and orange (594 nm) light delivered by a laser-coupled optical fibre to the cochlea. oABR energy thresholds of ChReef-expressing mice (Fig. 4g; 522 nm: $170 \pm 120$ nJ, $N = 10$ versus $270 \pm 240$ nJ, $N = 10$ for ChRmine; $P = 0.25$, Student's $t$-test) were a magnitude lower than the previously reported lowest threshold found with the fast red-light activated ChR f-Chrimson[19] ($1,500 \pm 100$ nJ, $P < 0.0005$). ChReef oABR thresholds were significantly lower than those for ChRmine for longer-wavelength light (594 nm, $1,250 \pm 1,070$ nJ, $n = 16$ versus $2,450 \pm 1.160$ nJ, $N = 13$; $P = 0.008$, $t$-test) relevant for future clinical oCI. Next, we investigated the potential of ChReef for optogenetic restoration of auditory function in two mouse models of deafness: (1) acute ototoxic (kanamycin) ablation of hair cells and (2) genetic disruption of otoferlin, a synaptic protein essential for synaptic sound encoding[37]. oABR thresholds and latencies remained unaltered in both models (Fig. 4g and Extended Data Fig. 7). To evaluate the temporal fidelity of optogenetic SGN stimulation, we recorded oABRs in response to green (522 nm) and orange (594 nm) light pulses at different stimulation rates (1 ms, 10 mW; Fig. 4i and Supplementary Fig. 12) and pulse durations (Fig. 4h and Supplementary Fig. 12). oABR amplitudes declined with increasing stimulation rate, whereby ChReef showed detectable oABRs to stimulation rates up to 100 Hz compared to 50 Hz for ChRmine. Both ChReef and ChRmine enabled oABRs in response to 20-µs-long durations of green light pulses whereby the duration eliciting the maximal $P_1$-$N_1$ oABR amplitude, defined as the difference between the positive and the following negative peak of the first wave, was shorter for ChReef (Fig. 4i; 200 µs for ChReef versus 600 µs).

To further scrutinize the temporal SGN response to ChReef-mediated optogenetic stimulation (1 ms, 6 mW, 20 Hz, delivered as for oABR), we turned to in vivo recordings from single 'putative SGNs' (henceforth termed SGNs)[19,38,39] (Fig. 4a). Approximately two-thirds of the SGNs responded with a single spike and the other third with multiple spikes (mostly two to three but some with more tonic discharge; Fig. 4j,l–n), reminiscent of responses of CatCh-expressing SGNs in Mongolian gerbils[40]. Single-SGN energy thresholds (Fig. 4k; <2,000 nJ) tended to be higher than oABR thresholds but still much lower than all previously reported values (>6,000 nJ, for the state-of-the-art ChRs f-Chrimson, vf-Chrimson and Chronos[39,41]). First spike latency and first spike latency jitter were comparable among the two SGN groups (Fig. 4l,m). Next, we studied spike probability and spike synchronization with the stimulus (vector strength) as a function of stimulation rate for both SGN groups (Fig. 4n–p). Consistent with the oABR, many SGNs failed to reliably follow stimulus rates beyond 50 Hz (Fig. 4i). Both recordings of oABRs and single SGNs entailed optogenetic stimulation over several hours without obvious loss of responsiveness along the experiment. Finally, we performed patch-clamp recordings of optogenetically (green-light) driven synaptic transmission from ChReef-expressing SGNs to bushy cells in acute slices of the cochlear nucleus (Extended Data Fig. 8). Excitatory postsynaptic recordings were reliably evoked for rates of optogenetic stimulation ≤50 Hz.

To assess the potential of ChReef for future optogenetic hearing restoration, in a preliminary set of experiments, we measured the spectral spread of excitation in the cochlea, a limiting factor of eCIs. We performed multielectrode array recordings of multi-unit activity in the inferior colliculus (ref. 34; Fig. 5a and Supplementary Fig. 14) of Mongolian gerbils in response to green light (522 nm) elicited by either an optical fibre or LED-based multichannel cochlear implants[42]. Following postnatal cochlear injection of the engineered AAV2/9 derivative PHP.S carrying the human $SYN$ promotor and ChReef (AAV-PHP.S-hSYN-ChReef) (Fig. 5b and Supplementary Fig. 13), we recorded ChReef-mediated oABRs and moved on to inferior colliculus recordings in gerbils with low oABR thresholds ($215.5 \pm 153$ nJ, $n = 10$; Supplementary Fig. 14). The energy threshold for 1 ms light stimuli to elicit multi-unit activity in the inferior colliculus ($d'$ of 1) amounted to $101 \pm 73$ nJ for optical fibre stimulation ($N = 10$) and $238.5 \pm 235$ nJ when stimulating with single LEDs (531 nm, 11 LEDs in $n = 2$ gerbils; Fig. 5c and Extended Data Fig. 9c). The temporal fidelity of multi-unit activity (Supplementary Fig. 15) was consistent with results obtained in mouse SGNs (Fig. 4). The spread of excitation with fibre and LED stimulation was comparable to the natural hearing that is found with pure tones (Fig. 5d–g and Extended Data Fig. 9d). We found a clear shift of the centre of excitation when addressing LEDs of different tonotopic positions (Extended Data Fig. 9a,b). We then turned to ototoxic deafening confirmed with recordings among acoustic stimulation (Fig. 5c and

**Fig. 3 | ChReef-mediated vision restoration in rd1 mice has low light requirements. a**, Expression of ChRmine and ChReef in retinas of rd1 mice after intravitreal injection. Scale bars, 0.5 mm. **b**, Example brain slice showing the recording site in V1. Scale bar, 0.5 mm. **c**, Raster plots of two example cells. **d**, Normalized average firing rates of V1 cells in response to varying light intensities. Line and shaded area represent mean ± s.e.m. Grey lines indicate time clusters of significant difference between the two conditions (permutation test (two-sided); see Methods; from top to bottom, Bonferroni–Holm corrected, $P = 0.4062/0.5021/0.0162/0.0042/0.0048/0.0066$; $n = 151$ cells in 6 mice (ChRmine), $n = 123$ cells in 6 mice (ChReef). **e**, Average normalized firing rate during the stimulus. Mann–Whitney $U$-test (two-sided); from top to bottom, Bonferroni–Holm corrected, $P = 0.2349/0.1468/0.0671/0.0451/0.0001/0.0049$; $n = 151$ cells in 6 mice (ChRmine), $n = 123$ cells in 6 mice (ChReef). Central line indicates median, box includes data from 25th to 75th percentiles, and the whiskers span the most extreme data points without outliers (empty circles). **f**, Average normalized firing rate during the stimulus as in **e**. Error bars represent ± s.e.m.; Mann–Whitney $U$-test (two-sided); from top to bottom, Bonferroni–Holm corrected, $P = 0.2349/0.1468/0.0671/0.0451/0.0001/0.0049$; $n = 151$ cells in 6 mice (ChRmine), $n = 123$ cells in 6 mice (ChReef). **g**, Schematic depicting the experimental approach for **h** and **i**. **h**, Raster plots of two example V1 neurons. **i**, Normalized firing rates of V1 single units in response to the iPad-delivered full-field flash, $n = 90$ cells in 5 mice (ChRmine), $n = 110$ cells in 6 mice (ChReef). Line and shaded area represent mean ± s.e.m. Inset shows the average normalized firing rate during the stimulus; Mann–Whitney $U$-test (two-sided); $P = 0.0093$. **j**, Schematic of the behavioural experiment. **k**, Epifluorescence fundus images from an uninjected rd1 mouse (left) and a ChReef-injected rd1 mice (right). Scale bar, 0.5 mm. **l**, Quantification of the percentage of time spent in the light-room for sighted (wild type; $n = 9$), rd1 ($n = 11$) and rd1-ChReef ($n = 10$) mice. Circles represent individual mice, and the horizontal lines represent the mean. The dotted horizontal line represents 3 s.d. from the mean for the wild-type distribution. *$P < 0.05$, **$P < 0.01$, ***$P < 0.001$. Credit: **b,g**, Julia Kuhl; **j**, Science Figures.

Supplementary Fig. 14e). The energy threshold for 1 ms light stimuli to elicit multi-unit activity (that is, a *d*′ of 1) amounted to 345.4.6 ± 349 nJ (Fig. 5c), and the spectral spread remained comparable (Extended Data Fig. 9d).

In an effort to further evaluate the potential of ChReef for clinical translation, we turned to the common marmoset as a non-human primate model (Fig. 5h). AAV-PHP.eB-hSYN-ChReef along with kanamycin was administered into the left cochlea of mature marmosets

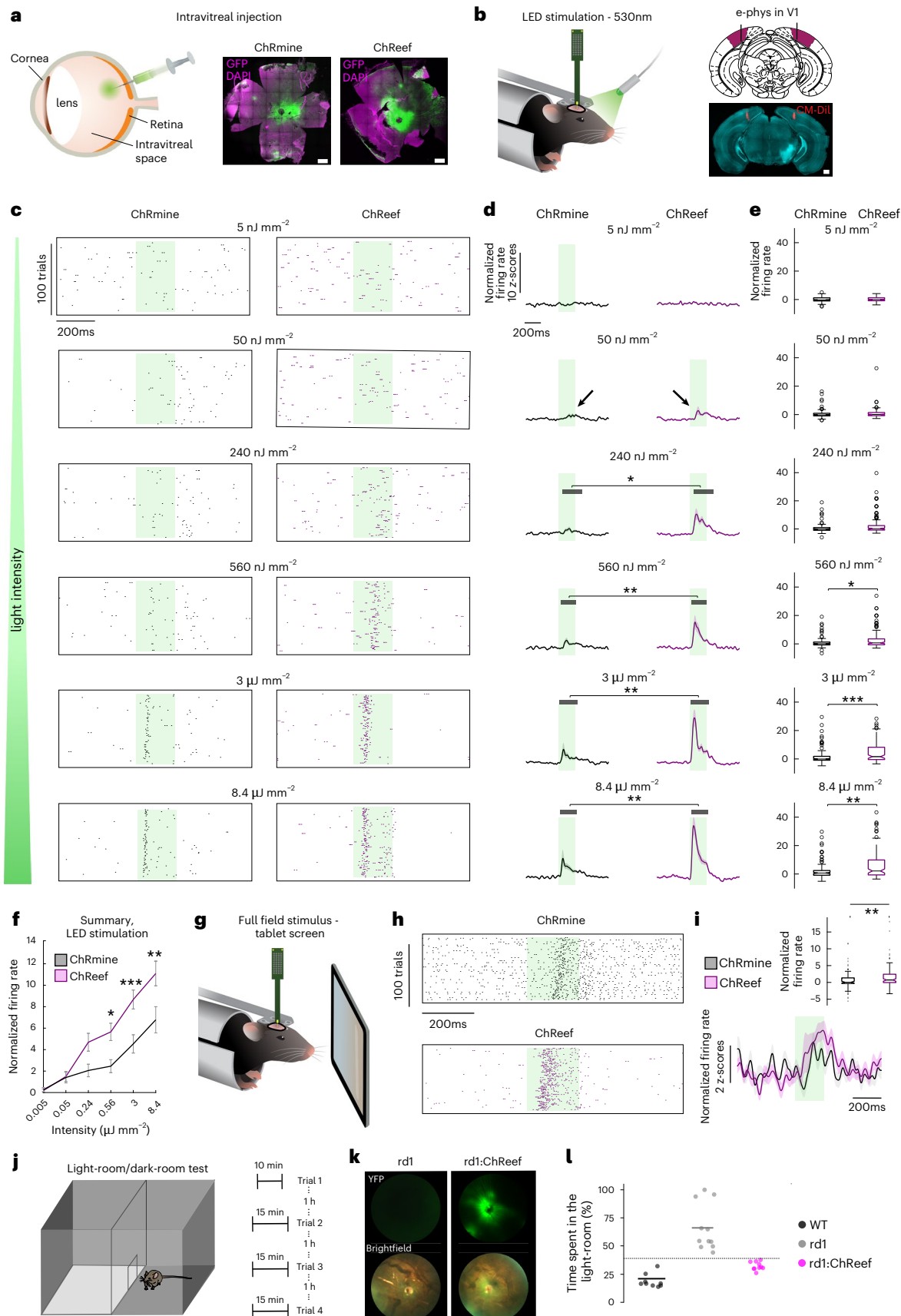

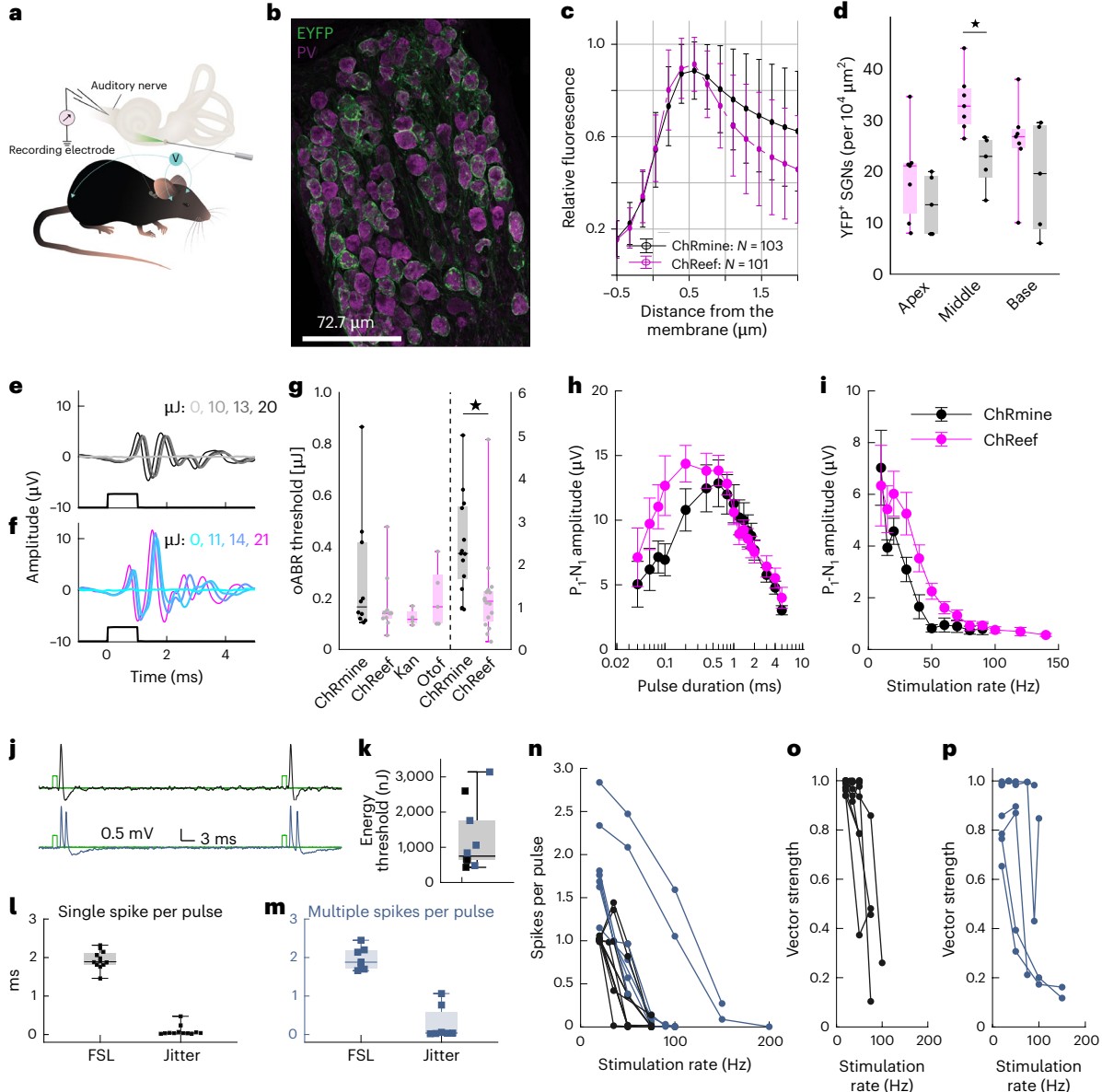

**Fig. 4 | ChReef-mediated optogenetic stimulation of the mouse auditory nerve has low light requirements and acceptable temporal fidelity. a**, Schematic representation of optical auditory brainstem recordings (oABR) and juxtacellular recordings from single putative SGNs. **b**, Mid-modiolar cochlear slice at the apical turn of exemplary ChReef-injected cochleae (representative of $N = 10$ ChRmine-injected cochleae and $N = 10$ ChReef-injected cochleae) showing SGNs (magenta, parvalbumin (PV)) and transduction (green, enhanced yellow fluorescent protein (eYFP)). **c**, Line profile analysis in single SGNs for ChRmine- and ChReef-expressing SGNs. **d**, Density of GFP-positive cells across all cochlear turns for ChRmine (black; $N = 5$) and ChReef (magenta; $N = 7$) injected mice; for the middle turn, $P = 0.003$, two-sided $t$-test. **e,f**, Exemplary oABRs driven with varying radiant flux (colours code the radiant energy in μJ) for exemplary mice injected with ChReef (**e**) or ChRmine (**f**). **g**, Energy threshold of oABRs in wild-type C57B/6J mice with ChRmine (black; $N = 10$) or with ChReef (magenta; $N = 10$), with local kanamycin injection (Kan; $N = 4$), and in otoferlin-knockout

mice (Otof; $N = 5$) using 522 nm (left, presented on separate axis) and 594 nm (right; $n = 15$ for ChReef and $n = 12$ for ChRmine at 594 nm; $P = 0.008$, two-sided $t$-test). **h**, $P_1$-$N_1$ amplitudes of oABRs for varying pulse durations using ~10 mW pulses at 10 Hz. **i**, $P_1$-$N_1$ amplitudes of oABRs at varying repetition rates using 1 ms, 10 mW pulses. For **h** and **i**, data represent mean ± s.d. with 522 nm (ChReef, magenta, $N = 10$; ChRmine, black, $N = 10$). **j**, Exemplary responses of SGNs to 1 ms, ~6 mW pulses at 20 Hz illustrating two SGN response categories: single spike responding (black) and multiple spikes responding SGNs (blue). **k**, Energy threshold of individual SGNs and median threshold ($n = 10$). **l,m**, Median of first spike latency and first spike jitter for single spike (**l**) ($n = 11$) and for multiple spikes (**m**) ($n = 7$) responding SGNs. **n,o**, Spike probability (**n**) and vector strength (**o**) of single spike and (**p**) multiple spikes responding SGNs at varying stimulation rates (samples size as in **l** and **m**). In **j**–**p**, data from three animals ($N = 3$). In **d**,**g** and **k**–**m**, box shows 25th, 50th (median) and 75th percentiles; solid line spans data range. *$P < 0.05$. Credit: **a**, Julia Kuhl.

from a micropump via a microcatheter inserted into the scala tympani via the round window from a posterior tympanotomy. To ensure efficient administration of the viral suspension and to avoid cochlear pressure build-up, a vent was generated next to the stapes footplate or by stapes removal rupturing the oval window membrane. Eleven weeks (75 ± 19 days; means ± s.d.) later, we tested for oABRs to green light stimulation from an optical fibre placed into the round

window and cochleostomies (Extended Data Fig. 10a,b). This way, we could demonstrate first proof of concept for optogenetic hearing in one out of nine marmosets (Fig. 5j). The latencies of oABR waves (Supplementary Fig. 16) were generally compatible with them reflecting auditory brainstem response (ABR) waves I, III and V but tended to be longer than acoustic ABRs[43], which also showed larger amplitudes (Supplementary Fig. 16). No responses were elicited by directly illuminating

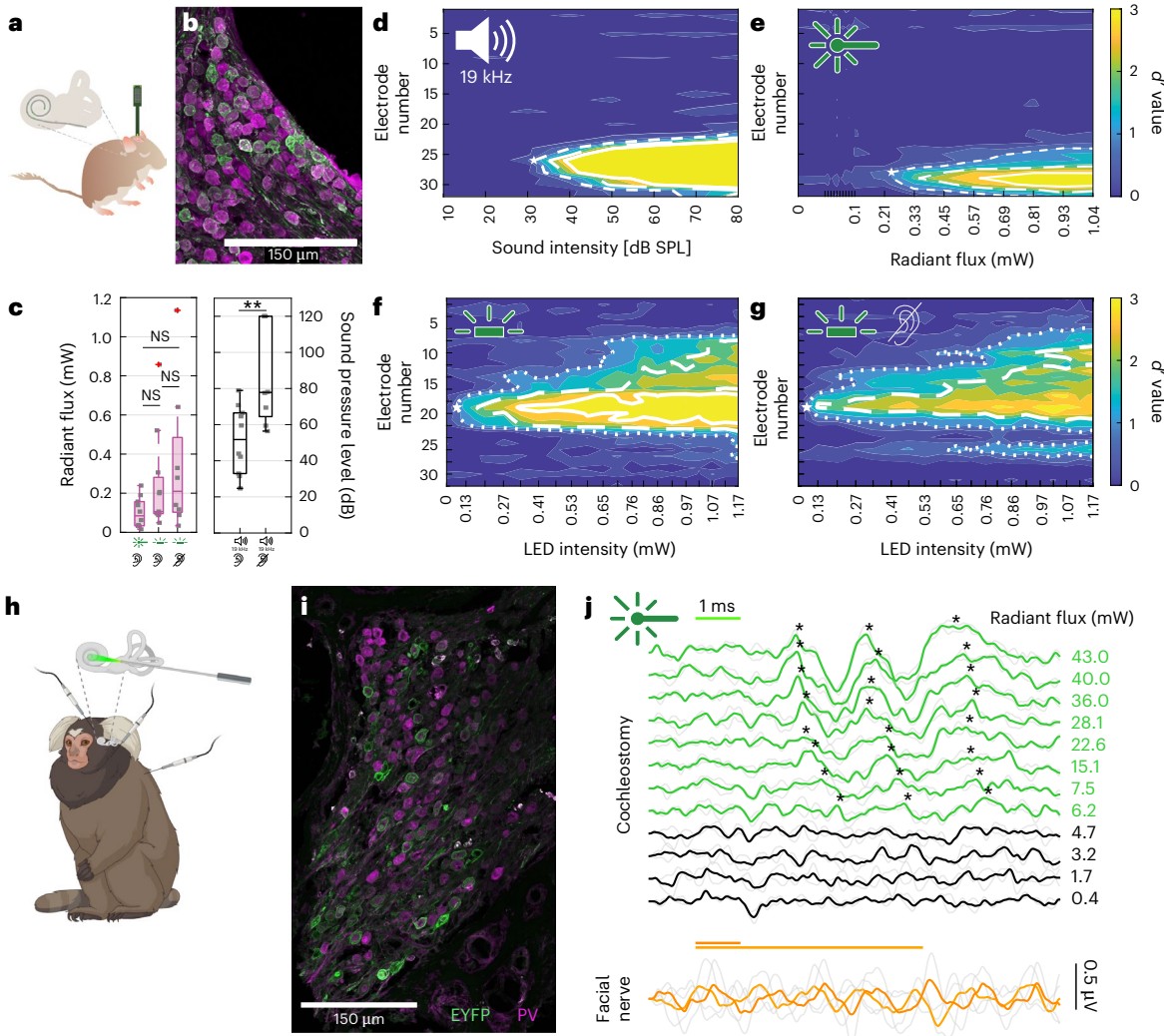

**Fig. 5 | Characterizing the optogenetic activation of the auditory pathway by inferior colliculus recordings in Mongolian gerbil and ABR in the common marmoset. a**, Schematic of experimental set-up. **b**, Example image of injected cochlea of gerbil showing ChReef (GFP; green) expression (representative of *N* = 7 ChReef-injected cochleae) in SGN (red, parvalbumin). Scale bar, 150 μm. **c**, Neuronal thresholds computed as level eliciting a *d'* of 1. Type of stimulation from left to right: optical fibre (522 nm, *N* = 10); LED (11 individual LEDs, *N* = 2); LED after deafening (8 detectable LEDs, *N* = 2); acoustic stimulation at 19 kHz before (*N* = 10) and after deafening (*N* = 8; *P* = 0.0088, Welch's paired *t*-test). For the multiple group comparison, a two-tailed Kruskal–Wallis test was used (*P* = 0.1151, no significant difference found, NS). Box shows 25th, 50th (median) and 75th percentiles; dashed line spans data range. \*\**P* < 0.01. **d**–**g**, Exemplary spatial tuning curves for different stimulus modalities: 19 kHz stimulation (**d**), laser pulses delivered via optical fibre (**e**) and optical stimulation delivered by single LED before (**f**) and after deafening (**g**). White stars indicate the electrode with the

smallest intensity to reach *d'* of 1 (best electrode). **h**, Schematic of experimental set-up for marmosets. **i**, Confocal imaging of ChReef expression in the apical turn of the left cochlea of the animal with positive ABR (1 out of 9); 30% of SGNs identified with parvalbumin (red) and weakly expressed eYFP. **j**, oABR recordings in a common marmoset (1 positive out of 9 animals) reflecting successful activation of the auditory pathway by ChReef-mediated photostimulation of the primate cochlea. An optical fibre emitting green light (525 nm, top panel) was inserted through a cochleostomy enlarging the round window niche. Auditory responses were observed at ≤6.2 mW with 1-ms-long pulses while no responses were found even with 43 mW green light pointing directly at the facial nerve with pulses of 1 and even 5 ms duration (bottom panel). Identified peaks of the responses are marked by asterisks, likely representing ABR waves I (SGNs), III (cochlear nucleus) and V (midbrain). The number of trials resulting in an average response (green, black or orange thick lines) were split into first and second halves (thin grey lines). Panel **h** created with BioRender.com. Credit: **a**, Julia Kuhl.

the neighbouring facial nerve with green light at maximally possible radiant fluxes. Histological examination revealed ChReef expression in SGNs of the treated (Fig. 5i, left) cochlea in three out of nine animals (expression rate, 33% (oABR positive), 23.3% and 0.7% SGN (oABR negative)). None of the SGNs across all turns of the right cochlea were positive (Extended Data Fig. 10). Immunohistochemical analysis of whole mounts of the organ of Corti revealed near-complete loss of hair cells on the kanamycin-injected side (Supplementary Fig. 17). Together, these data prove the principle of nanojoule and sustained optogenetic stimulation of the rodent and primate auditory pathway (microjoule for the preliminary primate report), capitalizing on the comparatively high single-channel conductance and sustained photocurrents of ChReef.

## Discussion

In this study, we engineered ChReef, a ChRmine variant with reduced photocurrent desensitization. We characterized ChReef in depth and propose it as optogenetic actuator for sustained optogenetic control of excitable cells enabling firing rates of up to 50–100 Hz with nanojoule light requirements, which we expect to facilitate a wide range of optogenetic applications. Throughout the study, we compared ChReef to ChRmine under identical experimental conditions. Like ChRmine, ChReef has good plasma membrane targeting, higher single-channel conductance than green algal ChRs, and moderate open times. Unlike ChRmine, ChReef has low photocurrent desensitization, to which we attribute improved use of ChReef in different optogenetic applications

including efficient red-light optogenetic depolarization block of cardiomyocytes, optogenetic vision and hearing restoration at lower light intensities. For vision, both ChReef- and ChRmine-mediated optogenetic stimulation of the retina of blind rd1 mice by an iPad screen elicited neural activity in the visual cortex, which was more robust for ChReef. ChReef- and ChRmine-mediated optogenetic stimulation of the cochlea lowered the energy threshold for activation of the rodent auditory pathway by one order of magnitude. We note that optical SGN thresholds also depend on the level of plasma membrane expression of the ChR achieved. This is co-determined by the time point, route and dose of viral vector administration as well as the choice of vector and promoter and, last but not least, the membrane targeting of the ChR. This study found lower (nanojoule) thresholds in three rodent models of deafness (mouse, ototoxic and genetic; gerbil, ototoxic) with two different AAVs (mouse, AAV2/9; gerbil, AAV-PHP.S) applied early postnatally with doses of ~3–7 × 10[13] viral genomes achieving expression rates similar to previous studies using similar titres of other powerful vectors such as AAV2/6 and the engineered AAV2/9 derivatives AAV-PHP.B and AAV-PHP.eB carrying the human synapsin promotor[13,19,39,41]. Like in previous studies[13,39,41] we used trafficking sequences of the inward rectifying potassium channel $K_{ir}$2.1 to improve the plasma membrane targeting. Together, we conclude that the lower thresholds are primarily caused by the favourable properties of ChReef rather than by differences in the AAV-mediated gene transfer.

When using longer-wavelength (orange) light, ChReef enabled lower energy thresholds of oABRs than ChRmine. We then turned to evaluating the use of ChReef for future optogenetic hearing restoration. We used two mouse models of human deafness: acute ototoxic deafening and otoferlin-related auditory synaptopathy, and we could elicit oABRs with comparable energy thresholds, serving as an important control of oABRs to reflect direct SGN stimulation, rather than a consequence of potential optoacoustic or optogenetic hair cell stimulation. Recordings from the auditory midbrain of Mongolian gerbils revealed efficient ChReef-mediated activation of the auditory pathway with sub-microjoule energy thresholds comparable to those of oABRs in mice and successful stimulation by LED-based oCIs. By contrast, oABRs in a marmoset that underwent microcatheter-dosing of AAV at adult age showed higher thresholds, which we attribute to lower expression rate (30% compared to 72–86% transduction across each turn for mice and 36–80% transduction across each turn for gerbils) and thicker modiolar bone covering SGNs compared to the rodents. This indicates the need for further improvement of gene therapy of SGNs for achieving sufficient ChR expression in preparation for clinical translation. Future efforts include the search for potent AAV capsids, for example, by capsid peptide library screens[44,45], and promotors as well as refined administration of the viral vectors.

The closing kinetics of ChReef and ChRmine (~30 ms deactivation time constant) provide a good match to the temporal fidelity required for optogenetic control of cardiomyocytes and RGCs. For the auditory system, the temporal fidelity of SGN stimulation was inferior to that observed with ultrafast ChRs such as f-Chrimson, vf-Chrimson and Chronos (deactivation time constants of 3, 1.5 and <1 ms at body temperature, respectively)[13,19,39,41]. Current eCI coding strategies use high stimulation rates (~1 kHz) for sampling the envelope of the sound signal. However, good speech understanding was obtained at much lower stimulation rates (from 20 Hz (ref. 46)) in the range achievable by ChReef-mediated SGN stimulation (up to ~100 Hz). Nonetheless, future efforts should aim for generating ChReef variants with faster closing kinetics to enhance the temporal fidelity of optogenetic stimulation and to shift the absorption to longer wavelengths for better alignment with red-light laserdiode-based medical devices such as oCIs. Moreover, enhancing target cell specific expression of ChReef variants by developing optimal combinations of capsids and promoters and derisking the transgene by removing the fluorescent tag[47] remain important objectives for future studies before clinical translation.

## Methods

### ChR variant generation

The humanized DNA sequence, coding for *Rl*CCR1-309 (*Rhodomonas lens* ChR, also known as ChRmine, accession number MN194599 (refs. 7,9)), C-terminally fused to TS-eYFP-ES was cloned into the mammalian expression vector pcDNA3.1 (−) (Invitrogen). The previously described targeting sequences of the inward rectifying potassium channel $K_{ir}$2.1 (trafficking signal (and export signal) were used for optimized plasma membrane expression[7,12,13]. The mutations F219Y, T218L and S220A were introduced into ChRmine by site-directed mutagenesis using the primers shown in Supplementary Table 7.

The pcDNA3.1(−) derivatives carrying the humanized DNA sequences of Chronos-eYFP (*Stigeoclonium helveticum* ChR, accession number KF992040 (ref. 48)), ChR2-eYFP (C-terminally truncated variant Chop2-315 of ChR2 from *Chlamydomonas reinhardtii*, accession number AF461397 (ref. 49)), CatCh-eYFP (ChR2 L132C), Chrimson-eYFP (*Chlamydomonas noctigama* ChR, accession number KF992060 (ref. 48)) and f-Chrimson-eYFP (Chrimson Y261F/S267M (ref. 39)) were provided by E. Bamberg (Max Planck Institute of Biophysics). The humanized DNA sequence, coding for *Co*ChR (*Chloromonas oogama* ChR, accession number KF992041 (refs. 48,50), Addgene plasmid 59070), C-terminally fused to eGFP and the humanized DNA sequence, coding for *Gt*CCR4 (*Guillardia theta* ChR 4, accession number MF039475.1 (refs. 50,51)), C-terminally fused to eYFP, were cloned into the mammalian expression vector pcDNA3.1 (−) (Invitrogen). The humanized DNA sequence, coding for *Gt*CCR2 (*G. theta* ChR2, accession number XM_005841372 (ref. 51)) C-terminally fused to eYFP, cloned into the mammalian expression vector pcDNA3.1 (−), were obtained from Addgene (plasmid 159121). The previously described ChR variants ChR2 T159C[19], ChRmine Y260F[9] and *Co*ChR H94E/L112C/K264T[52] were generated by site-directed mutagenesis using the primers shown in Supplementary Table 7.

### In vitro measurements

**Manual patch-clamp recordings and data analysis.** The electrophysiological characterization of the ChR variants was conducted by whole-cell patch-clamp recordings of transiently transfected NG108-15 cells (Supplementary Note 1) at membrane potential of −60 mV, if not stated differently. Cells were patched 2 days after transfection under voltage-clamp conditions using the Axopatch 200B amplifier (Axon Instruments) and the DigiData1322A interface (Axon Instruments). The bath solution contained 140 mM NaCl, 2 mM $CaCl_2$, 2 mM $MgCl_2$, 10 mM HEPES, pH 7.4; the pipette solution contained 110 mM NaCl, 2 mM $MgCl_2$, 10 mM EGTA, 10 mM HEPES, pH 7.4. All recordings were performed at room temperature (20 °C).

The light intensity dependence of the stationary photocurrent of the ChRmine variants was investigated by photocurrent measurements upon illumination with 2 s light pulses ($\lambda$ = 532 nm). The $EC_{50}$ values were determined by hyperbolic fitting. For wild-type ChRmine, the fitting range was thereby restricted to the range, which could be approximated by a hyperbolic function. The light intensity dependence of wild-type ChRmine was furthermore fitted to an equation corresponding to a steady state minimal kinetic model describing a substrate inhibition of the partial type[14]:

$$I = I_{\max} \frac{\left(1 + \frac{E}{K_i} R_c\right) E}{\left(1 + \frac{E}{K_i}\right) E + K_a} \tag{1}$$

where $I$ = stationary photocurrent, $I_{\max}$ = maximal stationary photocurrent if $K_i \to \infty$, $E$ = irradiance, $R_c$ = effective ratio of single-channel currents ($i_1/i_2$), $K_a$ = effective equilibrium constant for light-dependent closed (C) to open transitions (O1), and $K_i$ = effective equilibrium constant for light-dependent transitions between open states (O1/O2). The light intensity dependence shown in Fig. 1 was fitted with the

following parameters: $K_a = 0.021 \pm 0.005$, $K_i = 0.062 \pm 0.029$ and $R_c = 0.47 \pm 0.05$. Although the informative value of the fitted parameters is limited due to the absence of a more detailed description of the ChRmine photocycle, which could serve as a basis for a precise kinetic model, the fit shows that the light intensity dependence of the stationary photocurrent of ChRmine resembles a substrate inhibition of the partial type.

The stationary current densities, stationary–peak ratios, peak recovery, the action spectra and the closing kinetics were determined as described in Supplementary Note 2.

**Automated patch-clamp recordings and data analysis.** Automated patch clamp of transiently transfected cells (Supplementary Note 1) was performed using the SyncroPatch 384 system (Nanion; Supplementary Note 3). The robot was controlled using Biomek (v.2.1), and data were acquired with PatchControl (v.2.1) software. Cells were digested using TrypLE (ThermoFisher) before the experiment (Supplementary Note 4). Illumination was carried out using prototype illumination units (Supplementary Note 5). The intensity of light was specified for each type of measurement.

Data analysis was performed using a proprietary Python library and custom scripts developed in-house. Supplementary notes provide detailed protocols for data collection and analysis, including stationary and non-stationary noise analyses (Supplementary Note 6), a comparison of basic biophysical properties of ChRs (Supplementary Note 7), intensity dependence of photocurrents (Supplementary Note 8) and intra- and extracellular ion composition exchanges for relative ion-permeability determination (Supplementary Note 9).

**Automated spinning-disc confocal microscopy.** To assess the subcellular localization of ChRs in NG108-15 cells, transfected cells were stained with Hoechst 33342 (1:2,000, Invitrogen, H3570) and CellTracker Deep Red (1:1,000, Invitrogen, 2521042). Stained cells were imaged using the CellVoyager CQ1 microscope (Yokogawa) (Supplementary Note 10). A custom Python algorithm was used to automatically analyse the obtained data, with a manual verification step (Supplementary Note 11). The algorithm was used to calculate fluorescence line profiles and corrected partial cell fluorescence per area, serving as measures of subcellular protein distribution and total protein abundance, respectively.

**Ex vivo and in vivo experiments—general**
**rAAV production for ex vivo and in vivo experiments.** Viral vector purification procedure was performed as previously published[13], and descriptions are available in ref. 53 and Supplementary Note 12. The final constructs used for gene therapy in rodents consisted of ChRmine (mice) or ChReef (mice, gerbils and marmosets) under control of the human synapsin (neuronal targeting) or CAG promoter (cardiac cluster stimulation) tagged with eYFP with enhancement of the trafficking signal and export signal packed into PHP.eB for vision restoration experiments, for stimulation of the auditory pathway AAV2/9 for mice and/or PHP.S for gerbils and PHP.eB for marmosets, and in AAV2/9 for cardiac cell cluster stimulation.

**Animals.** All animal experiments were carried out in compliance with the relevant national and international guidelines (European Guideline for animal experiments 2010/63/EU) as well as in accordance with German laws governing animal use. The procedures have been approved by the responsible regional government office. Rodents were kept in a 12 h light/dark cycle with ad libitum access to food and water.

**Optogenetically evoked contraction of cardiomyocyte clusters.** Experiments with beating cardiomyocyte clusters were performed between days 3 and 5 after plating for the neonatal cardiomyocytes (Supplementary Note 13) and after 3–7 days for the hiPSC-derived cardiomyocytes (Supplementary Note 14) cluster with continuous perfusion with Tyrode solution (comprising 140 mM NaCl, 5.4 mM KCl, 1.8 mM CaCl$_2$, 2 mM MgCl$_2$, 10 mM glucose and 10 mM HEPES; pH 7.4, adjusted with NaOH) to keep the cells at approximately 35 °C. Electrical stimulation (0.2 ms long biphasic pulses, 45 V for neonatal cardiomyocyte and 25 V for hiPSC-derived cardiomyocyte clusters) was performed with two platinum electrodes and the Myostim stimulator (Myotronic) and light stimulation with a LedHub (Omicron). Then, 510 nm light was generated by a 500–600 nm LED, filtered through a ET500/20 excitation filter and reflected onto the cells via a 561 nm beamsplitter (AHF, F38-561). Then, 630 nm light was generated with the 625 nm LED through a 635/18 excitation filter and reflected onto the cells via a 409/493/573/652 beamsplitter (AHF, F68-409) within an inverted IX73 microscope and through a ×20 objective (LUCPLFLN20XPH/0,45). Imaging light was filtered by a 780 nm long pass filter (Schott RG780) to avoid excitation of ChRmine and ChReef by the illumination, and images were taken with a UI-306xCP-M camera (iDS) and analysed online with the custom-made Myocyte online Contraction Analysis software recently described in detail[54] giving the absolute average of motion vectors per frame |V| as output through NI 9263 CompactDAQ (NI) recorded with Powerlab 8/35 and LabChart 8.1.16 software for analysis. The software and Powerlab were also used to trigger electrical pacing and LEDs. Light intensity was calibrated with the S170C sensor and Powermeter (Thorlabs). To determine the threshold of optical pacing, cardiomyocyte clusters were illuminated by ten consecutive light pulses (5 ms for 510 nm; 50 ms for 630 nm in neonatal cardiomyocytes and 5 ms for 630 nm in hiPSC-derived cardiomyocyte clusters) with the indicated repetition rates. Light intensity was lowered stepwise to the light intensity at which pacing capability was lost. The lowest intensity capable to trigger contractions 1:1 with the last 8 of 10 light pulses was defined as pacing threshold. Excitation block was tested during supramaximal electrical pacing (0.2 ms long biphasic pulses, 45 V for neonatal cardiomyocytes and 25 V for hiPSC-derived cardiomyocytes) at a repetition rate 0.5 Hz above the spontaneous beating frequency or with 1 Hz if no spontaneous beating was observed. Then, 5 s long light pulses were used and excitation block defined as successful, when electrical stimulation failed to induce any detectable contraction during the illumination period. The histology for neonatal heart cells as well as hiPSC-derived cardiomyocytes is described in Supplementary Note 15

**Methods for optogenetic stimulation of the visual system**
**Optical stimulation.** For visual stimulation at different light intensities, a LED (530 nm, Thorlabs, M530L4) in combination with a filter wheel and six different neutral density filters (Thorlabs, FW1AND) was used. Light intensities were measured using a digital optical power and energy meter (Thorlabs, PM100D). Green light was presented to the contralateral eye from a distance of 2 cm while the eye ipsilateral to the recording site was covered. Visual stimulation was timed (200 ms at 1 Hz) using a pre-programmed pulse generator (Doric, OTPG_8), which was synchronized with the electrophysiological recordings. Accordingly, for visual stimulation using a screen, white full-field flashes (200 ms at 1 Hz) were presented using PsychoPy (v.2021.1.0)[55] on an iPad screen (LG LP097QX1) 5 cm from the contralateral eye. A photodiode was used to synchronize the visual stimulation with the electrophysiological recordings.

**Electrophysiological recordings.** Three to four weeks after intravitreal virus injections (Supplementary Note 16), electrophysiological measurements were performed. To enable stable fixation in the recording set-up, a head-post was implanted onto the skull of vision-restored mice at least 5 days before electrophysiological recordings using standard surgical procedures. To record neural activity, 32-channel silicon probes (Cambridge NeuroTech, type H10b) with a 32-channel headstage (Intan Technologies, C3324) were connected via a SPI interface cable

(Intan Technologies, C3216) to a USB interface board (Intan Technologies, C3100). Data were recorded at 20 kHz using RHX 3.0.3. software. Mice were anaesthetized using fentanyl, medetomidine, midazolam (FMM) cocktail and head-fixed in a holding tube. The head of the mouse was aligned in all three axes to the coordinate system of the probe manipulator. Openings in the skull above the primary visual cortex (anterior-posterior: 3.9 mm; medio-lateral: 2.1 mm) were created using a drill, and the probe was lowered into V1 at a speed of 2 µm s$^{-1}$. Data were acquired at different depths (dorsoventral −0.4 mm to −0.8 mm below the brain surface) in both hemispheres during separate recordings and stimulation of the contralateral eye. Electrophysiological recordings were performed 10 min after reaching the desired depth. CM-Dil (Thermo Fisher, CellTracker) was applied to the probe before brain insertion to allow post hoc confirmation of the recording site. Electrophysiological recordings were analysed using kilosort3.0 (ref. [56]), and spike sorting results were manually curated using the phy 2.0 software (https://github.com/cortex-lab/phy) (Supplementary Note 17). To visualize retinal expression patterns, eyes of vision-restored animals after the electrophysiological experiments were collected and fixed in 4% PFA for 1 h at room temperature for subsequent histology (Supplementary Note 18).

**Light-room/dark-room test.** Mice were maintained on a standard 12 h light/dark cycle, in ventilated and humidity-controlled racks, at room temperature (Supplementary Note 19). As there was some variability across mice in the success of intravitreal injections, 3 weeks following intravitreal injections we performed a screen of fluorescence expression in their retina. Mice had to show clear YFP expression in their fundus images when viewed under an epifluorescence microscope (Phoenix Micron IV) to be selected to undergo behavioural testing with the light-room/dark-room test.

The test environment consisted of two custom-made 30 cm × 30 cm × 85 cm rooms built from black corrugated plastic. Both rooms had a lid, which housed four white-light LED arrays (YM E-Bright). The light intensity on the floor of the light-room was 120 µW (measured with Thorlabs S130C, set to 530 nm). There was a small opening between the two rooms to allow a mouse to pass from one room to the other. Experiments were performed 3 weeks after intravitreal injection. As we have found that for blind rd1 mice, a subset of them spent all their time in the room they start the trial in, which could bias results (for instance, if they started in the dark-room, and spent all their time there, it might look like they are sighted), for each trial we started mice in the light-room. Furthermore, to ensure that any preference for a room had to do with the lighting conditions, and not a preference for a specific one of the rooms independent of lighting conditions, we studied room preferences across four trials, switching which room was lit between trials. All trials were performed on the same day. The first trial lasted for 10 min, was considered habituation and was not used for analysis. Trials 2–4 lasted for 15 min each and were analysed and averaged together. We recorded the mouse's position with an overhead camera (Logitech C270 HD) in the light-room. Analysis was performed in Bonsai (v.2.8.0)[57], which was set up to detect the presence or absence of the mouse in the light room.

**Optogenetic stimulation of auditory pathway in rodents**
**Postnatal AAV injection into the cochlea in mice and Mongolian gerbils.** The same injection approach was performed for all animals which later would be subject to the following: auditory brainstem recordings, juxtacellular recordings from single putative SGNs, ex vivo acute slice electrophysiology in mice or recordings from the inferior colliculus in Mongolian gerbils. In brief, under general anaesthesia, the left cochleae of mouse and gerbil pups were exposed, then gently punctured using a quartz capillary pipette and injected with the virus solution. For details of AAV injections, see Supplementary Note 20.

**oABRs in mice.** Data were obtained from 40 adult C57BL/6J wild-type mice of either sex (22 male, 18 female); 78.9 ± 26.3 days after virus injection, mice were subjected to in vivo optical stimulation of the cochleae and recordings (Supplementary Notes 21 and 22). For evaluation in deafness models, four additional kanamycin-deafened (Supplementary Note 23) C57Bl/6J and five B6;129P2-Otoftm1.1Erei mice, all injected with ChReef-YFP, were subjected to the same type of recordings.

Stimulus generation and delivery, the experimental set-up, and data acquisition were as described in Supplementary Note 24. oABRs and acoustically evoked ABRs were recorded by placing needle electrodes behind the pinna, on the vertex and on the back of the anaesthetized mice. The difference in potential between the vertex and mastoid subdermal needles was amplified using a custom-designed amplifier, sampled at a rate of 50 kHz for 20 ms, filtered (300–3,000 Hz) and averaged across 1,000 stimulus presentations. The ABR threshold was defined and determined as the lowest light or sound intensity for which one of the three waves was reliably visible. The latency of a given wave was defined as the time delay between the stimulus onset and the peak of the wave of interest. The amplitude was defined as the difference between the positive peak and the negative trough of a wave of interest. The details of the analysis of mouse ABR data are presented in Supplementary Note 25. After all in vivo recordings, animals were humanely killed, and cochleae were extracted and processed for immunohistochemistry (Supplementary Note 26).

**Juxtacellular recordings from putative SGNs in mice.** Data were obtained from three adult mice in age between 163 and 307 days after AAV injection (Supplementary Note 20). Surgery was performed under general anaesthesia (Supplementary Note 21).

The experimental/surgical procedure for optical stimulation and kanamycin deafening was performed (Supplementary Notes 22 and 23), after which oABRs were recorded and their threshold was determined. The optical fibre was removed, and the animal underwent a tracheotomy procedure and was subsequently intubated. After the animal was positioned in a custom-designed stereotactic head holder, the optical fibre was replaced, the pinnae were removed, the scalp was reflected, and portions of the lateral interparietal and left occipital bone were removed. Next, the cerebellum was partially aspirated to expose the left semi-circular canal. A reference electrode was placed on the contralateral muscles behind the right ear and was constantly maintained in a moist state to ensure optimal contact. Using the left semi-circular canal as a reference point, a micromanipulator (Luigs Neumann SM 10) was used to insert a borosilicate glass capillary into the cochlear meatus (WPI 1B120F-4, length 100 mm, OD 1.5 mm, ID 0.84 mm). The optical stimulus consisted of laser pulses with a duration of 1 ms, a stimulation rate of 20 Hz and radiant flux of 6 mW. Using the micromanipulator's step function, putative optically sensitive SGNs were identified. Action potentials were amplified using an ELC-03XS amplifier (NPI Electronic), filtered (300–20,000 Hz), digitized (National Instruments card PCIe-6323), analysed and prepared for display using custom-written MATLAB (2012a) (The MathWorks) software. After all in vivo recordings, animals were humanely killed, and cochleae were extracted and processed for immunohistochemistry (Supplementary Note 26).

**Data analysis of juxtacellular recordings from optically driven single putative SGNs in mice.** In accordance with previous work[41], neurons were classified as 'putative SGNs' if they met the following criteria: they (1) showed a monophasic and (2) positive waveform, (3) were recorded at a depth exceeding 1,000 µm below the cerebellum surface, and (4) showed a first spike latency of less than 3 ms in response to a 1 ms light pulse presented at 6 mW. We detected the initial spike that followed each stimulus and calculated both its average latency and latency jitter, represented by the standard deviation of the first spike latency. Neurons were categorized as single- or multiple-spike per pulse if they showed ≤1.1 spikes per pulse or >1.1 spikes per pulse, respectively.

In addition, we computed the vector strength and spikes per pulse by examining the discharge rate starting 150 ms into the stimulus train and extending it for the entire 350 ms stimulation train. Neurons were included if they showed >0.005 spikes per pulse and a vector strength >0.3. This analysis characterizes the neuronal discharge during the adapted state of the neuron. For estimating the light intensity threshold, the laser intensity was gradually lowered until there were no more time-locked responses during the adapted state of the neuron. This protocol was only completed for ten recorded neurons. In total, 18 putative SGNs were recorded from 3 animals (mav189820, 4 putative SGNs; mav181066, 10 putative SGNs; mav190008, 4 putative SGNs).

**Ex vivo acute auditory brainstem slice electrophysiology.** At the age of 20–23 days, 2 weeks after AAV injection (Supplementary Note 20), brainstem slices were prepared as previously described[58] (Supplementary Note 31). An EPSC10 USB patch clamp amplifier was used for all recordings and was driven by the Patchmaster Next software (version 1.2, HEKA, Harvard Bioscience) (Supplementary Note 32). The 150-μm-thick slices that were used for electrophysiological recordings were fixed for 30 min in formaldehyde at room temperature after the experiment and were then used for immunohistochemistry (Supplementary Note 29).

**Optically evoked neural responses in the midbrain in Mongolian gerbils.** Multi-channel recordings of neuronal clusters were performed in the tonotopically organized central nucleus of the inferior colliculus of Mongolian gerbils at the age of 82.9 ± 18.1 days while stimulating SGNs optically (Supplementary Note 30) or acoustically as previously described[34]. In brief, a craniotomy was performed on the right hemisphere of the animal's skull to access the inferior colliculus, contralateral to the injected ear. A linear 32-electrode silicon probe (Neuronexus) was positioned above the brain ~2 mm lateral to lambda and then was slowly inserted ~3.3 mm into the brain (measured from the surface of visual cortex) using a micromanipulator. Mapping of recording sites was done using acoustic stimuli (Extended Data Fig. 4), and the probe was further advanced (or retracted) accordingly. Recordings were performed using a Digital Lynx 4S recording system (Neuralynx) along with custom-built hardware, in combination with custom-written and established MATLAB software. After all in vivo recordings, animals were humanely killed, and cochleae were extracted and processed for immunohistochemistry (Supplementary Note 26).

**Data analysis of inferior colliculus recordings.** The recorded signal from the 32 channels was bandpass filtered between 600 and 6,000 Hz and digitized, and spike times were computed as any waveform crossing three times the mean absolute deviation of the signal. A 1 ms refractory period was included to not overestimate the spike count of the signal. Post-stimulus time histograms were generated, time windows extracted and the cumulative distribution index ($d'$) calculated (see Supplementary Note 30 for details concerning analysis and calculation of $d'$). The $d'$ indicates a change/increase in the neuronal firing rate in relation to baseline levels; that is, a $d'$ of 1 indicates an increase in firing rate of 1 s.d. from baseline and distinguishes triggered signals from noise[34]. Based on these computed matrices, iso-contour lines at integer $d'$ values were constructed and minimums specifying thresholds to $d'$ indicated.

To analyse temporal properties, responsive recording channels, that is, channels reaching a $d'$ of 1 when optically stimulating at 1 mW, were analysed further. Spike rate, number of spike per stimulus and vector strength were extracted (see Supplementary Note 31 for temporal analysis details).

**LED-based multichannel cochlear implant calibration.** LED-based implants were assembled as previously described[42] but using green CreeLEDs (531 nm, C527TR2227). Before each experiment,

optical implants were calibrated using custom-made software built with MATLAB (v.2012a) and an integrating sphere (2P3, Thorlabs). Each LED is calibrated individually using pulsed stimulation with 1 ms pulses from 0% to 100% of maximum 10 mA current provided by a current source generated with a custom-made system based on NI-DAQ-Cards (NI PCI-6229; National Instruments).

**LED-based multichannel cochlear implant stimulation.** Upon careful insertion through the round window until the most proximal LED (to adapter), LEDs were individually addressed using a custom-made system based on NI-DAQ-Cards (NI PCI-6229; National Instruments). To screen for optical intensity thresholds, each LED was individually stimulated from 0 to 2 mA current at 0.1 mA steps and from 2 to 10 mA current at 1 mA steps. Following the first set of recordings, the implant was left in place, and cochlea were deafened with kanamycin (Supplementary Note 23), then the experimental protocols were repeated.

**Histological analysis of mice and Mongolian gerbil cochleae.** Directly after the end of the optogenetic measurements, cochleae from both sides were prepared as previously described[13,19,39] and in Supplementary Note 26 for immunohistochemical imaging.

Image analysis was performed by a custom-written MATLAB script modified from ref. 53. In brief, SGN somas and modiolus area were manually detected using a touch screen from the parvalbumin images. Next, individual somas were automatically segmented using Otsu's threshold method from every Z-stack, and a mask corresponding to the given SGN was defined for the Z-stack for which the mask was fulfilling the criteria of size (area and diameter) and circularity. In case the segmentation was not correct, the segmentation of the given SGN soma was performed manually. Next, the median GFP brightness of each SGN was measured, and its distribution was fitted with a Gaussian mixture model with up to three components. A threshold, above which SGNs somas were considered as transduced, was defined as mean + 3× s.d. of the Gaussian distribution with the lowest mean.

To assess the subcellular expression profiles of ChRmine and ChReef in SGNs, we used a custom-written Python program to plot fluorescence line profiles, incorporating a manual verification process (Supplementary Note 32). The analysis was performed similarly to that used for NG108-15 cells, but centroids were manually selected using a custom MATLAB script, as described in the Methods for determination of GFP-positive cell density in cryosections. Analysis of brain slides is further described in Supplementary Note 33.

**Optogenetic stimulation of the auditory pathway in the common marmoset**
**Auditory brainstem recordings.** Hearing status of animals (Supplementary Note 34) was assessed with a custom-built mobile audiometry set-up twice throughout the experiment (Supplementary Note 35): first, several weeks before virus injection and, second, after a recovery period of 2 to 4 weeks after virus injection (Supplementary Note 36). Eleven (±3 s.d.) weeks after virus injection, animals were subjected to in vivo optical stimulation and recordings. Here, the general surgical and anaesthesia procedure was the same as for the virus application. After again a retroauricular incision, to avoid excessive bleeding due to scar tissue development, the surgical approach to the cochlea was adjusted and included removal of the posterior bony part of the ear canal. Next, needle electrodes were placed behind the pinna, on the vertex and on the neck, and the difference in potential between the vertex and mastoid subdermal needles was amplified and treated as described for the acoustically evoked ABR measurements. A 200 μm optical fibre coupled to a 525 nm (Oxxius) laser was used for optical stimulation. Laser power was calibrated before each experiment using a laser power meter (Gentec EO-Solo, or S140C Thorlabs). After the cessation of optical stimulation experiments, animals were euthanized under deep anaesthesia via an injection of an overdose of pentobarbital

 

and transcardially perfused with 4% formaldehyde with the descending aorta clamped. After euthanasia, temporal bones were postfixated in 4% formaldehyde for 48 h for immunostaining and confocal imaging of cochlear cryosections (Supplementary Note 37).

**Data analysis.** During the injection, oABR measurements and histological analysis, experimenters were blinded to the experimental group. Unblinding was performed afterwards.

### Reporting summary

Further information on research design is available in the Nature Portfolio Reporting Summary linked to this article.

## Code availability

The code used for analysis is available via Zenodo at https://doi.org/10.5281/zenodo.15210800 (ref. 59).

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

## Acknowledgements

We thank C. Senger-Freitag for help with cloning of the ChR constructs and technical support; D. Gerke for virus production and histology of auditory samples; T. Becker for veterinary work on marmosets; A. Thirumalai for assistance in image analysis; G. Hoch for technical support; P. Räke-Kügler for administrative support; J. Kuhl for figure illustrations; A. Villemain for maintaining mouse colonies and assisting with intravitreal injections; and S. Tong for assistance with the Bonsai pipeline for tracking mice in the light-room. The work was funded by the German Research Foundation through the Cluster of Excellence (EXC2067) Multiscale Bioimaging (E.M., T.B., T. Moser, T. Mager), the Else Kröner Fresenius Foundation and zukunft.niedersachsen, the joint science funding program of the Lower Saxony Ministry of Science and Culture and the Volkswagen Foundation via the Else Kröner Fresenius Center for Optogenetic Therapies (M.J., K.K., E.M., T. Moser, T. Mager, B.J.W.), the European Research Council (Advanced Grant 'OptoHear' to T. Moser under the European Union's Horizon 2020 Research and Innovation program, grant 670759 as well as European Research Council Proof-of-Concept grant OptoWave, Horizon 2022 to T. Moser), the European Innovation Council grant OptoWavePro, Horizon-2023-EIC Transition, to T. Moser, grant 101158920 and the Fraunhofer and Max-Planck cooperation program (NeurOpto grant) supported by pact for research and innovation). Work in the laboratory of T. Moser was further supported by the Ernst Jung Prize for Medicine and by Fondation Pour l'Audition (FPA RD-2020-10) and the Volkswagen-Stiftung from the 'Niedersächsisches Vorab' (ZN3898 and ZN4000). Work in the Brügmann was further supported by the German Centre for Cardiovascular Research, the German Research Foundation (IRTG1826 (200857327, S.M.K.), Priority Program 1926 (315212873), CRC1002 (193793266, project A14), clinician scientist program 413501650 to M.A.H. and project 452139556. A.G.-C. was supported by the Alexander von Humboldt Foundation. V.H. and F.E.M. are fellows of the German Academic Scholarship Foundation (Studienstiftung des Deutschen Volkes). A.V. was supported by a HORIZON TMA MSCA Postdoctoral Fellowship (OPTOCODE, grant 101107675). D.S. was supported by the Swiss National Science Foundation (SNSF) Early Postdoc Mobility number 194957, SNSF PostdocMobility number 211087 and Deutsche Forschungsgemeinschaft Walter-Benjamin Programm (Stelle) number SI 2831/1-1. K.D.R.-G. was supported by Fonds de recherche du Québec (301004) and Consejo Nacional de Ciencia y Tecnología (773971) PhD fellowships and S.T. by a Canada Research Chair, a Canadian Institutes of Health Research Project Grant (398769) and an Early-Career Capacity Building Grant from Brain Canada and the Azrieli Foundation.

## Author contributions

T. Moser, T. Mager, B.J.W., E.M. and T.B. designed the research. ChR construct design and mutagenesis were conducted by T. Mager. In vitro characterization was a collaborative effort by A.A., M.Z., A.G.-C. and I.W. under supervision of T. Mager. Virus design and production was done by K.K. Optogenetic cardiac stimulation experiments were performed by M.A.H., S.M.K., M.D.D., L.C. and T.B. Optogenetic vision restoration research was undertaken by D.S., K.D.R.-G., S.T. and E.M. V.H., T.A., F.E.M., A.V. and B.J.W. performed the optogenetic stimulation of the auditory periphery with supervision of T. Moser. Marmoset work was performed by M.J., A.M., B.J.W., A.H., K.K. and A.R. E.K. and P.R. designed, manufactured and assembled the LED-based oCIs. A.A., V.H., M.Z., A.V., F.E.M., D.S., T.B. and T.A. prepared figures with input from other authors. T. Moser, T. Mager and B.J.W. wrote the paper with contributions from E.M. and T.B. T. Moser, E.M. and T.B. acquired funding. All authors have reviewed and approved the final version of the manuscript for submission.

## Funding

## Competing interests

T. Moser is co-founder of the OptoGenTech Company. T. Moser, M.Z. and T. Mager are authors on a pending patent application related to this work, filed by Universitaetsmedizin Goettingen and OptoGenTech (EPA-22155173). The remaining authors declare no competing interests.

## Additional information

**Extended data** is available for this paper at https://doi.org/10.1038/s41551-025-01461-1.

**Correspondence and requests for materials** should be addressed to Bettina J. Wolf, Thomas Mager or Tobias Moser.

[1]Institute for Auditory Neuroscience and InnerEarLab, University Medical Center Göttingen, Göttingen, Germany. [2]Advanced Optogenes Group, Institute for Auditory Neuroscience, University Medical Center Göttingen, Göttingen, Germany. [3]Cluster of Excellence 'Multiscale Bioimaging: from Molecular Machines to Networks of Excitable Cells' (MBExC), University of Göttingen, Göttingen, Germany. [4]Else Kröner Fresenius Center for Optogenetic Therapies, University Medical Center Göttingen, Göttingen, Germany. [5]Auditory Neuroscience and Synaptic Nanophysiology Group, Max-Planck-Institute for Multidisciplinary Sciences, Göttingen, Germany. [6]Department of Otolaryngology, University Medical Center Göttingen, Göttingen, Germany. [7]Cognitive Hearing in Primates Group, Auditory Neuroscience and Optogenetics Laboratory, German Primate Center, Göttingen, Germany. [8]Brain-Wide Circuits for Behavior Research Group, Max Planck Institute for Biological Intelligence, Planegg, Germany. [9]Dynamics of Excitable Cell Networks Group, Department of Ophthalmology, University Medical Center Göttingen, Göttingen, Germany. [10]Montreal Neurological Institute, McGill University, Montreal, Quebec, Canada. [11]Institute for Cardiovascular Physiology, University Medical Center Göttingen, Göttingen, Germany. [12]Functional Auditory Genomics Group, Auditory Neuroscience and Optogenetics Laboratory, German Primate Center, Göttingen, Germany. [13]Stem Cell Unit, Clinic for Cardiology and Pneumology, University Medical Center Göttingen, Göttingen, Germany. [14]German Centre for Cardiovascular Research (DZHK), Partner site Göttingen, Göttingen, Germany. [15]Translational Neuroinflammation and Automated Microscopy, Fraunhofer Institute for Translational Medicine and Pharmacology ITMP, Göttingen, Germany. [16]Department of Microsystems Engineering (IMTEK), University of Freiburg, Freiburg, Germany. [17]BrainLinks-BrainTools Center, University of Freiburg, Freiburg, Germany. [18]Auditory Neuroscience and Optogenetics Laboratory, German Primate Center, Göttingen, Germany. [19]Auditory Circuits Lab, Institute for Auditory Neuroscience, University Medical Center Göttingen, Göttingen, Germany. [20]These authors contributed equally: Alexey Alekseev, Victoria Hunniford, Maria Zerche, Marcus Jeschke, Fadhel El May, Anna Vavakou, Dominique Siegenthaler. ✉e-mail: bettinajulia.wolf@med.uni-goettingen.de; Thomas.Mager@med.uni-goettingen.de; tmoser@gwdg.de

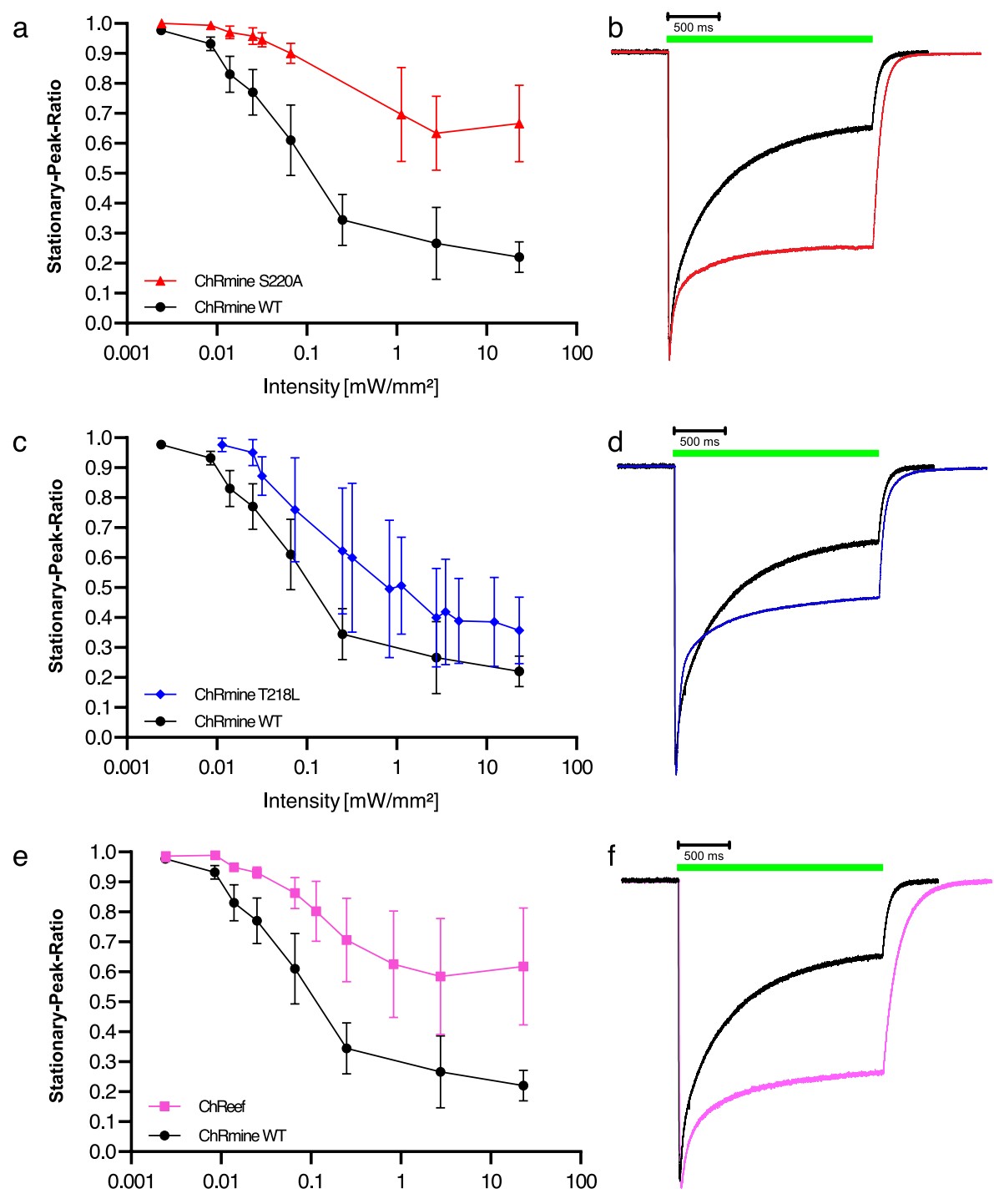

**Extended Data Fig. 1 | Desensitization of ChRmine variants at different light intensities.** NG cells transiently transfected with ChRmine (black circle, WT), ChRmine S220A (red triangle), ChRmine T218L (blue rhombus) and ChReef (magenta squares) were investigated by whole-cell patch-clamp recordings at a membrane potential of −60 mV. Photocurrents were measured upon illumination with a 2 s light pulses of a wavelength of λ = 532 nm at different light intensities ranging from 0.0024 mW/mm2 to 23 mW/mm2. The stationary-peak-ratios were calculated as the quotient of the mean stationary current of the last 100 ms of the 2 s light pulse and the peak current. Shown are the light dependencies of the stationary-peak-ratios for (**a**) ChRmine S220A (n = 4), (**c**) ChRmine T218L (n = 4) and (**e**) ChReef (n = 4) in comparison to ChRmine (n = 5). Exemplary photocurrent traces at saturating light intensity (23 mW/mm2) are depicted for (**b**) ChRmine S220A (red), (**d**) ChRmine T218L (blue) and (**f**) ChReef (magenta) in comparison to ChRmine (black). Error bars show SD, n indicates the number of measured cells.

## HEK293T: green light (530 nm)

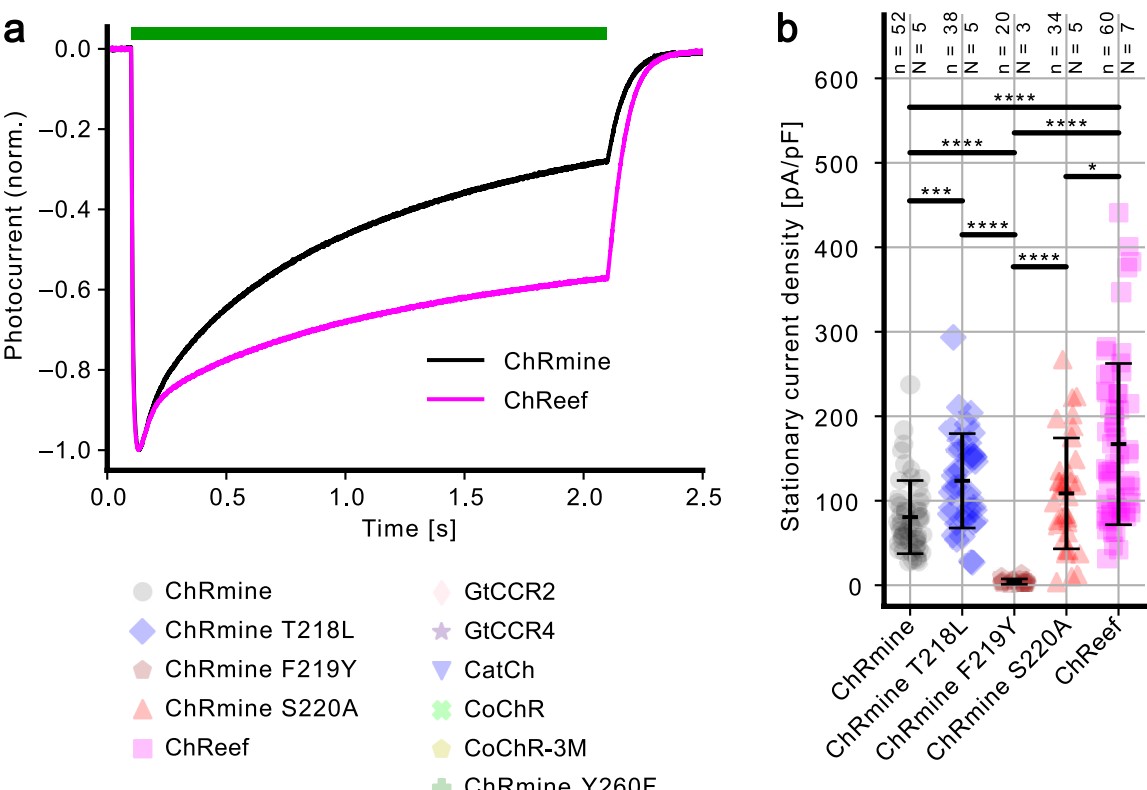

## HEK293T: green (530 nm) or blue (470 nm) light

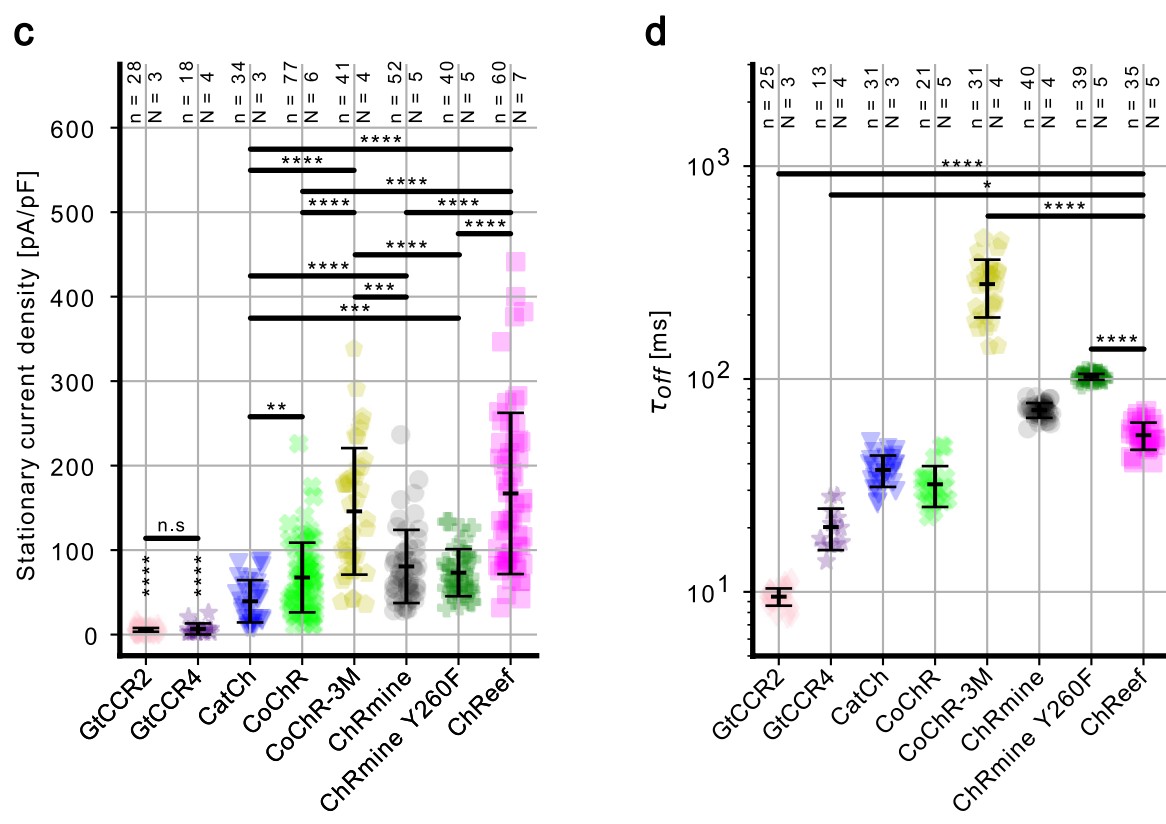

**Extended Data Fig. 2 | See next page for caption.**

**Extended Data Fig. 2 | Comparison of ChR properties based on automated photocurrent measurements in HEK293T cells.** (**a**) Exemplary photocurrents of ChRmine and ChReef expressing HEK cells at a membrane potential of −60 mV in response to 2 s illumination with green light at irradiance of 5.6 mW/mm2. (**b**) Stationary photocurrent densities of ChRmine (black circles), ChRmine T218L (blue rhombi), ChRmine F219Y (brown pentagons), ChRmine S220A (red 'down' triangles), ChReef (ChRmine T218L/S220A, magenta squares). (**c, d**) Stationary photocurrent densities (**c**) and off-kinetics (**d**) of GtCCR2 (pink thin rhombi), GtCCR4 (indigo stars), CatCh (blue 'up' triangles), CoChR (lime crosses), CoChR-3M (yellow pentagons), ChRmine (black circles), ChRmine Y260F (green pluses), ChReef (magenta squares). (**b, c, d**) Numbers at the top represent

number of individual cells (small n) and number of transfections on different days (big N). Horizontal bars indicate mean values, the error bars show SD. The comparisons of the current densities (**b, c**) were done with Kruskal-Wallis test, pairwise comparisons were done with Mann-Whitney U-test with Bonferroni correction for number of comparisons. Four asterisks (****) above GtCCR2 and GtCCR4 in (**c**) indicate significant difference to all other channelrhodopsins in the panel. The comparisons of the off-kinetics values (**d**) were done with Tukey's HSD and only comparisons to ChReef were shown to avoid overloading of the panel. (p-value < 0.05 (*), 0.005 (**), 0.001 (***), 0.0001(****)). Exact p-values for **b-d** are in Supplementary Tables 9–11.

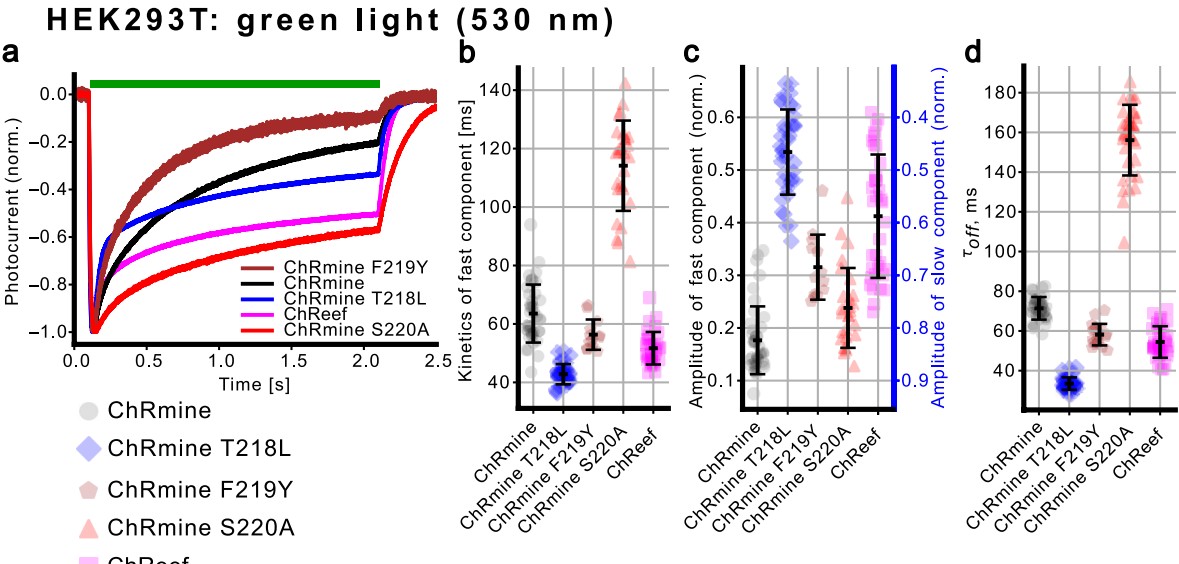

**Extended Data Fig. 3 | Kinetics of photocurrent desensitization at high light intensities.** (**a**) Exemplary photocurrents of ChRmine and its mutants expressing HEK cells at a membrane potential of −60 mV in response to 2 s illumination with green light at irradiance of 5.6 mW/mm2. Legend labels are ordered from more pronounced desensitization to less pronounced. (**b**) Time constant of fast component obtained from double exponential fits of photocurrent decays from peaks to stationary levels (ChRmine: n = 41, ChRmine T218L: n = 38, ChRmine F219Y: n = 13, ChRmine S220A: n = 27, ChReef: n = 35). (**c**) Amplitude of the fast (black axis) and slow (blue axis) components normalized by sum of amplitudes of fast and slow components (fractional contribution of fast and slow components) as obtained from the double exponential fit of photocurrent desensitization process (ChRmine: n = 41, ChRmine T218L: n = 38, ChRmine F219Y: n = 13, ChRmine S220A: n = 27, ChReef: n = 35). (**d**) Off-kinetics obtained in monoexponential fits of photocurrent decays after 5 ms light pulses for ChRmine and its mutants (ChRmine: n = 40, ChRmine T218L: n = 39, ChRmine F219Y: n = 20, ChRmine S220A: n = 35, ChReef: n = 35). n indicates the number of measured cells.

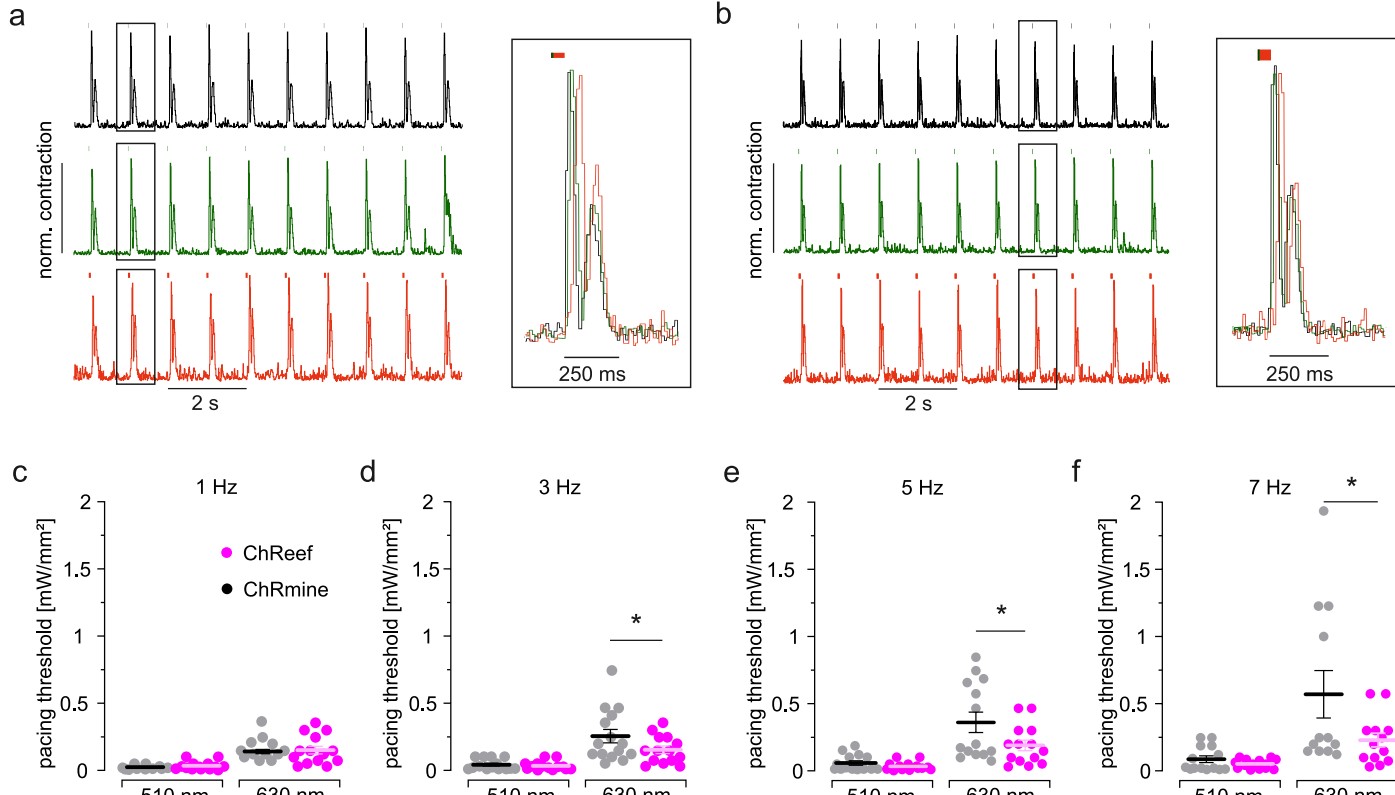

**Extended Data Fig. 4 | Optical pacing of neonatal mouse CM clusters expressing ChRmine and ChReef. a-b**, Representative contractions of a CM clusters expressing ChRmine (**a**) or ChReef (**b**) upon electrical stimulation (top: black lines, 0.2 ms biphasic, 45 V), green light (middle: green lines, 510 nm, 5 ms, 50 μW/mm2) and red light (bottom: red lines, 630 nm, 50 ms, 250 μW/mm2 in a and 150 μW/mm2 in b). **c-f**, Aggregated data of the pacing threshold using green and red light for ChRmine (black) and ChReef (red) for increasing pacing rates: 1 Hz (**c**, p(510 nm) = 0.86; p(625 nm) = 0.90), 3 Hz (**d**, p(510 nm) = 0.98; p(625 nm) = 0.043), 5 Hz (**e**, p(510 nm) = 0.90; p(625 nm) = 0.016) and 7 Hz (**f**, p(510 nm) = 0.95; p(625 nm) = 0.018). Statistical analysis was performed using a 2-way ANOVA with Sidak's multiple comparison test (N = 2 independent isolations and AAV transductions; n ≥ 11 individual CM clusters indicated by single dots).

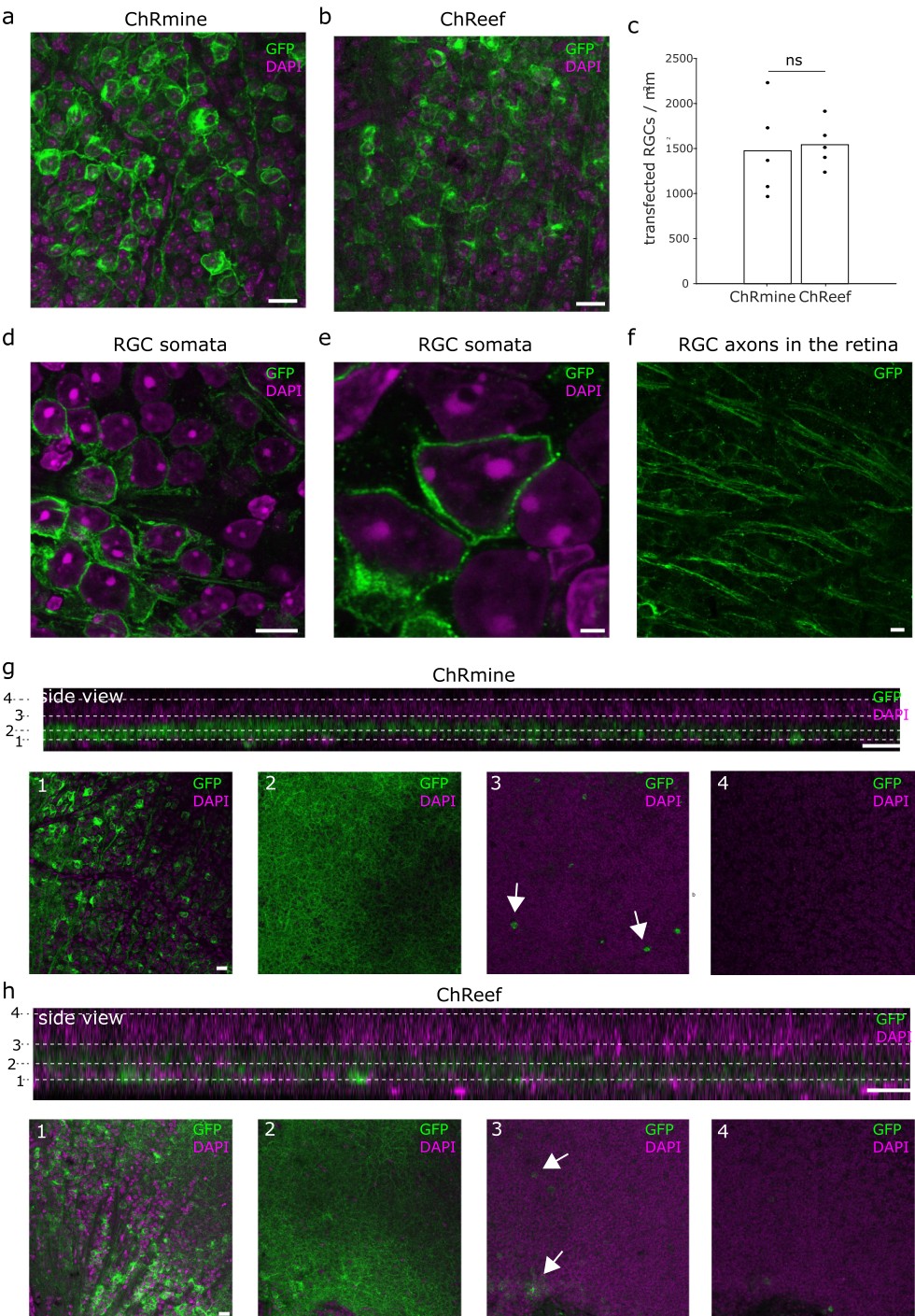

**Extended Data Fig. 5 | Histology of the retina of ChRmine and ChReef injected mice used for vision restoration experiments. a-b** Example images of ChRmine (**a**) and ChReef (**b**) infected retinas at high resolution in the RGC layer. Scale bar represents 20 μm. **c**, Quantification of GFP+ RGCs in both conditions (n = 5 retinas, each point represent average of three images per retina). No significant difference in the number of transfected cells is observed (Mann-U-Whitney test, p = 0.691). **d-e**, RGC somata expressing ChReef-EYFP. **f** ChReef-EYFP expressing RGC axons in the retina projecting towards the optic nerve. Scale bars represent 10 μm (**d**), 2 μm (**e**) and 10 μm (**f**). **g-h**, Upper panels show side view of z-stack trough the retina of ChRmine (**g**) and ChReef (**h**) infected retinas. Lower panels (1–4) show indivdual planes at different depths of the retina indicated in the upper panels (1 RGC layer, 2 inner plexiform layer, 3/4 upper and lower inner nuclear layers). We observe strong expression in the RGCs and inner plexiform layers and sparse expression in cells of the inner and outer nuclear layers (arrows). Scale bars represent 20 μm.

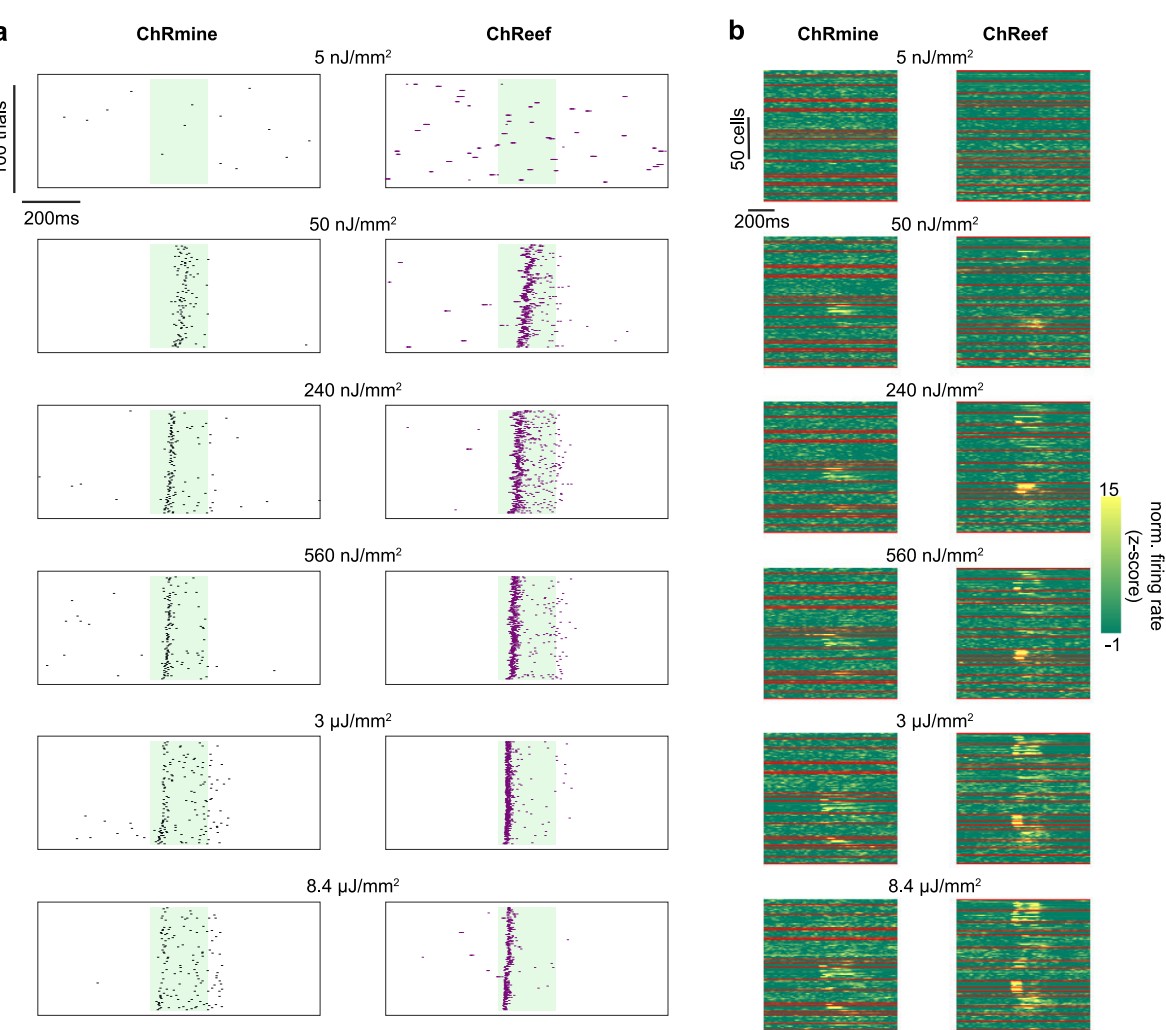

**Extended Data Fig. 6 | ChReef-mediated vision restoration in rd1 mice is reliable with low light requirements. a**, Raster plots of two highly light-sensitive, examplary single units recorded in primary visual cortex of ChRmine- and ChReef-injected rd1 mice in response to various light intensities as indicated.

In such highly sensitive cells, we observed robust responses at light intensities as low as 0.25 μW/mm2 (50 nJ/mm2). **b**, All recorded cells of both conditions and their responses to the same light intensities as in **a**. Red lines separate cells from different recording sites.

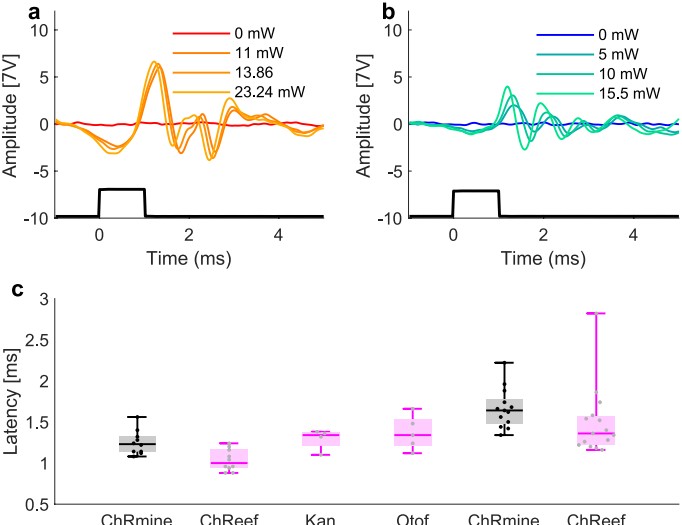

**Extended Data Fig. 7 | Optically evoked auditory brainstem responses in two mouse deafness models. a-b** Optically evoked auditory brainstem responses (oABR) in mice driven with varying radiant flux (1 ms pulses at 10 Hz) for exemplary mice injected with (**a**) ChReef expressing mice after local kanamycin (Kan) application (exemplary trace of N = 4), (**b**) Otof-KO mice expressing ChReef (exemplary trace of N = 5). **c**, Latency of first oABR wave at 10 mW in four experimental groups mouse groups expressing: ChRmine (N = 10), ChReef (N = 10), ChReef after local kanamycin application (Kan, N = 4), ChReef in Otof-KO mice (Otof, N = 5) and with an orange laser (594 nm) for ChRmine (N = 12) and ChReef expressing (N = 15) mice.

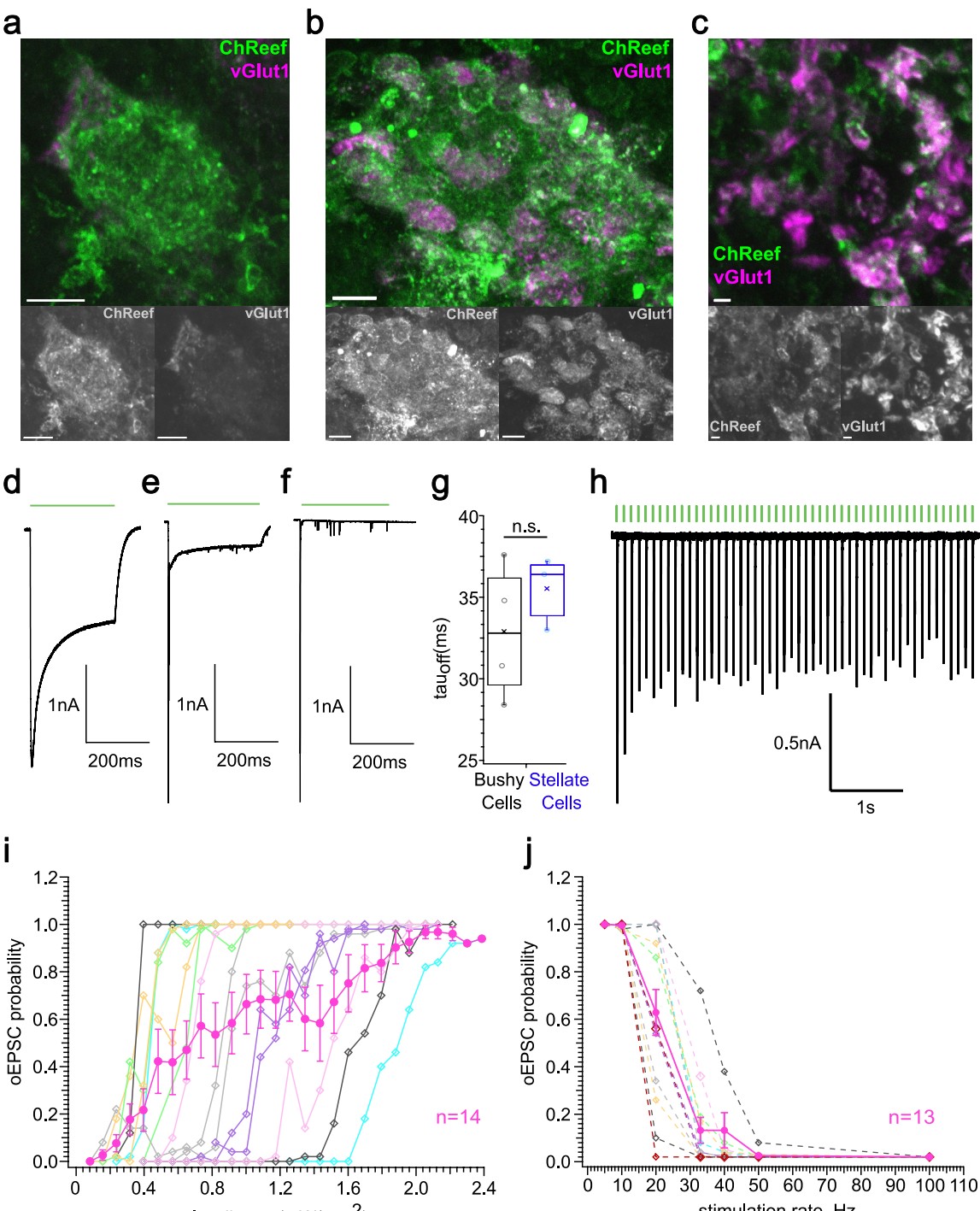

**Extended Data Fig. 8 | Utility of ChReef in mouse brainstem slice physiology.**
**a**–**c**, Parasagittal slices the anteroventral cochlear nucleus (AVCN) were harvested from AAV-injected mice at p20-p23 and fixed for 30 minutes in formaldehyde. The slices were immunolabelled against vGLUT1 and GFP to track the expression of ChReef across the SGN terminals (**b**) and/or transduced postsynaptic neurons, bushy cells (BCs), of the AVCN (**b/c**). **d**–**f**, Voltage clamp recordings from BCs at near-physiological temperature held at −70mV. Exposure to green light triggers ChReef photocurrents in directly transduced BCs (**d**), an overlay of EPSCs and direct currents, when both the endbulb of Held and the BC express ChReef (**e**) and optically evoked EPSCs (oEPSCs), when only the endbulb expresses the channelrhodopsin (**f**). **g**, The closing kinetics of ChReef were measured during voltage clamp recordings from BCs and stellate cells, points represent individual

cells. The exponential decay component's τoff was not significantly different between the two cell types. **h**, Train of 50 oEPSCs resulting from 1 ms light pulses delivered at 10 Hz. **i**, Different laser intensities were tested with 50 stimuli trains delivered at 10 Hz to determine the oEPSC probability as nsuccessful oEPSCs/50. Spike probability is plotted against the tested irradiance values for each voltage clamped BC (multi-colored traces). The average spike probability across all BCs for each irradiance value is depicted as the dark magenta trace and overlaid on the graph. **j**, BCs were exposed to 1 ms light pulses at different frequencies, using the smallest irradiance value that yielded an oEPSC probability of 1 at 10 Hz. The oEPSC probability across different frequencies is depicted for each BC included in the analysis (multi-colored traces). The average oEPSC probability across BCs for each frequency is overlaid as the dark magenta trace. Bars indicate SD.

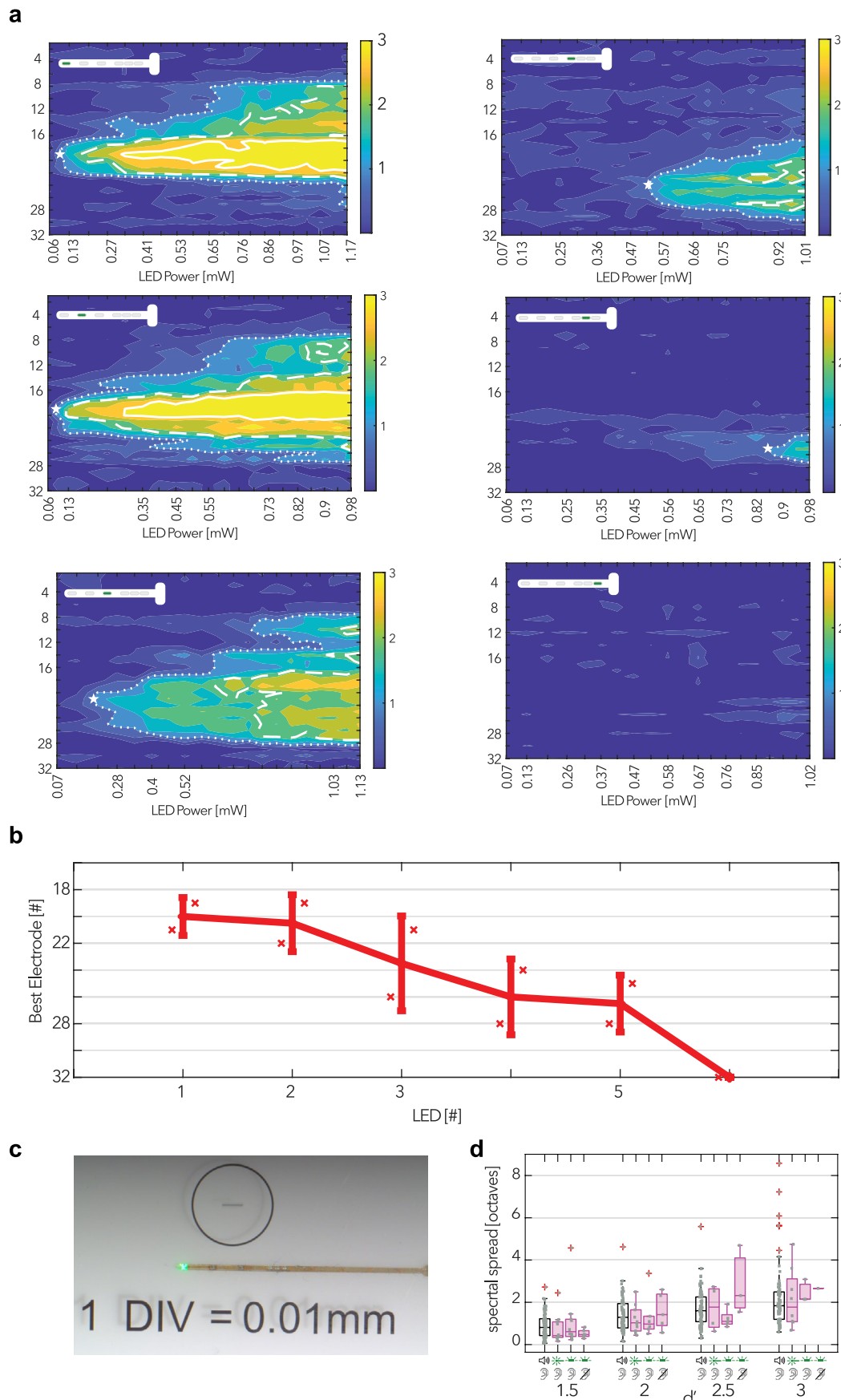

**Extended Data Fig. 9 | See next page for caption.**

**Extended Data Fig. 9 | Characterizing responses in the inferior colliculus among optical stimulation using LED based optical cochlear implants.**
**a**, Exemplary heatmaps of d′ prime computed as indicated in Fig. 5, insert on the top left indicate position of LED used for stimulation (representative of N = 2 ChReef-injected animals). **b**, Best electrode number (BE, 1 to 32 corresponds to most dorsal to most ventral) in response to individual LED used for stimulation. Red crosses indicated individual data points, red bold line indicates mean.

**c**, Example image of LED based optical cochlear implant used for stimulation. One division (DIV) of depicted calibration scale corresponds to 0.01 mm.
**d**, Spectral spread in octaves adjusted from each animal tonotopic slope for different types of stimulation, from left to right: acoustic stimulation (data replotted from Dieter et al. [34])1, stimulation with optical fiber inserted into the round window, stimulation with single LED prior kanamycin application and after kanamycin application. Points represent individual gerbils.

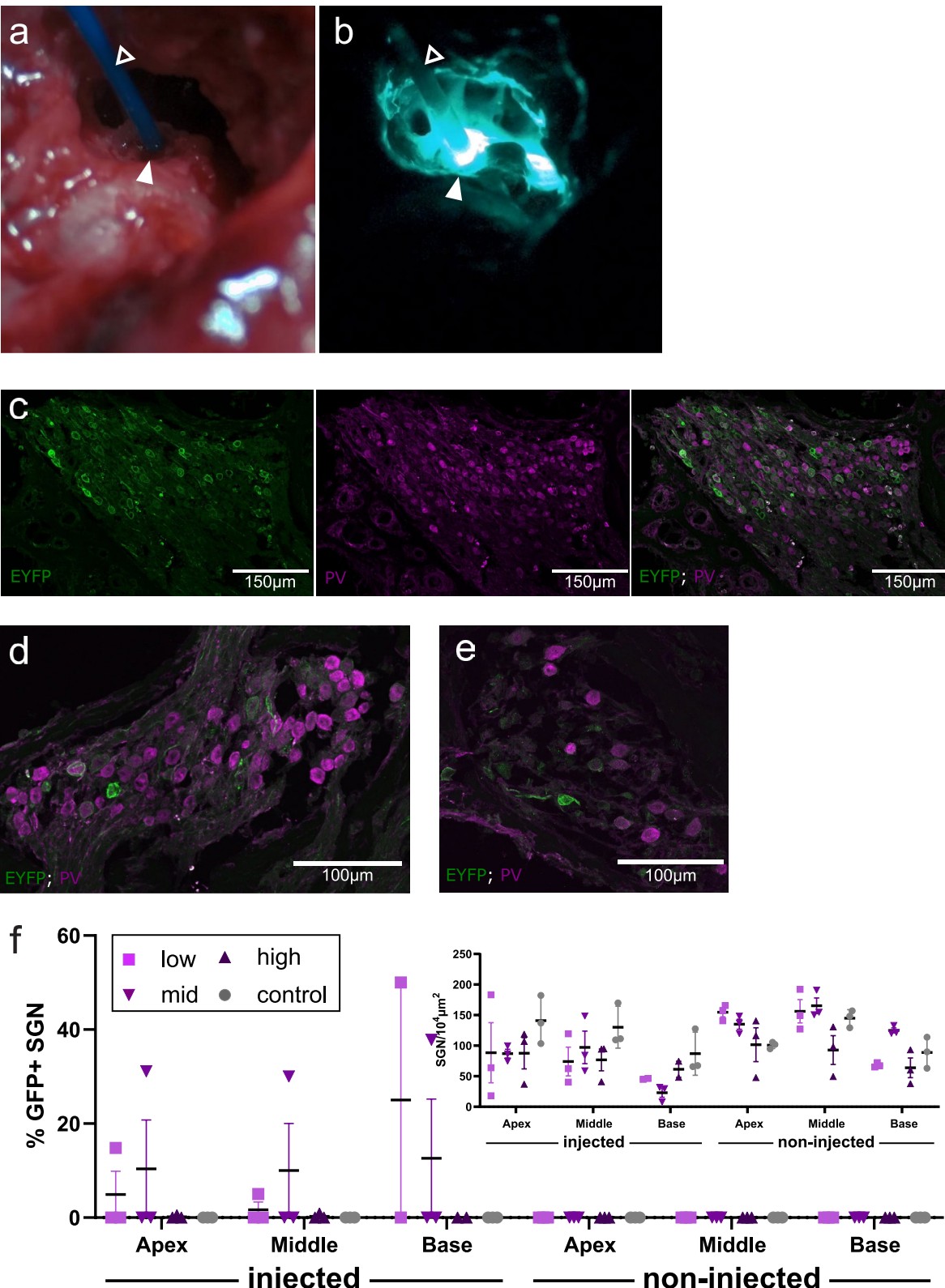

**Extended Data Fig. 10 | Surgical situs during oABR measurements and ChReef expression throughout the different turns in the common marmoset. a**, An optical fiber (open arrowhead) coupled to a 525 nm laser was inserted into the cochlea through the round window (filled arrowhead) which was surgically enlarged slightly. Light pulses of different radiant flux were used to stimulate the cochlea (**b**). **c**, Confocal imaging of ChReef expression in the apical turn of the left cochlea (EYFP channel (green), Parvalbumin channel (PV, red) and composite), same image as Fig. 5. **d-e**, Confocal microscopy images of ChReef expression in mid-turn (**d**, EYFP green, PV red) and basal turn (**e**, EYFP green, PV red) for the oABR positive animal (n = 1/9). **f**, Quantification of the transduction rate of the different turns compared between injected and non-injected side across the different titer and control groups. The inset shows SGN density of the different turns compared between injected and non-injected side.

# Reporting Summary

## Statistics

For all statistical analyses, confirm that the following items are present in the figure legend, table legend, main text, or Methods section.

| n/a | Confirmed | |
|---|---|---|
| ☐ | ☒ | The exact sample size (*n*) for each experimental group/condition, given as a discrete number and unit of measurement |
| ☐ | ☒ | A statement on whether measurements were taken from distinct samples or whether the same sample was measured repeatedly |
| ☐ | ☒ | The statistical test(s) used AND whether they are one- or two-sided<br>*Only common tests should be described solely by name; describe more complex techniques in the Methods section.* |
| ☒ | ☐ | A description of all covariates tested |
| ☐ | ☒ | A description of any assumptions or corrections, such as tests of normality and adjustment for multiple comparisons |
| ☐ | ☒ | A full description of the statistical parameters including central tendency (e.g. means) or other basic estimates (e.g. regression coefficient) AND variation (e.g. standard deviation) or associated estimates of uncertainty (e.g. confidence intervals) |
| ☐ | ☒ | For null hypothesis testing, the test statistic (e.g. $F$, $t$, $r$) with confidence intervals, effect sizes, degrees of freedom and $P$ value noted<br>*Give P values as exact values whenever suitable.* |
| ☒ | ☐ | For Bayesian analysis, information on the choice of priors and Markov chain Monte Carlo settings |
| ☒ | ☐ | For hierarchical and complex designs, identification of the appropriate level for tests and full reporting of outcomes |
| ☒ | ☐ | Estimates of effect sizes (e.g. Cohen's *d*, Pearson's *r*), indicating how they were calculated |

*Our web collection on statistics for biologists contains articles on many of the points above.*

## Software and code

Policy information about availability of computer code

| | |
|---|---|
| Data collection | Clampex 9.2, PatchControl v2.1, Yokogawa CQ1 1.0, Powerlab 8/35, Myocyte online Contraction Analysis (MoCA), RHX, Digital Lynx 4S, MATLAB 2012a |
| Data analysis | python 3.12.2, numpy 1.26.4, pandas 2.2.1, scipy 1.12.0, pingouin 0.5.4, scikit-learn 1.4.1, napari 0.4.19, MATLAB 2021b, Origin 9, Clampfit 10.7, LabChart 8.1.16, PsychoPy, kilosort3, phy, Bonsai, Origin 9, Clampfit 10.7, ImageJ 1.52p, GraphPad Prism 8, ZEN 2.6, custom software is available on Zenodo: https://doi.org/10.5281/zenodo.15210800 |

For manuscripts utilizing custom algorithms or software that are central to the research but not yet described in published literature, software must be made available to editors and reviewers. We strongly encourage code deposition in a community repository (e.g. GitHub). See the Nature Portfolio guidelines for submitting code & software for further information.

## Data

Policy information about availability of data

All manuscripts must include a data availability statement. This statement should provide the following information, where applicable:

- Accession codes, unique identifiers, or web links for publicly available datasets
- A description of any restrictions on data availability
- For clinical datasets or third party data, please ensure that the statement adheres to our policy

The manuscript does include a data availability statement and data is available on Zenodo https://doi.org/10.5281/zenodo.15210800

# Research involving human participants, their data, or biological material

Policy information about studies with human participants or human data. See also policy information about sex, gender (identity/presentation), and sexual orientation and race, ethnicity and racism.

| | |
|---|---|
| Reporting on sex and gender | N/A |
| Reporting on race, ethnicity, or other socially relevant groupings | N/A |
| Population characteristics | N/A |
| Recruitment | N/A |
| Ethics oversight | N/A |

Note that full information on the approval of the study protocol must also be provided in the manuscript.

# Field-specific reporting

Please select the one below that is the best fit for your research. If you are not sure, read the appropriate sections before making your selection.

☒ Life sciences ☐ Behavioural & social sciences ☐ Ecological, evolutionary & environmental sciences

For a reference copy of the document with all sections, see nature.com/documents/nr-reporting-summary-flat.pdf

# Life sciences study design

All studies must disclose on these points even when the disclosure is negative.

| | |
|---|---|
| Sample size | For most experiments no particular statistical method was used to predetermine sample sizes. Sample sizes were choosen on the basis of previously published experimental designs whenever disclosed in the manuscript. For some of the animal studies sample size was calculated using power analysis. |
| Data exclusions | No data were excluded from the analyses. |
| Replication | The number of replicated or independently peformed experiments is provided in the manuscript. Wherever applicable, the number of unsuccessful attempts is clearly stated in the manuscript. |
| Randomization | Randomization was not performed. |
| Blinding | Experimenters were blinded to the experimental group for the optogenetic stimulation of the auditory pathway in the common marmoset. Experimenters were blinded for histological assesment of cochlea tissue. |

# Reporting for specific materials, systems and methods

We require information from authors about some types of materials, experimental systems and methods used in many studies. Here, indicate whether each material, system or method listed is relevant to your study. If you are not sure if a list item applies to your research, read the appropriate section before selecting a response.

## Materials & experimental systems

| n/a | Involved in the study |
|---|---|
| ☐ | ☒ Antibodies |
| ☐ | ☒ Eukaryotic cell lines |
| ☒ | ☐ Palaeontology and archaeology |
| ☐ | ☒ Animals and other organisms |
| ☒ | ☐ Clinical data |
| ☒ | ☐ Dual use research of concern |
| ☒ | ☐ Plants |

## Methods

| n/a | Involved in the study |
|---|---|
| ☒ | ☐ ChIP-seq |
| ☒ | ☐ Flow cytometry |
| ☒ | ☐ MRI-based neuroimaging |

## Antibodies

| | |
|---|---|
| Antibodies used | For cardiomyocytes: cardiac troponin I (ab47003, Abcam: 1:800); Cy5 (711-175-152, JacksonLab, U.S., 1:400); |

| Antibodies used | For retinae: rabbit Proteintech, 50430-2-AP; AlexaFluor 488 (Invitrogen, A32970);<br>For rodemt cochleae: chicken anti-GFP (1:500, ab13970 Abcam, USA); guinea pig anti-parvalbumin (1:300, 195004 Synaptic Systems, Germany); goat anti-chicken 488 IgG (1:200, A-11039 Thermo Fisher Scientific, USA); goat anti-guinea pig 568 IgG (1:200, A-1107 Thermo Fisher Scientific, USA);<br>for rodent brainstem slides: chicken-anti-GFP (1:500, Abcam, Berlin, Germany), guinea-pig-anti-vGLUT1 (1:1000, Synaptic Systems GmbH, Göttingen, Germany); goat-anti-chicken 488 (1:200, Thermo Fisher Scientific, Waltham, USA) goat-anti-guinea-pig 568 (1:200, Thermo Fisher Scientific, Waltham, USA); for gebril brain histology: antibodies for parvalbumin (1:300, guinea pig, Synaptic Systems, Goettingen, Germany) and GFP (1:500, chicken, Abcam, Cambridge, UK);<br>Marmoset histology: chicken anti-GFP (1:500, ab13970 Abcam, USA); guinea pig anti-parvalbumin (1:300, 195004 Synaptic Systems, Germany); mouse anti-NF200 (1:400, Sigma, St. Louis, USA); goat anti-chicken 488 IgG (1:200, Invitrogen Scientific, USA), goat anti-guinea pig 568 IgG (1:200, Invitrogen, USA); anti-mouse 633 (1:200, Invitrogen); guinea pig anti-parvalbumin (1:200; Synaptic systems, Germany), rabbit anti-Otoferlin (1:500; SySy); chicken anti-GFP (1:500; Abcam); anti-chicken 488 (1:1000; Invitrogen), anti-guinea pig 568 (1:1000; Invitrogen); anti-rabbit 633 (1:1000; Thermo Fisher) |
|---|---|
| Validation | Validation was provided by companies. |

# Eukaryotic cell lines

Policy information about cell lines and Sex and Gender in Research

| Cell line source(s) | HEK293T cells (DSMZ, Braunschweig, Germany), NG108-15 (ATCC, HB-12377TM, Manassas, USA), Human cardiomyocyte like cells were differentiated from the induced pluripotent stem cell line UMGi014-A clone 2, which was created from peripheral mononuclear blood cells from a healthy male donor using integration-free Sendai virus. |
|---|---|
| Authentication | The authentication of HEK293T cells and NG108-15 cells were performed by the cell line sources. All cell lines were checked for their morphology and only used in early passage. |
| Mycoplasma contamination | The HEK293T cells and NG108-15 cells were tested negative for mycoplasma contamination. |
| Commonly misidentified lines<br>(See ICLAC register) | No commonly misidentified cell lines were used. |

# Animals and other research organisms

Policy information about studies involving animals; ARRIVE guidelines recommended for reporting animal research, and Sex and Gender in Research

| Laboratory animals | Only laboratory animals were used: Mus musculus (C3HeB/FeJ, JAX 000658; C57B/6J), Mongolian gerbils (Rj:MON), Common marmoset (bred at the German Primate Center) |
|---|---|
| Wild animals | The study did not involve wild animals. |
| Reporting on sex | For all animal studies both sex were used as no impact of the sex was exsepected. |
| Field-collected samples | No field-collected samples were used for this study. |
| Ethics oversight | Electrophysilogy studies in the visual system were approved by the An"Regierung von Oberbayern". Behavior experiments to test for vision restoration were approved by the Neuro Animal Care Committee (ACC) of the McGill University. Animal studies in the auditory system were approved my "`Niedersächsisches Landesamt für Verbraucherschutz und Lebensmittelsicherheit". |

Note that full information on the approval of the study protocol must also be provided in the manuscript.

# Plants

| Seed stocks | *Report on the source of all seed stocks or other plant material used. If applicable, state the seed stock centre and catalogue number. If plant specimens were collected from the field, describe the collection location, date and sampling procedures.* |
|---|---|
| Novel plant genotypes | *Describe the methods by which all novel plant genotypes were produced. This includes those generated by transgenic approaches, gene editing, chemical/radiation-based mutagenesis and hybridization. For transgenic lines, describe the transformation method, the number of independent lines analyzed and the generation upon which experiments were performed. For gene-edited lines, describe the editor used, the endogenous sequence targeted for editing, the targeting guide RNA sequence (if applicable) and how the editor was applied.* |
| Authentication | *Describe any authentication procedures for each seed stock used or novel genotype generated. Describe any experiments used to assess the effect of a mutation and, where applicable, how potential secondary effects (e.g. second site T-DNA insertions, mosiacism, off-target gene editing) were examined.* |

