## [Peer Review File · Nature Biomedical Engineering]

Efficient and sustained optogenetic control of sensory and cardiac systems

Corresponding Author: Prof Tobias Moser

Version 0:

Decision Letter:

Dear Prof Moser,

Thank you again for submitting to *Nature Biomedical Engineering* your manuscript, "Efficient and sustained optogenetic control of nervous and cardiac systems". The manuscript has been seen by 3 experts, whose reports you will find at the end of this message.

You will see that the reviewers appreciate the work. However, they express concerns about the degree of support for the claims, and provide useful suggestions for improvement. We hope that with substantial further work you can address the criticisms and convince the reviewers of the merits of the study. In particular, we would expect that a revised version of the manuscript provides:

- * Extended discussion on the work's rationale and on the development of ChReef, as highlighted by Reviewer #2 and #3.
- * Assessment of the mechanism at the base of the improved photocurrent.
- * Several additional controls as requested by all the reviewers, with emphasis on the cellular localisation of the opsin.
- * Thorough characterization and methodological reporting.

Editorially, the suggested behavioural studies would be welcomed but not mandatory.

When you are ready to resubmit your manuscript, please upload the revised files, a point-by-point rebuttal to the comments from all reviewers, the [reporting summary](https://www.nature.com/authors/policies/ReportingSummary.pdf), and a cover letter that explains the main improvements included in the revision and responds to any points highlighted in this decision.

Please follow the following recommendations:

- * Clearly highlight any amendments to the text and figures to help the reviewers and editors find and understand the changes (yet keep in mind that excessive marking can hinder readability).
- * If you and your co-authors disagree with a criticism, provide the arguments to the reviewer (optionally, indicate the relevant points in the cover letter).
- * If a criticism or suggestion is not addressed, please indicate so in the rebuttal to the reviewer comments and explain the reason(s).
- * Consider including responses to any criticisms raised by more than one reviewer at the beginning of the rebuttal, in a section addressed to all reviewers.
- * The rebuttal should include the reviewer comments in point-by-point format (please note that we provide all reviewers will the reports as they appear at the end of this message).
- * Provide the rebuttal to the reviewer comments and the cover letter as separate files.

We hope that you will be able to resubmit the manuscript within 20 weeks from the receipt of this message. If this is the case, you will be protected against potential scooping. Otherwise, we will be happy to consider a revised manuscript as long as

the significance of the work is not compromised by work published elsewhere or accepted for publication at *Nature Biomedical Engineering*.

We hope that you will find the referee reports helpful when revising the work. Please do not hesitate to contact me should you have any questions.

Best wishes,

Valeria

Dr Valeria Caprettini

Associate Editor, *Nature Biomedical Engineering*

Reviewer #1 (Report for the authors (Required)):

This manuscript presents an original work studying the effects of the channelrhodopsin ChReef, a variant of the ChRmine, in cardiomyocytes for inducing a depolarization block in cardiomyocyte clusters generated from neonatal mouse hearts, rd1 mouse model of retinal dystrophy (responses to optogenetic stimulation of the eye to a green LED or an iPad screen and multielectrode array records from the contralateral primary visual cortex) and auditory system in mice and gerbils (spiral ganglion neurons and cochlear nucleus neurons). The results obtained provide evidence about the potential of the assessed optogenetic tools for modeling in experimental conditions or future clinical therapeutic applications.

The improved light sensitivity and the dynamic of responses clearly represent the main advantages of this technology.

Comments and concerns:

The title "Efficient and sustained optogenetic control of nervous and cardiac systems" could be misleading as the observed effects in the visual and auditory systems still do not cover the entire nervous system.

In the abstract: "Towards clinical application we used AAV-based gene transfer to express ChReef in the optic nerve where it restores visual function in blind mice". The retinal ganglion cells (RGC) were targeted and this should be précised in the abstract, moreover, no data about expression in the optic nerve were shown. Such terminology is surprising. The choice of the promoter should be justified, as well as the cell specificity and the ensuing biodistribution in the tissues studied.

The section "Main or Introduction" could provide more general information about the properties of the bacteriorhodopsin-like cation channelrhodopsin ChRmine that make it suitable as optogenetic therapeutic and a rational basis for engineering of ChRmine variants. A detailed comparison of properties to some recently developed alternative proteins should be provided, e.g., ChR variant, *Chloromonas oogamy* (CoChR) mutants, CoChR-L112C and CoChR-H94E/L112C/K264T, with markedly enhanced light sensitivity.

Despite the use of sophisticated electrophysiological methods and nice illustrations, the manuscript could read more fluently with a more structured presentation of the results. Actually, one should rely mainly on the figures to follow the results and this makes difficult their understanding and interpretation during the reading. Some aspects of the experiments could be presented in more details also in the main text and not only in the methods, some points/questions below:

Has the technology been tested in adult cardiac cells, and in cells types other than myocytes?

Did the rd1 mice treated carry also the Gpr179 mutation ?

Have the morphological and functional parameters (eg OCT, ERG) been evaluated before and after treatment (control and transfected animals)?

Has the stage of tissue remodeling after photoreceptor degeneration been considered?

At what time point after intravitreal injection in vivo was assessed the level of expression (3-4 weeks?)

What cells were transduced besides RGCs? Was a dose ranging in terms of titers tested?

For how long did the expression persist in the transfected cells (transgene stability)?

Has the immune response in the transfected retinas been considered/evaluated?

Is it possible to associate the responses in the visual cortex as a result of direct activation of the RGCs expressing AAVPHP.eB carrying either of the EYFP-tagged ChRs under the control of the human synapsin promoter?

To demonstrate functional vision restoration, this work would benefit of including behavioral studies. Methods such as optokinetic stimulation would be relevant.

Reviewer #2 (Report for the authors (Required)):

The authors described an improved version of the very potent and useful channelrhodopsin ChRmine, which was named ChReef. ChReef was shown to have increased unitary conductance and stationary photocurrent due to decreased desensitization. Furthermore, ChReef has faster dark recovery kinetics and off-kinetics compared to its progenitor. Improved

biophysical properties were successfully translated in cultured cells and in vivo, enabling its successful applications for optogenetic stimulation of the mouse auditory nerve and vision restoration and outperforming ChRmine under identical conditions. If ChReef performs as described, it will be a valuable addition to the channelrhodopsin family and described in vivo demos may prompt its further validation in vivo and application for therapies. However, I do have several comments, which I believe the authors should be able to address easily.

It is known that optogenetic stimulation may have many undesired artifacts on cellular state and function. The classic example is neuronal activity rebound after neuronal silencing using activation of chloride light-driven pumps in neurons or acidification of neurons during extended activation of fast ChR2 variants. The exact mechanism of side effects from optogenetic activation depends on the ionic species conducted by rhodopsins and their cellular localization (axon vs soma vs dendrites). It is also known that mutations that increase photocurrent can largely change the selectivity of ionic conductance (ChRH and CatCh are classic examples). However, the authors did not even try to discuss the potential mechanism of improved photocurrent and the potential alternation of ion contribution to the photocurrent compared to the parental protein. This aspect might be crucial for proper validation of ChReef safety in vivo. It would be great if the authors could address this point.

Based on the histological analysis (resolution and magnification of the provided images are not ideal) ChReef demonstrates some internalization in neurons. This aspect can be crucial since it does not matter what unitary conductance a ChR may have if it is not properly localized to the plasma membrane. I could not find any information on the duration of ChReef expression in vivo. Do the authors observe a higher degree of mislocalization upon longer expression duration? Please comment on the localization of ChReef in vivo and provide high-resolution images of single cells.

Overall, the manuscript reads like a technical report, which is very concise and lacks some rationale, specifically in the first two parts of the Results section. The development of ChReef is not described at all. It is hard to follow the logic of the first and second parts of the Results section. The authors refer to Fig. 1a, which presents single and double mutants ChRmine while not mentioning them in the main text when they call this panel for the first time. Even later in the text, the authors did not explain how they arrived at the ChRmine variants that they reported in the manuscript. The statement "Mutations in Helix 6 accelerate open to closed state transition in green algal ChRs18" is not really helpful for understanding introduced mutations, it was never stated that introduced mutations are in helix 6 of ChRmine. I would recommend revising the manuscript to address the following points: what was the rationale behind introduced mutations? What is the potential mechanism of the increased photocurrent?

The authors presented lots of data on ChReef. However, it is not properly discussed in the manuscript. Please discuss ChReef development, its advantages and limitations compared to its progenitor and other established ChRs, and discuss its performance in vivo and how well it correlates with expected performance based on the measured biophysical parameters in vitro. Provide brief perspectives and outlook for future improvements of ChReef or further characterization/validation.

Minor comments:

The authors presented great results on photocurrent profiles for ChRmine and its mutants in Fig S2; why not further characterize rate of desensitization? It seems to be different among the mutants. It would be great to have the traces in the main text figure with absolute amplitude rather than normalized, as shown in Fig. S2.

Fig S2b,d,f – Y-axis label is missing. If these are normalized values, what was the normalization coefficient, and why should it be normalized?

Do the authors have any comments on the correlation of the results shown in Supp Fig 11f and Fig. 4o-r?

ChReef is not properly introduced in the main text and further inconsistent naming of the ChReef protein in the manuscript makes it difficult to read the manuscript. For example, in Fig. 1 it is referred to as ChReef (even before it was named later in the text) while in the first and second sections of Results it is referred to as ChRmine/T218L/S220A. Why not simply introduce in the last paragraph of introduction and use it throughout since then (all Figures have it as ChReef anyway). Also, in Fig S2, S4, and S5 there is ChREEF (is it the same with ChReef?).

I did not find a reference to Fig S1 in the main text.

"a major driver of progress in the life sciences" – quite an overstatement.

Reviewer #3 (Report for the authors (Required)):

Review nBME-24-0098

In this manuscript the authors present ChReef, an improved variant of the light-gated cation channel ChRmine. The advantage of ChReef over ChRmine is the low desensitization at largely preserved single-channel conductance and overall kinetics. The stationary photocurrent density is increased approx. by a factor of 4 compared to ChRmine and 2.6 compared to CatCh – an opsin that was previously used by the group for optogenetic manipulation of the cochlea. The authors compare the performance of ChReef and ChRmine in several model systems reaching from cultures of cardiomyocytes to the mouse retina in vivo to the cochlea and auditory brain stem in mice and gerbils. Overall, ChReef shows superior performance compared to ChRmine. Due to the red-shifted action spectrum compared to CatCh, it can still be activated with orange-red light, which presents an additional advantage given that long-wavelength light is less scattered and absorbed in biological tissue. Thus, ChReef is potentially useful for a number of in vivo applications, which are demonstrated in this study. However, a number of concerns remain.

Major

1. The choice of the mutation sites to generate ChReef is not well explained and does not become apparent. It is not clear why these particular two amino acid exchanges were chosen to generate ChReef. The two mutations do not seem to be related to any of the helix F mutations in the paper cited for the acceleration of closing kinetics (doi: 10.1038/s41467-018-

04146-3). The sentence “Next, we aimed to overcome ChRmine desensitization by introducing helix 6 mutations to unleash its potential for life science and medical applications.” does not make sense / is not followed by a reasonable motivation. According to how this paragraph is written, it appears as if the authors aimed for a faster version of ChRmine (again, why exactly those 2 point mutations?), but rather discovered a variant with lower desensitization. The strategy and the rationale need to be better explained. Were only these two mutations made? Were other residues targeted? If not, why did the authors stick to exactly those two mutations?

In addition, ChRmine mutants with similarly improved desensitization properties were previously reported (doi: 10.1016/j.cell.2022.01.007). One mutant (Y260F, helix 7 or G) showed similarly low desensitization with preserved photocurrent amplitudes. A comparison of ChReef to this mutant and an explanation of the authors’ strategy is required to assess the advance of the current tool.

2. The authors need to show wavelength-dependency of the desensitization. ChRmine does almost not desensitize under red light (doi: 10.1126/science.aaw5202 and 10.1016/j.cell.2022.01.007). Given this low desensitization and the same spectral response of ChRmine and ChReef (suppl. Fig. 1) it is surprising that ChReef performs so much better than ChRmine under red light in Fig. 2e and f. Without a proper spectral characterization of photocurrents, this effect is difficult to explain.

3. The manuscript contains exaggerated claims in many places. For example:

- The abstract states “...we used AAV-based gene transfer to express ChReef in the optic nerve where it restores visual function in blind mice”. From what I can see, no experiments were done that assess vision. Only responses in V1 were reported.

- Keywords include “hearing, vision, cardiac defibrillation”. However, none of these aspects is shown in the paper. These are clearly not keywords of the present study and mislead the reader.

- The same goes for the running title and the last sentence of the intro. No control of “the heart” is shown.

Such statements are strongly overselling the content of the paper and are not backed up by the data. In this form the manuscript should not be published.

4. No raw traces/example recordings are shown in figure 1, 3, 5. Furthermore, the entire figure 1 is not very well composed and makes a very provisional impression. The plots are hard to read due to the legends being positioned inside the data area. Particularly bad examples are panel c and e. Additionally, the order of the symbols in the legends is inconsistent (compare c and d). In panels 1c-d the values for ChReef and perhaps other opsins are highly skewed and a comparison of the mean values appears inappropriate. Statistics and plots should take into account the non-normality of the data distribution. In particular in the case of ChReef, a few extreme outliers seem to dominate the mean value. The authors need to perform normality tests on all their data and apply correct statistics accordingly.

5. Fig. 2: Why was there an irregular electrical stimulation interval used in panel e? The stimulation rhythm changes within the recording and the stimulation frequency is different between the ChRmine and ChReef conditions. Thus, comparison between the two opsins is limited.

6. No negative controls are shown anywhere in the paper. Especially the “vision restoration” experiments need a control with no opsin expression. Moreover given the use of PHP.eB viruses in these experiments, expression in other brain regions than the retina needs to be excluded. In general, histology is very limited. The example in panel a suggests that expression density was higher for ChReef. Expression needs to be quantified (see also next point).

7. Fig. 4: According to the methods, it seems that the data shown in panels h-r were obtained from two mice – one of them deafened. What is the purpose and why is this not indicated in the figure or the text? It also seems like deafening was not validated anywhere. I find it quite concerning that at those numbers of animals (n = 2) mixed treatments are used – especially in the case where two opsins are compared. Which mouse was ChReef and which one ChRmine injected? These experiments need to be repeated under proper conditions. In addition, histological characterization is not very conclusive. From Suppl. Fig. 9 it looks like ChReef expression was stronger than ChRmine expression, at least in the left cochlea. Can this explain the difference in performance? What is the number of animals compared? It is not indicated in the legend. This aspect needs to be properly evaluated and quantified. If better expression explains the better performance of ChReef, this is fine, but at this stage, cannot be judged properly.

8. I am very skeptical about figure 5. It does not add any additional value to the manuscript (the relevant information is already in figure 4). It shows neither comparison to ChRmine, to any other opsin, or to auditory stimulation. Panels g-h are not even referred to in the main text. Why are they plotted? It is not explained why different animal numbers were used (5c n = 7, ABR recordings in suppl. fig. 14 n = 5). More importantly, the authors used a PHP.S capsid for viral transduction in neonatal animals. Thus, it is highly likely that the virus spread further into the brain and did not remain local in the cochlea. In fact, this is directly seen in suppl. fig. 11. Here, large photocurrents are observed in cells of the ventral cochlear nucleus. Thus direct light effects in the inferior colliculus from non-cochlear sources cannot be disentangled from those responses originating in the cochlea itself. Also the synaptic recordings in suppl. fig. 11 are problematic, because residual photocurrents and local network effects cannot be excluded. According to suppl. fig. 12 it seems that the virus even spread to the contralateral cochlea. These gerbil experiments weaken the manuscript and should be removed.

Minor

9. It is not clear why closing kinetics of 30 ms are reported in the abstract and discussion. The fastest off kinetics reported for ChReef are between 50 and 60 ms (Suppl. Fig. 4 and suppl. tables 1 and 2).

10. Fig. 2: The legend states that 625 nm was used for photostimulation. However, given the optical filters used in this experiment, the light emitted from a 625 nm LED that reached the cells was between 626 and 644 nm. This is relevant with regard to the low desensitization of ChRmine in this wavelength range and therefore requires proper spectral characterization of the photocurrents as stated under main point 2.

11. Fig. 4.: The caption mentions radiant flux in mW, but the legend is given in μJ

12. Suppl. Fig. 4.: why are data points missing? Is this due to absence of currents at the reversal wavelength? If so, why is this not an issue with ChRmine photocurrents?

13. Suppl. fig. 8: why are the red lines in drawn over the data?

14. Suppl. Fig. 11.: "AVCN" nowhere defined

Version 1:

Decision Letter:

Dear Tobias,

Thank you for your revised manuscript, "Efficient and sustained optogenetic control of nervous and cardiac systems", which has been seen by the original reviewers. In their reports, which you will find at the end of this message, you will see that they acknowledge the improvements to the work and raise a few additional technical criticisms that we hope you will be able to address.

As before, when you are ready to resubmit your manuscript, please upload the revised files, a point-by-point rebuttal to the comments from all reviewers, the [reporting summary](https://www.nature.com/authors/policies/ReportingSummary.pdf), and a cover letter that explains the main improvements included in the revision and responds to any points highlighted in this decision.

As a reminder, please follow the following recommendations:

- * Clearly highlight any amendments to the text and figures to help the reviewers and editors find and understand the changes (yet keep in mind that excessive marking can hinder readability).
- * If you and your co-authors disagree with a criticism, provide the arguments to the reviewer (optionally, indicate the relevant points in the cover letter).
- * If a criticism or suggestion is not addressed, please indicate so in the rebuttal to the reviewer comments and explain the reason(s).
- * Consider including responses to any criticisms raised by more than one reviewer at the beginning of the rebuttal, in a section addressed to all reviewers.
- * The rebuttal should include the reviewer comments in point-by-point format (please note that we provide all reviewers will the reports as they appear at the end of this message).
- * Provide the rebuttal to the reviewer comments and the cover letter as separate files.

We expect that you will be able to resubmit the manuscript within 15 weeks of receiving this message. If this is the case, you will be protected against potential scooping. Otherwise, we will be happy to consider a revised manuscript as long as the significance of the work is not compromised by work published elsewhere or accepted for publication at *Nature Biomedical Engineering*.

We look forward to receive a further revised version of the work. Please do not hesitate to contact me should you have any questions.

Best wishes,

Valeria

Dr Valeria Caprettini

Associate Editor, Nature Biomedical Engineering

Reviewer #1 (Report for the authors (Required)):

The Authors performed an extensive revision of the manuscript, properly addressing almost all of the points raised by the reviewers. A substantial amount of new data and analyses has been added and this improved significantly the quality of study results and conclusions. The newly introduced descriptive and experimental material (including the demonstration of functional vision restoration upon the studied optogenetic approach), supplementary figures and references further supported the working hypothesis and scientific importance of the obtained results.

They don't address some of the comments as they state that existing knowledge of the animal models makes it unnecessary to analyze potential changes induced by the therapy on retinal morphology. Inflammatory responses should also be investigated. This will certainly be part of their future work.

José-Alain Sahel

Reviewer #2 (Report for the authors (Required)):

The authors addressed all my concerns and comments in full by providing new experimental results and revising the text. The revised manuscript was substantially improved. The quality and data representation in Figures were also improved. In addition, the authors demonstrated the possibility of optogenetic hearing in common marmosets, which is crucial for the clinical value of ChReef applicability. Unfortunately, the manuscript contains a major issue. In the checklist, the authors checked two main items:

“The exact sample size (n) for each experimental group/condition, given as a discrete number and unit of measurement”
“A statement on whether measurements were taken from distinct samples or whether the same sample was measured repeatedly”

However, the figure legends for Figure 1, Figure 4, Supplementary Figures 1,2,4,6 contain neither a unit of measurement nor a statement on whether measurements were taken from distinct samples or whether the same sample was measured repeatedly”. However, Supplementary Figure 5 contains proper description of both units and repeated measurements.

Furthermore, I have two concerns regarding data and code availability statements:

Data availability statement: The data that support the findings of this study is available from the corresponding authors upon reasonable request.

Code availability statement: The code used for analysis is available from the corresponding authors upon reasonable request.

Why not upload the source data files for Supplementary Figures and the most critical raw dataset to a public repository such as FigShare or Zenodo? This is a great work, but the lack of immediate transparency may complicate the independent reproducibility of the results by other researchers. Do the authors have any concerns regarding sharing the most critical files on public file repositories and providing DOI in the manuscript?

Moreover, I did not see source data files for the figures. I thought providing source files during the review process was mandatory for all Nature journals.

Reviewer #3 (Report for the authors (Required)):

The authors have addressed a number of concerns and added a lot of new analyses to the manuscript. While the manuscript has improved considerably, there are still some remaining questions. It is essential to address these points, given the prominent claims it makes about the breakthrough applications attributed to the new ChReef.

Now that the authors included raw traces in figure 1, it looks like ChReef has overall larger photocurrents, both stationary, but also peak (1e). This notion is also supported by the fact that photocurrent density is almost 5x higher compared to ChRmine, whereas the ratio between stationary and peak photocurrent is improved only 3x (see suppl. Table 1). It is important to address this point and explain this overall improved photocurrent density. Is it due to better membrane trafficking / expression of the opsin? Single channel conductance is ruled out as a candidate mechanism, as it is identical between ChRmine and ChReef (1j). Since this is a biomedical engineering journal, there must be a rigorous explanation and experimental confirmation of the observed effects. As it is now, the community will not get sufficient insight into the actual improvements of the tool itself.

Even though figure 1 has improved, there are still some problems with it. Labels and legends are still inside the plotting area. Especially in panels c and d this is problematic as the horizontal significance lines are inside the range of plotted data points. Another problem is the re-use of the same data in panels b and c (ChRmine and ChReef). Even though the authors indicate that the data are re-plotted, they cannot use them twice for different statistical tests. Either all variants shown in b and c need to be compared against each other in one test, or ChReef and ChRmine data have to be compared to those measurements that were done in the same experimental cohort. The same applies to supplementary tables 1 and 2. It is also

not clear why statistical differences are shown by asterisks in some panels and not in others (e.g. a, d).

Figure 3: The new behavioral analyses should be shown in the main manuscript (and the typos in the figure caption should be fixed). This would add relevance to the vision restoration aspect of the paper.

The authors now show new experiment with a multichannel optical cochlear LED implant emitting green light used for the first time, which is a major improvement compared to blue light emitting ones. This aspect is important, especially since these LEDs match the action spectrum of ChReef and thus, require less light energy (enabling deeper tissue penetration). This aspect is highly relevant and should be presented more prominently in the main paper. Related to this, Supplementary fig. 24d is not readable due to low quality.

Figure 4d now shows a comparison of opsin-expressing cells in the cochlea. The density of ChRmine expressing cells appears to be higher on average (approx. 25 vs. 15 GFP+SGNs/1000 μ m²). In the text, it is stated that "Expression of both ChR variants under the control of the human synapsin promoter was found in approximately 75% of the SGNs". This contradicts the plot in panel d – at least if we assume that the total density of SGNs was the same between animals. Or were more cells lost with ChRmine over time, despite initial similar transduction rate?

Figure 5j: it is not obvious in which of the traces there is an objectively measurable response. It looks like we see one single experiment with single trials at different light intensities. The authors state that "auditory responses were observed from 4.7 mW" (is this the 4th line from the bottom?). According to the methods section, "one of the (...) 2 waves (oABR) was reliably visible." What is meant by "reliable", given that I see only one trial? Where is this wave in the 4.7 mW trial and where is it in the ones with higher irradiance?

Except for one t-test in figure 4 g, I could not see any statistical analyses of all the auditory data presented in the entire manuscript (figures 4 and 5 and supplementary figures 18-26). There are many quantitative plots comparing ChRmine and ChReef and various manipulations or expression levels where it remains unclear what the data mean. For example, suppl. fig 18, 19, 21, fig. 4 c, d, h, i, 5 c all need quantitative statistics. If no significant differences are found, p values need to be shown and the reader needs to know why superiority of ChReef over ChRmine may be less pronounced in the auditory system. Along these lines, it looks like there is only a clear difference between ChRmine and ChReef at oABRs evoked with 594 nm.

All the experiments that follow are showing some convincing improvements towards optogenetic hearing restoration in general. However, it remains unclear to what extent this is due to ChReef being superior to ChRmine. All gerbil and marmoset experiments are done exclusively done with ChReef. This needs to be more clearly pointed out by mentioning that they opted for the best opsin/virus combination to realize applications in the auditory system that failed with other constructs previously and that similar results may be achieved with ChRmine using the improved parameters (PHP.S capsid, early virus transduction, green LEDs, etc.).

It is nowhere mention whether ChReef will be made publicly available. Ideally, the authors should deposit the DNA on a public repository, so that the community can benefit from the new tool in an unrestricted way.

Color schemes are not consistent throughout the manuscript. Sometimes, magenta/green, sometimes red/green lookup tables are used. The latter is not recommended due to discrimination of people with impaired color vision.

Version 2:

Decision Letter:

Dear Prof Moser,

Thank you for your revised manuscript, "Efficient and sustained optogenetic control of sensory and cardiac systems", which has been seen by Reviewers #3. In their reports, which you will find at the end of this message, you will see that the reviewer acknowledge the improvements to the work and raise a few additional technical criticisms that we hope you will be able to address. In particular, we would expect that the next version of the manuscript address in detail factors that helped improve optogenetic control of the auditory system,, and a thorough rewrite of the discussion to highlight the meaning of the results without overselling the proposed tool.

As before, when you are ready to resubmit your manuscript, please upload the revised files, a point-by-point rebuttal to the comments from all reviewers, the [reporting summary](https://www.nature.com/authors/policies/ReportingSummary.pdf), and a cover letter that explains the main improvements included in the revision and responds to any points highlighted in this decision.

As a reminder, please follow the following recommendations:

* Clearly highlight any amendments to the text and figures to help the reviewers and editors find and understand the changes (yet keep in mind that excessive marking can hinder readability).

* If you and your co-authors disagree with a criticism, provide the arguments to the reviewer (optionally, indicate the relevant points in the cover letter).

* If a criticism or suggestion is not addressed, please indicate so in the rebuttal to the reviewer comments and explain the reason(s).

* Consider including responses to any criticisms raised by more than one reviewer at the beginning of the rebuttal, in a section addressed to all reviewers.

* The rebuttal should include the reviewer comments in point-by-point format (please note that we provide all reviewers will the reports as they appear at the end of this message).

* Provide the rebuttal to the reviewer comments and the cover letter as separate files.

We expect that you will be able to resubmit the manuscript within 12 weeks of receiving this message. If this is the case, you will be protected against potential scooping. Otherwise, we will be happy to consider a revised manuscript as long as the significance of the work is not compromised by work published elsewhere or accepted for publication at *Nature Biomedical Engineering*.

We look forward to receive a further revised version of the work. Please do not hesitate to contact me should you have any questions.

Best wishes,

Valeria

Dr Valeria Caprettini

Senior Editor, *Nature Biomedical Engineering*

Reviewer #3 (Report for the authors (Required)):

After this new round of revisions, some issues remain unaddressed. Contrary to what is stated in the response to my points regarding Figure 5j, no changes were made to the figure panel or the legend. Actually, the legend is misleading. "oABR recordings in a common marmoset resulting in successful activation of the auditory pathway." is not correct. Rather, oABR recordings in a common marmoset resulting FROM successful activation of the auditory pathway is correct. It is still not apparent where the visually detected ABR waves occur in the 4.7 mW trial (or which one the 4.7 mW trial even is). Every trial needs to be labeled with the corresponding radiant flux. The color code is useless. The authors should rather color the non-response trials in one color and the trials showing a response in another color. The response itself needs to be labeled by some kind of indicator (arrow head, etc.), so that it becomes evident to a non-expert reader. The first response, which is clearly visible to me only occurs at around 15.1 mW. For plausibility, the response latencies should also be compared to electrically evoked ones (can be literature values).

The authors also did not seriously address my concern regarding other factors that helped improve optogenetic control of the auditory system. In this study, several factors were optimized compared to their previous work (virus capsids, expression/transduction protocols/light sources) and this needs to be made clear to the reader. The authors mention in their response that they now "communicate the purposeful focus on ChReef for the more demanding studies with gerbils and marmosets". First, I cannot see what has changed in the manuscript from the previous version. Second, the marmoset ABR responses required supra-uJ light energies and are therefore not yet compatible with the microscale green LED emitters. As mentioned above, these aspects need to be critically discussed. The discussion section is currently not worth its title and still reads like an advertisement section with some trivial suggestions on how to improve ChReef. As mentioned in the previous two rounds, the manuscript clearly shows a greatly improved optogenetic approach for stimulating multiple systems. However, the discussion also needs to clearly point out the overall improvements achieved with ChRmine itself, new light sources, new virus serotypes, etc. It also needs to take into account the limitations and discrepancies of the study (just one example: much larger difference in steady-state photocurrent densities between ChRmine and ChReef in NG cells (about 5x) compared to HEK cells (about 2x) and only small differences in the auditory system).

Version 3:

Decision Letter:

Dear Prof Moser,

Thank you for your revised manuscript, "Efficient and sustained optogenetic control of sensory and cardiac systems", which has been seen by Reviewer #3. In their reports, which you will find at the end of this message, you will see that the reviewers acknowledge the improvements to the work and raise a few additional technical criticisms that we hope you will be able to address. In particular, we would expect that the next version of the manuscript could comment in particular on the 4.7mW wave, and to fix the range of stimulation.

As before, when you are ready to resubmit your manuscript, please upload the revised files, a point-by-point rebuttal to the comments from all reviewers, the [reporting summary](https://www.nature.com/authors/policies/ReportingSummary.pdf), and a cover letter that explains the main improvements included in the revision and responds to any points highlighted in this decision.

As a reminder, please follow the following recommendations:

- * Clearly highlight any amendments to the text and figures to help the reviewers and editors find and understand the changes (yet keep in mind that excessive marking can hinder readability).
- * If you and your co-authors disagree with a criticism, provide the arguments to the reviewer (optionally, indicate the relevant points in the cover letter).
- * If a criticism or suggestion is not addressed, please indicate so in the rebuttal to the reviewer comments and explain the reason(s).
- * Consider including responses to any criticisms raised by more than one reviewer at the beginning of the rebuttal, in a section addressed to all reviewers.
- * The rebuttal should include the reviewer comments in point-by-point format (please note that we provide all reviewers will the reports as they appear at the end of this message).
- * Provide the rebuttal to the reviewer comments and the cover letter as separate files.

We expect that you will be able to resubmit the manuscript within 6 weeks of receiving this message. If this is the case, you will be protected against potential scooping. Otherwise, we will be happy to consider a revised manuscript as long as the significance of the work is not compromised by work published elsewhere or accepted for publication at *Nature Biomedical Engineering*.

We look forward to receive a further revised version of the work. Please do not hesitate to contact me should you have any questions.

Best wishes,

Valeria

Dr Valeria Caprettini
Senior Editor, <http://www.nature.com/nbme> *Nature Biomedical Engineering*

Reviewer #3 (Report for the authors (Required)):

The authors have now labeled the trials in Figure 5j and indicated the ABR waves. They also compare these waves to aABRs from the literature. Thank you.
However, since there is apparently no objective/quantitative way to detect ABR waves, I refuse to believe that the wave in the 4.7 mW trial represents a real signal. I leave it to the editors to decide how to handle this.
Regardless, the statement (line 489-490) "Together, these data prove the principle of nano-Joule and sustained optogenetic stimulation of the rodent and primate auditory pathway,..." is still not correct. $4.7 \text{ mW} \times 1 \text{ ms} = 4.7 \text{ micro joules}$. This is not in the nano-joule range, as stated in the manuscript.

I have criticized these inaccuracies in all rounds of review. But they are still not completely removed from the paper.

There is also an incorrect figure reference in line 481 (Fig. SX) of the newly added text.

Although still quite compact, the discussion has improved considerably.

As a last remark, my criticism was not about underestimating ChRmine. It was about not considering many other factors that have contributed to the general improvement of optogenetics in the auditory system. It is clear that ChReef is a great tool. However, because other factors also improved (more potent viruses, improved light delivery), some of the performance improvement would have been seen with other opsins, as well. There is no harm in discussing this openly.

Version 4:

Decision Letter:

Dear Tobias,

Thank you for your revised manuscript, "Efficient and sustained optogenetic control of sensory and cardiac systems". Having checked your responses to the remaining criticisms, I am pleased to write that we shall be happy to publish the manuscript in *Nature Biomedical Engineering*.

We will be performing detailed checks on your manuscript, and in due course will send you a checklist detailing our editorial and formatting requirements. You will need to follow these instructions before you upload the final manuscript files.

Best wishes,

Valeria

Dr Valeria Caprettini

Senior Editor, Nature Biomedical Engineering

Version 5:

Decision Letter:

Dear Prof Moser,

I am happy to inform you that your manuscript, "Efficient and sustained optogenetic control of sensory and cardiac systems", has now been accepted for publication in *Nature Biomedical Engineering*.

Over the next few weeks, the figures will be checked for production quality, the text edited to ensure that it conforms to house style, and the manuscript typeset.

Our Articles are published about 40 days after the acceptance date (we recommend that you inform your institutional press office of this timeframe), and you will be notified of the actual publication date a few days in advance. Articles can be published any working day of the week, and are pushed live shortly after 10 am London time.

Publishing agreement. You will be asked to digitally sign a publishing agreement (grant of rights). After the signed publishing agreement has been received, the proofs of the article will be sent to you for review. If you have any queries during the production process, or you cannot meet the requested deadline for returning the proofs, please contact rjsproduction@springernature.com.

Nature Biomedical Engineering is a Transformative Journal. Authors may publish their research with us through the traditional subscription access route, or make their paper immediately open access through payment of an article-processing charge. More information about publication options is available.

You may need to take specific actions to comply with funder and institutional open-access mandates. If the

work described in the accepted manuscript is supported by a funder that requires immediate open access (as outlined, for example, by [Plan S](https://www.springernature.com/gp/open-research/plan-s-compliance)) and your manuscript was originally submitted on or after January 1st 2021, then you should select the gold OA route. Authors selecting subscription publication will need to accept our standard licensing terms (including our [self-archiving policies](https://www.springernature.com/gp/open-research/policies/journal-policies)), and these will supersede any other terms that the author or any third party may assert apply to any version of the manuscript.

Acceptance of your manuscript is conditional on agreement, by all authors, with both our [media embargo](http://www.nature.com/authors/policies/embargo.html) and [confidentiality and pre-publicity](http://www.nature.com/authors/policies/confidentiality.html) policies. In particular, you may arrange your own publicity of the Article (for instance, through your institutional press office), as long as you ensure that journalists strictly adhere to the media embargo.

To assist you in disseminating the work, as soon as the Article is published you will be able to take advantage of the Springer Nature [SharedIt](https://www.springernature.com/gp/researchers/sharedit) initiative to [generate a unique shareable link to the Article](http://authors.springernature.com/share) that will allow anyone (with or without a subscription) to read it. Recipients of the link who are subscribers will also be able to download and print the PDF.

Thank you for having submitted this work to *Nature Biomedical Engineering*.

Best wishes,

Barbara Cheifet
Editor
Nature Biomedical Engineering

Reviewer #1:

This manuscript presents an original work studying the effects of the channelrhodopsin ChReef, a variant of the ChRmine, in cardiomyocytes for inducing a depolarization block in cardiomyocyte clusters generated from neonatal mouse hearts, rd1 mouse model of retinal dystrophy (responses to optogenetic stimulation of the eye to a green LED or an iPad screen and multielectrode array records from the contralateral primary visual cortex) and auditory system in mice and gerbils (spiral ganglion neurons and cochlear nucleus neurons). The results obtained provide evidence about the potential of the assessed optogenetic tools for modeling in experimental conditions or future clinical therapeutic applications. The improved light sensitivity and the dynamic of responses clearly represent the main advantages of this technology.

We would like to thank the reviewer for the appreciation of our work and the comments that helped us to further improve our manuscript. We have addressed all points, performed additional experiments and added a substantial amount of new data and analysis. We detail our responses to the comments below.

Comments and concerns:

The title “Efficient and sustained optogenetic control of nervous and cardiac systems” could be misleading as the observed effects in the visual and auditory systems still do not cover the entire nervous system.

Thanks for the comment, changed to “Efficient and sustained optogenetic control of sensory and cardiac systems”

In the abstract: “Towards clinical application we used AAV-based gene transfer to express ChReef in the optic nerve where it restores visual function in blind mice”. The retinal ganglion cells (RGC) were targeted and this should be précised in the abstract, moreover, no data about expression in the optic nerve were shown. Such terminology is surprising. The choice of the promoter should be justified, as well as the cell specificity and the ensuing biodistribution in the tissues studied.

We changed abstract as requested to “Towards clinical application we used AAV-based gene transfer to express ChReef in retinal ganglion cells where it restores visual function in blind mice with light sources as weak as an iPad screen.”

Moreover, we performed behavioral experiments that show restoration of vision in blind mice by ChReef expression in the retina.

In response to the comment on the promotor choice, we added a section in the results as well as two supplementary figures proving looking on ChReef expression among other cells in the retina as well as in the whole brain (Fig. S14 and S15). Moreover, we also mention the need for further studies including those aiming at more target cell specificity in the discussion section.

The section “Main or Introduction” could provide more general information about the properties of the bacteriorhodopsin-like cation channelrhodopsin ChRmine that make it suitable as optogenetic therapeutic and a rational basis for engineering of ChRmine variants. A detailed comparison of properties to some recently developed alternative proteins should be provided, e.g., ChR variant, Chloromonas oogamy (CoChR) mutants, CoChR-L112C and CoChR-H94E/L112C/K264T, with markedly enhanced light sensitivity.

We are grateful for this comment and now provide more information about the beneficial properties of the ChRmine variants. The low single-channel conductance of depolarizing ChRs limits their utility, as the required high levels of expression and high light doses bear the risk of proteostatic stress and

phototoxicity in the target tissue. Therefore, a key property, which qualifies ChRmine variants, in particular ChReef, for future optogenetic therapies, namely the comparatively high unitary conductance, is now explicitly mentioned in the introduction. In addition, we provide a detailed comparison to CoChR wt and CoChR-H94E/L112C/K264T (CoChR-3M), which generates bigger photocurrents than other CoChR variants (Ganjawala *et al.*, 2019), in the results section. The photocurrent densities of ChReef and CoChR-3M are comparable (Fig S5), which is in accordance with the enhanced light sensitivity of both. Due to the more red-shifted action spectrum and much faster channel closing kinetics, which reduces the risk of phototoxic effects and enables higher frequency photostimulation, we deem ChReef better suited for future optogenetic treatments than CoChR-3M (Fig S5, Fig S7, Ganjawala *et al.*, 2019).

Discussion:

Despite the use of sophisticated electrophysiological methods and nice illustrations, the manuscript could read more fluently with a more structured presentation of the results. Actually, one should rely mainly on the figures to follow the results and this makes difficult their understanding and interpretation during the reading.

We completely overhauled the manuscript and further improved the reading flow.

Some aspects of the experiments could be presented in more details also in the main text and not only in the methods, some points/questions below:

Has the technology been tested in adult cardiac cells, and in cells types other than myocytes?

The technology of optogenetic stimulation has been tested and demonstrated by us and others many times in adult cardiomyocytes and intact hearts (first: Bruegmann *et al.*, Nature methods 2010, first defibrillation: Bruegmann, Boyle *et al.*, J Clin Invest 2016 and many others). Expression and function of ChReef have so far not been tested in adult cardiac cells. In response to the reviewer's comment we have performed additional experiments with cardiomyocytes derived from human induced pluripotent stem cell (hiPSC) and confirmed the data obtained with neonatal mouse cardiomyocytes (Fig. S13). While those cells do not necessarily offer the phenotype of adult cardiomyocytes, they are a relevant model given their human origin. To test ChReef in adult cardiomyocytes, experiments with transgenic animals would be required or an *in vivo* gene transfer both of which we deem to be beyond the scope of the current study. We more clearly conveyed the potential impact of the choice of the cellular models in the MS.

Did the rd1 mice treated carry also the Gpr179 mutation ?

No, this strain was tested negative for the Gpr179 mutation (Change *et al.* 2015, PMC4575902).

Have the morphological and functional parameters (eg OCT, ERG) been evaluated before and after treatment (control and transfected animals)?

No – this is a well described mouse line, frequently used in optogenetic vision restoration experiments¹⁻³.

Has the stage of tissue remodeling after photoreceptor degeneration been considered?

Not in detail: we relied on previous characterization⁴⁻⁶ and consider rd1 mice a standard animal model in the field for vision restoration¹⁻³ and use them in a comparable time window.

At what time point after intravitreal injection in vivo was assessed the level of expression (3-4 weeks?)

4-5 weeks

What cells were transduced besides RGCs?

We performed immunohistochemical analysis of the retinas used for the vision restoration experiments. We observe strong expression in the RGCs and inner plexiform layers and sparse expression in cells of the inner and outer nuclear layers (Supplementary Figure 14).

Was a dose ranging in terms of titers tested?

For the presented electrophysiology data for vision restoration only one titer was tested per opsin:

PHP.eB-hsyn-ChRmine-TS-EYFP-ES (1,89E+13 GC/ml, dPCR)

PHP.eB-hsyn-ChReef-TS-EYFP-ES (8,59E+13 GC/ml, dPCR)

A newly added supplementary figure (Supplementary Figure 14a, b) shows that there was no difference between the transduction rate and the two constructs.

For how long did the expression persist in the transfected cells (transgene stability)?

Intravitreal injections were performed 3-4 weeks before the electrophysiological measurements. We added further histological analysis of retina cells as well as brain tissue of animals used for vision restoration experiments (Fig. S15).

Has the immune response in the transfected retinas been considered/evaluated?

No not at this point. We and the field are under the general impression that the immunoprivilege of the retina^{e.g. 7,8} prevents major immune responses to capsid and transgene, yet careful analysis is warranted in follow up studies.

Is it possible to associate the responses in the visual cortex as a result of direct activation of the RGCs expressing AAVPHP.eB carrying either of the EYFP-tagged ChRs under the control of the human synapsin promoter?

To ensure that recorded responses originate from RGCs stimulation we performed additional tissue analysis. Histology of retina cells as well as brain tissue of animals used for vision restoration experiments (Fig. S14 and S15) proof main expression in RGC cells which is why we believe that recorded signal originate from RGCs.

To demonstrate functional vision restoration, this work would benefit of including behavioral studies. Methods such as optokinetic stimulation would be relevant.

Thanks for the advice. We are happy to provide the requested proof on behavioral level. (Supplementary Figure 16)

Reviewer #2 (Report for the authors (Required)):

The authors described an improved version of the very potent and useful channelrhodopsin ChRmine, which was named ChReef. ChReef was shown to have increased unitary conductance and stationary photocurrent due to decreased desensitization. Furthermore, ChReef has faster dark recovery kinetics and off-kinetics compared to its progenitor. Improved biophysical properties were successfully translated in cultured cells and in vivo, enabling its successful applications for optogenetic stimulation of the mouse auditory nerve and vision restoration and outperforming ChRmine under identical conditions. If ChReef performs as described, it will be a valuable addition to the channelrhodopsin family and described in vivo demos may prompt its further validation in vivo and application for therapies. However, I do have several comments, which I believe the authors should be able to address easily.

We would like to thank the reviewer for the appreciation of our work and the comments that helped us to improve our manuscript. We have addressed all points, performed additional experiments and added a substantial amount of new data and analysis. We detail our responses to the comments below.

It is known that optogenetic stimulation may have many undesired artifacts on cellular state and function. The classic example is neuronal activity rebound after neuronal silencing using activation of chloride light-driven pumps in neurons or acidification of neurons during extended activation of fast ChR2 variants. The exact mechanism of side effects from optogenetic activation depends on the ionic species conducted by rhodopsins and their cellular localization (axon vs soma vs dendrites). It is also known that mutations that increase photocurrent can largely change the selectivity of ionic conductance (ChRH and CatCh are classic examples). However, the authors did not even try to discuss the potential mechanism of improved photocurrent and the potential alternation of ion contribution to the photocurrent compared to the parental protein. This aspect might be crucial for proper validation of ChReef safety in vivo. It would be great if the authors could address this point.

We agree to the reviewer that information regarding the conducted ions is of importance and therefore determined ion permeability ratios from the shift of the reversal potential upon ion replacement using the Goldman–Hodgkin–Katz equation (Results, Fig. S8). The ion permeability ratios were not significantly different in ChReef and ChRmine wt (Fig. S8). ChReef is an unselective cation channel ($P_K/P_{Na} \sim 1$) that exhibits a considerable proton conductance ($P_H/P_{Na} \sim 10^5$) and virtually no Ca^{2+} conductance. In regard to ChReef safety validation, we reason that compared to ChR2 much lower proton and calcium permeability (ChR2: $P_H/P_{Na} \sim 10^6$, $P_{Ca}/P_{Na} \sim 0.15$)⁹ of ChReef reduces a possible risk for adverse effects that may result from Ca^{2+} influx or lowered intracellular pH values, which could occur upon long-term stimulation of the cells.

Based on the histological analysis (resolution and magnification of the provided images are not ideal) ChReef demonstrates some internalization in neurons. This aspect can be crucial since it does not matter what unitary conductance a ChR may have if it is not properly localized to the plasma membrane. I could not find any information on the duration of ChReef expression in vivo. Do the authors observe a higher degree of mislocalization upon longer expression duration? Please comment on the localization of ChReef in vivo and provide high-resolution images of single cells.

We have discussed the points raised by the reviewer.

We added example images of retina ganglion cells clearly showing membrane expression of ChReef (Supplementary Figure: 14d).

Furthermore, we performed line profile analysis in spiral ganglion neurons of mice used for hearing restoration experiments. They clearly proof that ChReef and ChRmine are mainly localized at the cell membrane (Fig. 4c-d).

Overall, the manuscript reads like a technical report, which is very concise and lacks some rationale, specifically in the first two parts of the Results section. The development of ChReef is not described at all. It is hard to follow the logic of the first and second parts of the Results section. The authors refer to Fig. 1a, which presents single and double mutants ChRmine while not mentioning them in the main text when they call this panel for the first time. Even later in the text, the authors did not explain how they arrived at the ChRmine variants that they reported in the manuscript. The statement “Mutations in Helix 6 accelerate open to closed state transition in green algal ChRs¹⁸” is not really helpful for understanding introduced mutations, it was never stated that introduced mutations are in helix 6 of ChRmine. I would recommend revising the manuscript to address the following points: what was the rationale behind introduced mutations? What is the potential mechanism of the increased photocurrent?

We thank the reviewer for the constructive criticism and accordingly provide a more detailed description of the development of ChReef and the potential mechanism in the results part. In our own prior work it was shown that the F219Y mutation in helix 6 and mutations at homologous positions significantly accelerated channel closing in green algal ChRs¹⁰. We investigated the effect of mutations at the homologous position (F219Y) and the adjacent positions (T218L and S220A) to the ChR2 F219Y mutation in the cryptophyte ChR ChRmine. The electrophysiological characterization of the ChRmine mutants showed that the channel closing kinetics of ChRmine F219Y, ChRmine T218L and ChRmine T218L/S220A (ChReef) were similar to the channel closing kinetics of ChRmine wt, whereas ChRmine S220A had slower channel closing kinetics. ChRmine F219Y showed strongly reduced photocurrents. In contrast, the photocurrents of ChRmine T218L, ChRmine S220A and ChReef were considerably bigger than the photocurrent of ChRmine wt. In those mutants, the light dependent inactivation mechanism resembling a substrate inhibition of the partial type was abrogated, desensitization was reduced and, as shown for ChReef, the high-frequency power spectral density component was absent. We therefore hypothesize that the T218L mutation and the S220A mutation accelerate the transition from a low conducting parallel photocycle into the main photocycle, thereby increasing the stationary photocurrent.

The authors presented lots of data on ChReef. However, it is not properly discussed in the manuscript. Please discuss ChReef development, its advantages and limitations compared to its progenitor and other established ChRs, and discuss its performance in vivo and how well it correlates with expected performance based on the measured biophysical parameters in vitro. Provide brief perspectives and outlook for future improvements of ChReef or further characterization/validation.

Done: In response, we have completely overhauled the MS to provide more rationale, guidance and information on the experiments and also expanded the discussion section. The brevity of MS is reinforced by the journal's requirements and we request the editorial team to accommodate the changes implemented on request of the reviewers.

Minor comments:

The authors presented great results on photocurrent profiles for ChRmine and its mutants in Fig S2; why not further characterize rate of desensitization? It seems to be different among the mutants.

We thank the reviewer for the helpful suggestion to further characterize the rate of desensitization, which gave valuable insights that we added to the results part. The photocurrent desensitization kinetics of the ChRmine variants at saturating light intensities could be approximated by a bi-exponential function ($\tau_{DES1} \sim \tau_{off}$, $\tau_{DES2} > 1$ s, Fig. S9). The values of the fast time constants were similar to channel closing kinetics. The relative amplitude factor of the slow time constant, which shows the contribution of a slow process that might reflect the substrate inhibition of the partial type to photocurrent desensitization, was much smaller in ChReef than in ChRmine wt (Fig. S9).

It would be great to have the traces in the main text figure with absolute amplitude rather than normalized, as shown in Fig. S2. Fig S2b,d,f – Y-axis label is missing. If these are normalized values, what was the normalization coefficient, and why should it be normalized?

As suggested by the reviewer, we now show the photocurrent traces in the main text figure (Fig. 1) with absolute amplitude rather than normalized. Fig. S2 b, d, f shows peak normalized values. The figure caption is corrected accordingly. Normalization enables a better visual assessment of the extent of desensitization, the desensitization kinetics and the channel closing kinetics.

Do the authors have any comments on the correlation of the results shown in Supp Fig 11f and Fig. 4o-r?

We increased the number of measurements as well as the analysis for the juxtacellular recordings which now lines up nicely with the measurements in the brain slice physiology.

ChReef is not properly introduced in the main text and further inconsistent naming of the ChReef protein in the manuscript makes it difficult to read the manuscript. For example, in Fig. 1 it is referred to as ChReef (even before it was named later in the text) while in the first and second sections of Results it is referred to as ChRmine/T218L/S220A. Why not simply introduce in the last paragraph of introduction and use it throughout since then (all Figures have it as ChReef anyway).

We thank the reviewer. In the revised manuscript ChReef is as suggested, introduced in the last paragraph of the introduction.

Also, in Fig S2, S4, and S5 there is ChREEF (is it the same with ChReef?).

We corrected the figures accordingly.

I did not find a reference to Fig S1 in the main text.

We now give a reference to Fig S1 (now Fig. S7) in the main text.

“a major driver of progress in the life sciences” – quite an overstatement.

We toned down the statement: “With an ever-growing toolkit of light-sensitive proteins (opsins) and optical devices, optogenetics has driven progress in the life sciences¹¹.”

Reviewer #3 (Report for the authors (Required)):

Review nBME-24-0098

In this manuscript the authors present ChReef, an improved variant of the light-gated cation channel ChRmine. The advantage of ChReef over ChRmine is the low desensitization at largely preserved single-channel conductance and overall kinetics. The stationary photocurrent density is increased approx. by a factor of 4 compared to ChRmine and 2.6 compared to CatCh – an opsin that was previously used by the group for optogenetic manipulation of the cochlea. The authors compare the performance of ChReef and ChRmine in several model systems reaching from cultures of cardiomyocytes to the mouse retina in vivo to the cochlea and auditory brain stem in mice and gerbils. Overall, ChReef shows superior performance compared to ChRmine. Due to the red-shifted action spectrum compared to CatCh, it can still be activated with orange-red light, which presents an additional advantage given that long-wavelength light is less scattered and absorbed in biological tissue. Thus, ChReef is potentially useful for a number of in vivo applications, which are demonstrated in this study. However, a number of concerns remain.

We would like to thank the reviewer for the appreciation of our work and the comments that helped us to improve our manuscript. We have addressed all points, performed additional experiments, and added an substantial amount of new data and analysis. We detail our responses to the comments below.

Major

1. The choice of the mutation sites to generate ChReef is not well explained and does not become apparent. It is not clear why these particular two amino acid exchanges were chosen to generate ChReef. The two mutations do not seem to be related to any of the helix F mutations in the paper cited for the acceleration of closing kinetics (doi: 10.1038/s41467-018-04146-3). The sentence “Next, we aimed to overcome ChRmine desensitization by introducing helix 6 mutations to unleash its potential for life science and medical applications.” does not make sense / is not followed by a reasonable motivation. According to how this paragraph is written, it appears as if the authors aimed for a faster version of ChRmine (again, why exactly those 2 point mutations?), but rather discovered a variant with lower desensitization. The strategy and the rationale need to be better explained. Were only these two mutations made? Were other residues targeted? If not, why did the authors stick to exactly those two mutations?

We thank the reviewer for the constructive criticism and accordingly provide a more detailed description of the development of ChReef in the results part. In our own prior work it was shown that the F219Y mutation in helix 6 and mutations at homologous positions significantly accelerated channel closing in green algal ChRs¹⁰. We investigated the effect of mutations at the homologous position (F219Y) and the adjacent positions (T218L and S220A) to the ChR2 F219Y mutation in the cryptophyte ChR ChRmine. The electrophysiological characterization of the ChRmine mutants showed that the channel closing kinetics of ChRmine F219Y, ChRmine T218L and ChRmine T218L/S220A (ChReef) were similar to the channel closing kinetics of ChRmine wt, whereas ChRmine S220A had slower channel closing kinetics. ChRmine F219Y showed strongly reduced photocurrents. In contrast, the photocurrents of ChRmine T218L, ChRmine S220A and ChReef were considerably bigger than the photocurrent of ChRmine wt. In those mutants, the light dependent inactivation mechanism resembling a substrate inhibition of the partial type was abrogated, desensitization was reduced and, as shown for ChReef, the high-frequency power spectral density component was absent. We therefore hypothesize that the T218L mutation and the S220A mutation accelerate the transition from a low

conducting parallel photocycle into the main photocycle, thereby increasing the stationary photocurrent.

2. In addition, ChRmine mutants with similarly improved desensitization properties were previously reported (doi: 10.1016/j.cell.2022.01.007). One mutant (Y260F, helix 7 or G) showed similarly low desensitization with preserved photocurrent amplitudes. A comparison of ChReef to this mutant and an explanation of the authors' strategy is required to assess the advance of the current tool.

We performed additional experiments with ChRmine Y260F for the comparative assessment of its properties. ChRmine Y260F exhibits considerably smaller photocurrents and slower channel closing kinetics than ChReef (Fig S5).

3. The authors need to show wavelength-dependency of the desensitization. ChRmine does almost not desensitize under red light (doi: 10.1126/science.aaw5202 and 10.1016/j.cell.2022.01.007). Given this low desensitization and the same spectral response of ChRmine and ChReef (suppl. Fig. 1) it is surprising that ChReef performs so much better than ChRmine under red light in Fig. 2e and f. Without a proper spectral characterization of photocurrents, this effect is difficult to explain.

We understand the reviewers' concern. In order to address it, we investigated photocurrent desensitization at different light intensities upon stimulation with green ($\lambda \sim 530$ nm), orange ($\lambda \sim 590$ nm) and red light ($\lambda \sim 632$ nm) (Fig. S10). The photocurrent desensitization kinetics of the ChRmine variants at saturating light intensities could be approximated by a bi-exponential function ($\tau_{DES1} \sim \tau_{off}$, $\tau_{DES2} > 1$ s, Fig. S9). The relative amplitude factor of the slow time constant (τ_{DES2}), which shows the contribution of a slow process that might reflect the substrate inhibition of the partial type to photocurrent desensitization, was much smaller in ChReef than in ChRmine wt (Fig. S9). Photocurrent desensitization generally results from the difference in the distribution of open and closed states in the pre-steady-state (peak current at high intensities) and the steady-state (stationary current). Experiments at suboptimal wavelength ($\lambda \sim 632$ nm) and subsaturating intensities (~ 1 mW/mm²) showed that even at conditions at which, due to asynchronous activation, no peak current could be measured, the stationary photocurrent of ChReef remained elevated (Fig. S11), which indicates a favorable open to closed state distribution in the steady-state that may, as proposed, result from the accelerated transition of the low conducting parallel photocycle into the main photocycle.

About data in figure 2, we would like to point out that we have a different interpretation when looking at the traces in the mentioned paper analyzing the currents of ChRmine also with red light stimulation. We rather see no peak current anymore and only steady state currents similar to low light intensities with blue and green light. Independent of the interpretation and explanation of this observation, the steady state current will play the important role and we carefully checked other potential causes such as different expression rate. The reason that the effect of the higher required light intensity is visible only for red light and not blue/green light is the huge current of both variants making pacing possible with very low light intensities which are rarely above detection threshold (in most cases ~ 10 μ W/mm²). Thus, we lose the sensitivity to detect the differences at such low light intensities. This is enabled for red light since currents are <20% of the currents induced by 510 nm. To clarify this, we changed the wording in the text.

4. The manuscript contains exaggerated claims in many places. For example: - The abstract states "...we used AAV-based gene transfer to express ChReef in the optic nerve where it restores visual function in blind mice". From what I can see, no experiments were done that assess vision. Only responses in V1 were reported. - Keywords include "hearing, vision, cardiac defibrillation".

However, none of these aspects is shown in the paper. These are clearly not keywords of the present study and mislead the reader.

We have toned down the statement in abstract and changed keywords. In addition, we have performed behavioral experiments on vision and show now also include immunohistochemical data demonstrating ChReef expression in retinal ganglion cells. Since auditory brainstem responses are a clinically established method to assess hearing, we have refrained from changing the key word “hearing”. Furthermore, we now included two different mouse models of deafness (Fig. 4f) and performed the newly added measurement in gerbils and non-human primates in deafened individuals.

- The same goes for the running title and the last sentence of the intro. No control of “the heart” is shown. Such statements are strongly overselling the content of the paper and are not backed up by the data. In this form the manuscript should not be published.

Thanks for the comment, we have changed the title to “Efficient optogenetic control of sensory pathways and cardiomyocytes” as well as some other sentences.

5. No raw traces/example recordings are shown in figure 1, 3, 5. Furthermore, the entire figure 1 is not very well composed and makes a very provisional impression. The plots are hard to read due to the legends being positioned inside the data area. Particularly bad examples are panel c and e. Additionally, the order of the symbols in the legends is inconsistent (compare c and d). In panels 1c-d the values for ChReef and perhaps other opsins are highly skewed and a comparison of the mean values appears inappropriate. Statistics and plots should take into account the non-normality of the data distribution. In particular in the case of ChReef, a few extreme outliers seem to dominate the mean value. The authors need to perform normality tests on all their data and apply correct statistics accordingly.

We thank the reviewer for the helpful comment. The revised figures 1 now includes raw traces. We revised Figure 1 according to the reviewers suggestions. In order to take into account the non-normality of the data distribution we employed Kruskal-Wallis with a Dunn’s Bonferroni post hoc test for statistical comparisons.

6. Fig. 2: Why was there an irregular electrical stimulation interval used in panel e? The stimulation rhythm changes within the recording and the stimulation frequency is different between the ChRmine and ChReef conditions. Thus, comparison between the two opsins is limited.

a) We thank the reviewer for pointing out the different pacing rates in the example. We stated in the methods that pacing rate was set to be 0.5 Hz above the spontaneous beating rate but did not further mention this in the legends. To clarify this now, we show you the actual pacing rates of all examples below including the information how many of them have been successful:

(first number = successful depolarization block; second number = total number of tested cardiac clusters):

	0.5 Hz	1 Hz	1.5 Hz	2 Hz	2.5 Hz
ChRmine		3/6	4/6	0/2	0/3
ChReef	1/1	9/10	5/5		
ChRmine hIPS		6/19			
ChReef hIPS		18/18			

As you can see, unsuccessful attempts can be found in the case of ChRmine at any pacing rate. We still thought that an example with the same pacing rate would be more representative and less misleading and thus exchanged the example accordingly. Furthermore, we have done experiments in hPSC-derived cardiomyocytes expressing ChRmine and ChReef. Since these have a much lower spontaneous beating rate, all attempts could be performed at the same pacing rate (1 Hz) and we observed the same difference.

b) In regard to the new start of the electrical pacing simultaneous with the start of the continuous light pulse, we chose this way of stimulation since the light will induce an extrabeat when illumination starts after the refractory period of the previous beat. Starting both together, guaranteed that within the time window of 5 s illumination duration, we always tested the same amount of electrical pulses to be able to excite the cardiomyocytes or not.

7. No negative controls are shown anywhere in the paper. Especially the “vision restoration” experiments need a control with no opsin expression. Moreover given the use of PHP.eB viruses in these experiments, expression in other brain regions than the retina needs to be excluded. In general, histology is very limited. The example in panel a suggests that expression density was higher for ChReef. Expression needs to be quantified (see also next point).

We added examples for optical pacing of WT controls in the case of hPSC derived cardiomyocytes as well as representative examples of negative wildtype cells for the conduction block with 510 nm (n=4) and 630 nm (n=1).

We added histological analysis of transduced cells in the retina of ChRmine and ChReef injected mice (Supplementary figure: 14). We do not observe a difference between number of cells transduced between the two groups. Furthermore, we provided an example image of a coronal brain slice of an ChReef injected mouse used for the vision restoration experiments. Despite using PHP.eB we do not observe expression of ChReef in the brain except for RGC axons in the lateral geniculate nucleus and superior colliculus (Supplementary figure: 15).

In addition, we also checked in brains of the Mongolian gerbils used for IC recordings. Here we also do not find GFP expression (Supplementary Figure 22d).

Furthermore, we now included two additional mouse models of deafness to proof ChReef’s potential for optogenetic hearing restoration (Fig. 4f; S18)

8. Fig. 4: According to the methods, it seems that the data shown in panels h-r were obtained from two mice – one of them deafened. What is the purpose and why is this not indicated in the figure or the text? It also seems like deafening was not validated anywhere. I find it quite concerning that at those numbers of animals (n = 2) mixed treatments are used – especially in the case where two opsins are compared. Which mouse was ChReef and which one ChRmine injected? These experiments need to be repeated under proper conditions. In addition, histological characterization is not very conclusive. From Suppl. Fig. 9 it looks like ChReef expression was stronger than ChRmine expression, at least in the left cochlea. Can this explain the difference in performance? What is the number of animals compared? It is not indicated in the legend. This aspect needs to be properly evaluated and quantified. If better expression explains the better performance of ChReef, this is fine, but at this stage, cannot be judged properly.

We thank the reviewer for this comment. We performed additional experiments. Now the figure shows data only from mice expressing ChReef (n=3) and undergoing acute kanamycin treatment inducing

hair cell death. This was done to be sure that the measured response was originating for optical stimulation of SGNs not hair cells.

In addition, we improved our histological analysis for the ChReef and ChRmine comparison and provide a line profile analysis showing no difference in the expression between the two (Fig. 4c).

9. I am very skeptical about figure 5. It does not add any additional value to the manuscript (the relevant information is already in figure 4). It shows neither comparison to ChRmine, to any other opsin, or to auditory stimulation. Panels g-h are not even referred to in the main text. Why are they plotted? It is not explained why different animal numbers were used (5c n = 7, ABR recordings in suppl. fig. 14 n = 5). More importantly, the authors used a PHP.S capsid for viral transduction in neonatal animals. Thus, it is highly likely that the virus spread further into the brain and did not remain local in the cochlea. In fact, this is directly seen in suppl. fig. 11. Here, large photocurrents are observed in cells of the ventral cochlear nucleus. Thus direct light effects in the inferior colliculus from non-cochlear sources cannot be disentangled from those responses originating in the cochlea itself. Also the synaptic recordings in suppl. fig. 11 are problematic, because residual photocurrents and local network effects cannot be excluded. According to suppl. fig. 12 it seems that the virus even spread to the contralateral cochlea. These gerbil experiments weaken the manuscript and should be removed.

Thank you for your feedback. We have carefully reworked this section of results as well as figure 4. This includes data of additional experiments into the revised MS obtained from gerbils and marmosets. We consider this data and extended presentation a very important part of the study: gerbils and marmosets are key species of translational hearing research and the proof of concept achieved in both species is important for preparing clinical trials. We very much hope that the revised MS will convince the reviewer.

For the inferior colliculus recordings in the gerbil, we now moved to stimulations with an LED based multichannel optical cochlear implant first time operating green LEDs. In contrast to prior shown data in mice, recordings in the mid brain allow analysis of the spectral spread of excitation, which is the key advantage of the optical cochlea implant¹². The data shows that stimulation with an optical fiber as well as with the LED implant, spectral spread is comparable to acoustic pure tone stimulation. Furthermore, we now also included data of acutely deafened gerbils, mimicking the future cochlea implant patient.

In reference to this, we would like to apologize that it was not clear that the data from S11 was generated in mice injected with AAV2/9, while the inferior colliculus recordings were performed in Mongolian gerbils injected with PHP.S. To show that the ChR is limited to the spiral ganglion neurons, we provided an additional supplementary figure 22.

Finally, we now also provide proof of principle that our newly developed optogenetic tool is working in a non-human primate model (Fig. 5h-j; S25,26).

Minor

10. It is not clear why closing kinetics of 30 ms are reported in the abstract and discussion. The fastest off kinetics reported for ChReef are between 50 and 60 ms (Suppl. Fig. 4 and suppl. tables 1 and 2).

We thank the reviewer for making us aware of the missing description of our measurements at 36 °C, from which the closing kinetics of ~30 ms (35 ± 3 ms, n=6) was derived. We added the missing information to the results part.

11. Fig. 2: The legend states that 625 nm was used for photostimulation. However, given the optical filters used in this experiment, the light emitted from a 625 nm LED that reached the cells was between 626 and 644 nm. This is relevant with regard to the low desensitization of ChRmine in this wavelength range and therefore requires proper spectral characterization of the photocurrents as stated under main point 2.

We agree with the reviewer's remark and have changed the wavelength accordingly.

12. Fig. 4.: The caption mentions radiant flux in mW, but the legend is given in uJ

Thanks for spotting: changed to radiant energy in caption.

13. Suppl. Fig. 4.: why are data points missing? Is this due to absence of currents at the reversal wavelength? If so, why is this not an issue with ChRmine photocurrents?

The missing τ_{off} values are close to the reversal potential. As the photocurrent close to the reversal potential is small, the quality of the mono-exponential fits was compromised by suboptimal signal-to-noise ratios. In revised Figure S4 (now Fig. S6), we accordingly removed the τ_{off} value of ChRmine wt at 0 mV.

14. Suppl. fig. 8: why are the red lines in drawn over the data?

Thanks for spotting this. We reworked the figure.

Suppl. Fig. 11.: "AVCN" nowhere defined

Thanks for spotting this. The figure caption now includes a definition of the anteroventral cochlear nucleus.

References

1. Cehajic-Kapetanovic, J. *et al.* Restoration of Vision with Ectopic Expression of Human Rod Opsin. *Current biology : CB* **25**, 2111–22 (2015).
2. Berry, M. H. *et al.* Restoration of high-sensitivity and adapting vision with a cone opsin. *Nature Communications* **10**, 1221 (2019).
3. Lagali, P. S. *et al.* Light-activated channels targeted to ON bipolar cells restore visual function in retinal degeneration. *Nat Neurosci* **11**, 667–675 (2008).
4. Strettoi, E. & Pignatelli, V. Modifications of retinal neurons in a mouse model of retinitis pigmentosa. *Proc. Natl. Acad. Sci. U.S.A.* **97**, 11020–11025 (2000).

5. Strettoi, E., Porciatti, V., Falsini, B., Pignatelli, V. & Rossi, C. Morphological and functional abnormalities in the inner retina of the rd/rd mouse. *J. Neurosci.* **22**, 5492–5504 (2002).
6. Strettoi, E., Pignatelli, V., Rossi, C., Porciatti, V. & Falsini, B. Remodeling of second-order neurons in the retina of rd/rd mutant mice. *Vision Res.* **43**, 867–877 (2003).
7. Maguire, A. M. *et al.* Safety and efficacy of gene transfer for Leber’s congenital amaurosis. *New England Journal of Medicine* **358**, 2240–2248 (2008).
8. Bennett, J. *et al.* AAV2 gene therapy readministration in three adults with congenital blindness. *Sci Transl Med* **4**, 120ra15 (2012).
9. Kleinlogel, S. *et al.* Ultra light-sensitive and fast neuronal activation with the Ca²⁺-permeable channelrhodopsin CatCh. *Nat Neurosci* **14**, 513–518 (2011).
10. Mager, T. *et al.* High frequency neural spiking and auditory signaling by ultrafast red-shifted optogenetics. *Nat Commun* **9**, 1750 (2018).
11. Emiliani, V. *et al.* Optogenetics for light control of biological systems. *Nat Rev Methods Primers* **2**, 1–25 (2022).
12. Dieter, A., Duque-Afonso, C. J., Rankovic, V., Jeschke, M. & Moser, T. Near physiological spectral selectivity of cochlear optogenetics. *Nature Communications* **10**, 1962 (2019).

Reviewer #1 (Report for the authors (Required)):

The Authors performed an extensive revision of the manuscript, properly addressing almost all of the points raised by the reviewers. A substantial amount of new data and analyses has been added and this improved significantly the quality of study results and conclusions. The newly introduced descriptive and experimental material (including the demonstration of functional vision restoration upon the studied optogenetic approach), supplementary figures and references further supported the working hypothesis and scientific importance of the obtained results.

They don't address some of the comments as they state that existing knowledge of the animal models makes it unnecessary to analyze potential changes induced by the therapy on retinal morphology. Inflammatory responses should also be investigated. This will certainly be part of their future work.

José-Alain Sahel

We would like to thank Professor Sahel for appreciating the work and the revisions and his advice that helped us to improve the manuscript. We understand the need for further studies in the future and take strong motivation from his encouragement.

Reviewer #2 (Report for the authors (Required)):

The authors addressed all my concerns and comments in full by providing new experimental results and revising the text. The revised manuscript was substantially improved. The quality and data representation in Figures were also improved. In addition, the authors demonstrated the possibility of optogenetic hearing in common marmosets, which is crucial for the clinical value of ChReef applicability.

We would like to thank the reviewer for appreciating the work and the revisions and his advice that helped us to improve the manuscript. We have carefully addressed the additional points raised during re-review: please see our point-per-point response below and the revised MS.

Unfortunately, the manuscript contains a major issue. In the checklist, the authors checked two main items: "The exact sample size (n) for each experimental group/condition, given as a discrete number and unit of measurement", "A statement on whether measurements were taken from distinct samples or whether the same sample was measured repeatedly". However, the figure legends for Figure 1, Figure 4, Supplementary Figures 1,2,4,6 contain neither a unit of measurement nor a statement on whether measurements were taken from distinct samples or whether the same sample was measured repeatedly". However, Supplementary Figure 5 contains proper description of both units and repeated measurements.

We now reworked the indicated sections and added information on the unit of measurement and statements on whether measurements were taken from distinct samples or whether they were measured repeatedly.

Furthermore, I have two concerns regarding data and code availability statements: Data availability statement: The data that support the findings of this study is available from the corresponding authors upon reasonable request. Code availability statement: The code used for analysis is available from the corresponding authors upon reasonable request. Why not upload the source data files for Supplementary Figures and the most critical raw dataset to a public repository such as FigShare or Zenodo? This is a great work, but the lack of immediate transparency may complicate the independent reproducibility of the results by other researchers. Do the authors have any concerns regarding sharing the most critical files on public file repositories and providing DOI in the manuscript? Moreover, I did

not see source data files for the figures. I thought providing source files during the review process was mandatory for all Nature journals.

We agree with the concerns of the reviewer and now uploaded data and code on Zenodo to follow the idea of open science. The information can be found under the following doi: [10.5281/zenodo.13963753](https://doi.org/10.5281/zenodo.13963753)

Reviewer #3 (Report for the authors (Required)):

The authors have addressed a number of concerns and added a lot of new analyses to the manuscript. While the manuscript has improved considerably, there are still some remaining questions. It is essential to address these points, given the prominent claims it makes about the breakthrough applications attributed to the new ChReef.

We would like to thank the reviewer for appreciating the work and the revisions and the advice that helped us to improve the manuscript. We have carefully addressed the additional points raised during re-review: please see our point-per-point response below and the revised MS.

Now that the authors included raw traces in figure 1, it looks like ChReef has overall larger photocurrents, both stationary, but also peak (1e). This notion is also supported by the fact that photocurrent density is almost 5x higher compared to ChRmine, whereas the ratio between stationary and peak photocurrent is improved only 3x (see suppl. Table 1). It is important to address this point and explain this overall improved photocurrent density. Is it due to better membrane trafficking / expression of the opsin? Single channel conductance is ruled out as a candidate mechanism, as it is identical between ChRmine and ChReef (1j). Since this is a biomedical engineering journal, there must be a rigorous explanation and experimental confirmation of the observed effects. As it is now, the community will not get sufficient insight into the actual improvements of the tool itself.

We thank the reviewer for raising this point. Given the variability of the data shown in Fig. 1a and Fig. 1b, we deemed the observation that the photocurrent density of ChReef-TS-EYFP-ES in NG cells is 4.5x higher compared to ChRmine-TS-EYFP-ES, whereas the stationary-to-peak-ratio is improved only 2.8x of minor relevance. In order to address the reviewer's comment, we have performed additional experiments in which we compared the plasma membrane targeted expression of ChRmine-TS-EYFP-ES and ChReef-TS-EYFP-ES in NG cells (Fig S7). From the fluorescence line profile analysis and the quantification of the average fluorescence within the cells by the determination of CPCF/area values as described in the materials and methods section, it becomes apparent that there is no considerable difference in plasma membrane targeted expression of ChReef and ChRmine in NG cells (Fig S7). We now provide this important finding in the main text.

Even though figure 1 has improved, there are still some problems with it. Labels and legends are still inside the plotting area. Especially in panels c and d this is problematic as the horizontal significance lines are inside the range of plotted data points. Another problem is the re-use of the same data in panels b and c (ChRmine and ChReef). Even though the authors indicate that the data are re-plotted, they cannot use them twice for different statistical tests. Either all variants shown in b and c need to be compared against each other in one test, or ChReef and ChRmine data have to be compared to those measurements that were done in the same experimental cohort. The same applies to supplementary tables 1 and 2. It is also not clear why statistical differences are shown by asterisks in some panels and not in others (e.g. a, d,).

We thank the reviewer for the helpful comments and accordingly provide a revised version of Figure 1. In panel b of the revised Fig. 1, we now provide a statistical comparison of all ChR variants, which were previously shown in panels 1b and 1c. Supplementary Table 1 shows the comparison (Stationary-

Peak-Ratio, EC₅₀ values, stationary current densities and τ_{off} values in NG cells) of the optimized ChRmine variants to ChRmine wt. Supplementary Table 2 shows the comparison (stationary current densities and τ_{off} values in NG cells) of the best ChRmine variant (ChReef) to other state-of-the-art ChRs.

Figure 3: The new behavioral analyses should be shown in the main manuscript (and the typos in the figure caption should be fixed). This would add relevance to the vision restoration aspect of the paper.

We appreciate that the reviewer acknowledges the value of this new data. As advised, we have added behavioral data to the vision restoration figure.

The authors now show new experiment with a multichannel optical cochlear LED implant emitting green light used for the first time, which is a major improvement compared to blue light emitting ones. This aspect is important, especially since these LEDs match the action spectrum of ChReef and thus, require less light energy (enabling deeper tissue penetrance). This aspect is highly relevant and should be presented more prominently in the main paper. Related to this, Supplementary fig. 24d is not readable due to low quality.

Done, we appreciate that the reviewer acknowledges the value of this new data. We also updated figure 24d allowing better readability.

Figure 4d now shows a comparison of opsin-expressing cells in the cochlea. The density of ChRmine expressing cells appears to be higher on average (approx. 25 vs. 15 GFP+SGNs/1000 μm^2). In the text, it is stated that "Expression of both ChR variants under the control of the human synapsin promoter was found in approximately 75% of the SGNs". This contradicts the plot in panel d – at least if we assume that the total density of SGNs was the same between animals. Or were more cells lost with ChRmine over time, despite initial similar transduction rate?

We were able to analyze some additional cochleae to check for the reviewer's concerns. We observe a difference when comparing transduction rate per SGNs of ChRmine and ChReef, but only in the mid turn of the cochlea. We indicate this in the main figure and changed the text accordingly.

Figure 5j: it is not obvious in which of the traces there is an objectively measurable response. It looks like we see one single experiment with single trials at different light intensities. The authors state that "auditory responses were observed from 4.7 mW" (is this the 4th line from the bottom?). According to the methods section, "one of the (...) 2 waves (oABR) was reliably visible." What is meant by "reliable", given that I see only one trial? Where is this wave in the 4.7 mW trial and where is it in the ones with higher irradiance?

We have now also mentioned in the legend that each trace is the average of 1000 trials (was in the methods and is standard in the field). Moreover, we have connected the numbers to respective traces for ease of orientation. Visual detection of ABR waves is still the standard of the field and reliable refers to the fact that the same wave can be found across suprathreshold traces where its latency typically gets shorter as the intensity increases.

Except for one t-test in figure 4 g, I could not see any statistical analyses of all the auditory data presented in the entire manuscript (figures 4 and 5 and supplementary figures 18-26). There are many quantitative plots comparing ChRmine and ChReef and various manipulations or expression levels where it remains unclear what the data mean. For example, suppl. fig 18, 19, 21, fig. 4 c, d, h, i, 5 c all need quantitative statistics. If no significant differences are found, p values need to be shown and the reader needs to know why superiority of ChReef over ChRmine may be less pronounced in the auditory system. Along these lines, it looks like there is only a clear difference between ChRmine and ChReef at oABRs evoked with 594 nm.

The reviewer has correctly concluded from our presentation of the data at 594 nm: this is exactly what we aimed to convey. We now added p values for all indicated figures.

All the experiments that follow are showing some convincing improvements towards optogenetic hearing restoration in general. However, it remains unclear to what extent this is due to ChReef being superior to ChRmine. All gerbil and marmoset experiments are done exclusively done with ChReef. This needs to be more clearly pointed out by mentioning that they opted for the best opsin/virus combination to realize applications in the auditory system that failed with other constructs previously and that similar results may be achieved with ChRmine using the improved parameters (PHP.S capsid, early virus transduction, green LEDs, etc.).

As the reviewer correctly pointed out, we consider the demonstration of sub- μ J thresholds for ChReef and ChRmine an important finding that better matches the radiant flux levels available from microscale emitters such as the green LEDs in this study. ChReef tended to allow still lower thresholds which was significant for 594 nm. We have followed the advice of the reviewer to communicate the purposeful focus on ChReef for the more demanding studies with gerbils and marmoset.

It is nowhere mention whether ChReef will be made publicly available. Ideally, the authors should deposit the DNA on a public repository, so that the community can benefit from the new tool in an unrestricted way.

Absolutely, we will reposit it at Addgene.

Color schemes are not consistent throughout the manuscript. Sometimes, magenta/green, sometimes red/green lookup tables are used. The latter is not recommended due to discrimination of people with impaired color vision.

We thank the reviewer for this comment. We now changed the look up table to magenta/ green for all provided histology.

Reviewer #3:

After this new round of revisions, some issues remain unaddressed. Contrary to what is stated in the response to my points regarding Figure 5j, no changes were made to the figure panel or the legend. Actually, the legend is misleading. "oABR recordings in a common marmoset resulting in successful activation of the auditory pathway." is not correct. Rather, oABR recordings in a common marmoset resulting FROM successful activation of the auditory pathway is correct. It is still not apparent where the visually detected ABR waves occur in the 4.7 mW trial (or which one the 4.7 mW trial even is). Every trial needs to be labeled with the corresponding radiant flux. The color code is useless. The authors should rather color the non-response trials in one color and the trials showing a response in another color. The response itself needs to be labeled by some kind of indicator (arrow head, etc.), so that it becomes evident to a non-expert reader. The first response, which is clearly visible to me only occurs at around 15.1 mW. For plausibility, the response latencies should also be compared to electrically evoked ones (can be literature values).

We would like to apologize. We by now realized that the uploaded file did not include the final version of the figure which we prepared for revision round 2. We have now labelled each individual trace in the colors, as requested by the reviewer, with its correspondent radiant flux. We have added graphical aids to highlight ABR waves. As we are not aware of any published work describing electrically evoked ABRs we have chosen to present the latencies of the waves as a function of radiant flux, compared to acoustically evoked ABRs from the literature (Harada & Tokuriki, 1997; doi: 10.1292/jvms.59.561) in supplementary figure 25.

The authors also did not seriously address my concern regarding other factors that helped improve optogenetic control of the auditory system. In this study, several factors were optimized compared to their previous work (virus capsids, expression/transduction protocols/light sources) and this needs to be made clear to the reader. The authors mention in their response that they now "communicate the purposeful focus on ChReef for the more demanding studies with gerbils and marmosets". First, I cannot see what has changed in the manuscript from the previous version. Second, the marmoset ABR responses required supra-ul light energies and are therefore not yet compatible with the microscale green LED emitters. As mentioned above, these aspects need to be critically discussed. The discussion section is currently not worth its title and still reads like an advertisement section with some trivial suggestions on how to improve ChReef. As mentioned in the previous two rounds, the manuscript clearly shows a greatly improved optogenetic approach for stimulating multiple systems. However, the discussion also needs to clearly point out the overall improvements achieved with ChRmine itself, new light sources, new virus serotypes, etc. It also needs to take into account the limitations and discrepancies of the study (just one example: much larger difference in steady-state photocurrent densities between ChRmine and ChReef in NG cells (about 5x) compared to HEK cells (about 2x) and only small differences in the auditory system).

We have revised and extended the discussion following the advice of the reviewer and are thankful for the help with improving the manuscript.

However, we respectfully disagree with the statements of the reviewer that suggest we did not carefully compare ChReef to ChRmine, underestimate ChRmine and overemphasize ChReef's utility. First of all, as shown in Figure 1 and stated throughout the MS, we recognize the relatively large single channel conductance of ChRmine and the ChRmine mutant ChReef described here, as beneficial for the various applications including the ones in the present MS. We have introduced the landmark study of Marshell et al., 2019 discovering, characterizing and applying ChRmine very prominently as the starting point for our work. Second, we have carefully compared the two ChRs by using the same experimental approach in parallel experiments (e.g. cells, same virus, promoter, experimental set-up

etc.). This way, we scrutinized differences in photocurrents and in the applications of ChRmine and ChReef to cardiomyocytes, retina and cochlea, which we largely attribute to the reduced desensitization of ChReef.

In regard to the statement that the marmoset experiments are not a proper comparison between ChReef and ChRmine, we would like to further clarify on this point: First, we consider the marmoset experiment as preliminary, as stated in the manuscript, and has yet to be further optimized. Second, given the ethical considerations of working with non-human primates we focused on the improved ChR variant, and reported the successful case along with the challenges of moving from rodents to primates.

Reviewer 3

The authors have now labeled the trials in Figure 5j and indicated the ABR waves. They also compare these waves to aABRs from the literature. Thank you.

We thank the reviewer for the appreciation of the work and continued effort.

However, since there is apparently no objective/quantitative way to detect ABR waves, I refuse to believe that the wave in the 4.7 mW trial represents a real signal. I leave it to the editors to decide how to handle this.

As mentioned before, this is a subjective analysis and done here by several observers. To please the reviewer we have removed the asterisk from the 4.7 mW trace and now state the threshold to be ≤ 6.2 mW (≤ 6 μ J).

Regardless, the statement (line 489-490) "Together, these data prove the principle of nano-Joule and sustained optogenetic stimulation of the rodent and primate auditory pathway,..." is still not correct. $4.7 \text{ mW} \cdot 1 \text{ ms} = 4.7$ micro joules. This is not in the nano-joule range, as stated in the manuscript. This has now been specified accordingly.

"Together, these data prove the principle of nano-Joule and sustained optogenetic stimulation of the **rodent auditory pathway (micro-Joule for the preliminary primate report)**, capitalizing on the comparatively high single-channel conductance and sustained photocurrents of ChReef."

I have criticized these inaccuracies in all rounds of review. But they are still not completely removed from the paper.

There is also an incorrect figure reference in line 481 (Fig. SX) of the newly added text.

done

Although still quite compact, the discussion has improved considerably.

As a last remark, my criticism was not about underestimating ChRmine. It was about not considering many other factors that have contributed to the general improvement of optogenetics in the auditory system. It is clear that ChReef is a great tool. However, because other factors also improved (more potent viruses, improved light delivery), some of the performance improvement would have been seen with other opsins, as well. There is no harm in discussing this openly.

We have no problem discussing things openly as the reviewer might have realized but needed to deal with the word limit imposed by the journal. So, we thank the reviewer for letting us elaborate further. In response to the reviewer's comment, we have added the following section:

"We note that optical SGN thresholds also depend on the level of plasma membrane expression of the ChR achieved. This is co-determined by the time point, route and dose of viral vector administration as well as the choice of vector and promoter and last not least, the membrane targeting of the ChR. This study found lower (nano-Joule) thresholds in three rodent models of deafness (mouse: ototoxic and genetic, gerbil: ototoxic) with two different AAVs (mouse: AAV2/9, gerbil: AAV.PHP.S) applied early postnatally with a doses of $\sim 3\text{-}7 \times 10^{13}$ viral genomes achieving expression rates similar to previous studies employing similar titers of other powerful vectors AAV2/6, AAV-PHP.B and AAV-PHP.eB and the same human synapsin promoter^{14,18,40,42}. Like in previous studies^{14,40,42} we employed Kir2.1 sequences to improve the membrane targeting. Together, we conclude that the lower thresholds are primarily caused by the favorable properties of ChReef rather than by differences in the AAV-mediated gene transfer."